 SciPost Phys. Lect. Notes 107 (2025)

# Engineering of anyons on M5-probes via flux quantization

## Hisham Sati[1,2,*] and Urs Schreiber[1,†]

**1** Mathematics, Division of Science; and Center for Quantum and Topological Systems,
NYUAD Research Institute, New York University Abu Dhabi, UAE
**2** The Courant Institute for Mathematical Sciences, NYU, NY

* hsati@nyu.edu , † us13@nyu.edu

## Abstract

These extended lecture notes survey a novel derivation of anyonic topological order (as seen in fractional quantum Hall systems) on single magnetized M5-branes probing Seifert orbi-singularities ("geometric engineering" of anyons), which we motivate from fundamental open problems in the field of quantum computing. The rigorous construction is non-Lagrangian and non-perturbative, based on previously neglected global completion of the M5-brane's tensor field by flux-quantization consistent with its non-linear self-duality and its twisting by the bulk C-field. This exists only in little-studied non-Abelian generalized cohomology theories, notably in a twisted equivariant (and "twistorial") form of unstable Cohomotopy ("Hypothesis H"). As a result, topological quantum observables form Pontrjagin homology algebras of mapping spaces from the orbi-fixed worldvolume into a classifying 2-sphere. Remarkably, results from algebraic topology imply from this the quantum observables and modular functor of Abelian Chern-Simons theory, as well as braid group actions on defect anyons of the kind envisioned as hardware for topologically protected quantum gates.

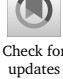

## Contents

SciPost Phys. Lect. Notes 107 (2025)

# 1   Motivation: Better anyon theory

While the hopes associated with the idea of *quantum computing* [104] [59] are hard to over-state [52] [10] [118], there are good arguments that commercial-value quantum computing will ultimately require quantum hardware exhibiting *anyonic topological order* [164] [131]. But microscopic theoretical derivations, from first principles, of such anyonic quantum states in strongly-coupled quantum systems had remained sketchy, which may explain the dearth of experimental realizations to date.

What we review here (based on [134] [137] [54]) is a rigorous theoretical account via "ge-ometric engineering on M-branes" subject to a previously neglected step of "flux-quantization" (the latter surveyed in [132]).

First, we expand on the motivation a little further:

**Ultimate need for Topological Quantum Protection.** Despite the fascinating reality of presently available Noisy Intermediate-Scale Quantum computers (NISQ [116]) and despite the mid-term prospect of their stabilization at the software-level via Quantum Error Correction (QEC [89] [117], at heavy cost of available system scale), serious arguments [77] [30] [87] [31] [32] [68] [51] [150] and experience [22] suggest that large-scale quantum computation is hardly attainable by incremental optimization of NISQ architectures, but [23][1] that more fundamental quantum principles will need to be exploited – notably *topological* error *protec-tion* already at the hardware-level [82] [46] [141] [140] in order to suppress quantum errors occurring in the first place.

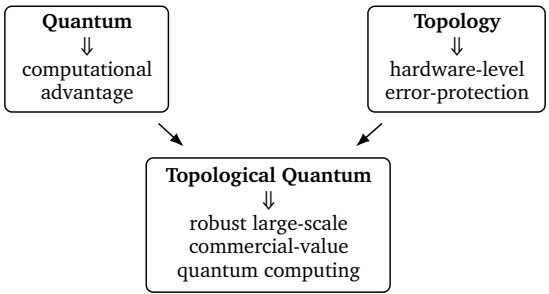

Figure 1: **Topological quantum.** In order to practically harness the computational power of quantum processes, quantum states need to be stabilized against decohering environmental noise. Apart from using quantum error correction at the software level, a plausibly necessary way to do so is by using topological quantum processes preventing quatum errors right at the hardware level.

While topological quantum protection is thus possibly indispensable for achieving commercial-value quantum computing, its ambitious development, in theory and practice, is in fact far from mature, is in need of new ideas and of further analysis, and leaves much room for development. Since this is not always made clear, to amplify this point:

---

[1] [23]: "The qubit systems we have today are a tremendous scientific achievement, but they take us no closer to having a quantum computer that can solve a problem that anybody cares about. [...] What is missing is the breakthrough [...]  bypassing quantum error correction by using far-more-stable qubits, in an approach called topological quantum computing."

**(i) Theoretical challenges:** While quantum theorists now routinely deal with the algebraic structure (namely: braided fusion categories) commonly *expected* [83] to describe interaction of anyon species *in toto*, the *microscopic* first-principles understanding of the formation of anyonic topological order as solitonic states in the many-body (electron) dynamics of quantum materials has remained at most sketchy, even in the best-understood case of the fractional quantum Hall effect [144], cf. [72].[2]

In fact, this is an instance of the general open problem of analytically establishing gapped bound states in any strongly coupled/correlated quantum system: The problem of formulating non-perturbative quantum field theory [6] [34]. The analogous issue in particle physics (there called the *Yang-Mills mass gap* problem [105]) has been recognized as being profound enough to be declared one of seven "Millennium Problems" [20].

**(ii) Practical challenges:** But without a robust theoretical prediction of anyonic solitons in actual quantum materials, it remains unclear where and how to look for them. As an unfortunate result, experimentalists have turned attention to mere stand-ins, such as "Majorana zero modes" at the ends of super/semi-conducting nanowires ( [81] [94] which, even if the doubts about their detection were to be removed [24], are by construction immobile and hence do not serve as hardware-protected quantum braid gates) and quantum-simulation of anyons on NISQ architectures ( [70] [43, Fig. 5], which might serve as software-level QEC but again offers no hardware-level protection.

In short: **Foundation and implementation** of topological quantum computing as a plausible long-term pathway to actual quantum value **deserves and admits thorough re-investigation**.

Concretely, the intrinsic tension haunting the traditional quantum computing paradigm is (cf. [17, p 272] [150, p 3]) that:

> (i) quantum gates are *implemented via interaction* of subsystems,
>
> (ii) while quantum coherence requires *avoiding all interaction*.

**The idea of topological protection** is to cut this Gordian knot by *quantum gates operating without interaction*. The physical principle that allows this to work [3] [4] [46, p 6] [110, p 50] is the *quantum adiabatic theorem* [120]: Gapped quantum systems frozen at absolute zero in one of several ground states, but dependent on external parameters, will defy interaction with noise quanta below the energy gap and yet have their ground state transformed by sufficiently gentle tuning of the parameters: a *holonomic quantum gate*. This is *topological* if it is invariant under local deformations of parameter paths, and thus protected also against classical noise. For an *anyonic braid gate* the parameters in question are the positions of defects in a 2-dimensional transverse space within a quantum material.

The remaining problem is to develop a precise mathematical theory describing these *anyons*.

**Improved Anyon Models via Geometric Engineering on M-branes.** A remarkable approach to the otherwise elusive microscopic analysis of such strongly-coupled/correlated quantum systems emerges in the guise of "geometric engineering" [80] [13] of quantum fields on "M-branes" probing orbifold singularities, whereby the given dynamics is (partially) mapped

---

[2] [72, p. 3]: "Though the Laughlin function very well approximates the true ground state at $v = 1/q$, the physical mechanism of related correlations and of the whole hierarchy of the FQHE remained, however, still obscure. [...] The so-called HH (Halperin-Haldane) model of consecutive generations of Laughlin states of anyonic quasiparticle excitations from the preceding Laughlin state has been abandoned early because of the rapid growth of the daughter quasiparticle size, which quickly exceeded the sample size. [...] the Halperin multicomponent theory and of the CF model advanced the understanding of correlations in FQHE, however, on phenomenological level only. CFs were assumed to be hypothetical quasi-particles consisting of electrons and flux quanta of an auxiliary fictitious magnetic field pinned to them. The origin of this field and the manner of attachment of its flux quanta to electrons have been neither explained nor discussed."

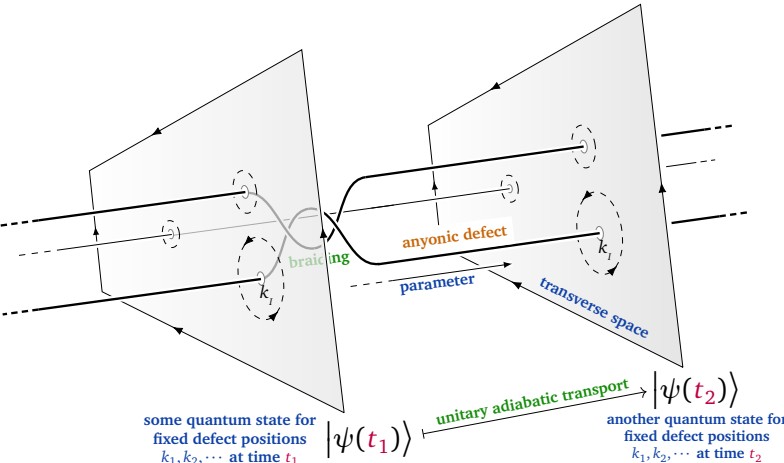

Figure 2: **The idea of topological quantum gates** by anyon braiding: Given an effectively 2-dimensional quantum material with point-like defects and degenerate ground states, the quantum adiabatic theorem implies that sufficiently slow (adiabatic) movement of the defect positions around each other (e.g. by externally tuning the material's properties) entails a unitary transformation on the Hilbert space of ground states. If these unitaries can be made to depend only on the homotopy class of the defect paths relative their endpoints, hence only on the *braids* formed by their worldlines, then their operation is topologically protected against noise in the operation of the gate.

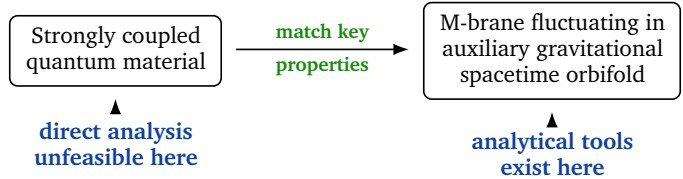

Figure 3: **Geometric engineering of quantum systems on M-branes** provides tools for analyzing otherwise elusive strongly coupled/correlated quantum phenomena.

onto the fluctuations of Membranes (whence *M-theory* [28]), and of higher-dimensional "M5-branes" [54], propagating within an auxiliary higher-dimensional gravitating spacetime orbifold [124].

This procedure is most famous in the (unrealistic) limit of large rank and hence of large numbers $N \to \infty$ of coincident such branes, where it extracts quantum correlators and quantum phase transitions entirely from classical gravitational asymptotics ("holographic duality" [1]). The application to quantum materials [163] [64] is now well-studied, notably in the case of quantum critical superconductors engineered in M-theory [67] [49] [50] [60] [26] [27] [2].

But we have established [54] [134] [135] [137] that after implementing a previously neglected step of "flux quantization" [132] on the M5-brane worldvolume, there provably appear general solitonic and specifically anyonic quantum states already in the more realistic situation of single ($N = 1$) coincident branes. (Similar results for $N = 2$ had previously only been conjectured [18] by appeal to the expected but notoriously undefined effective quantum field theory on coincident M5-branes.)

Moreover, in [138] we have proven that the resulting topological quantum states and their topological order agrees in fine detail with the expectations for FQH systems as also predicted by Abelian Chern-Simons theory, while at the same time predicting that and how *defect* anyons

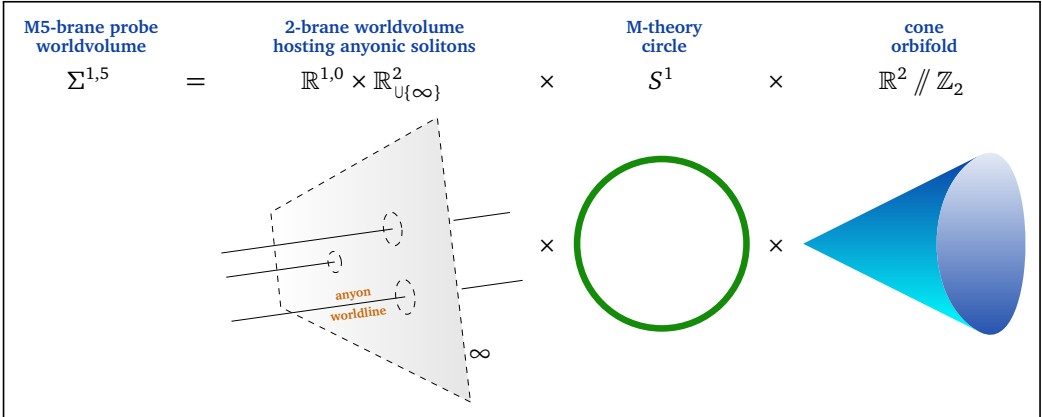

Figure 4: Brane diagram for **geometric engineering of anyons** on single M5- branes wrapping an orbi-singularity [137]. It is a subtle mechanism of *flux-quantization* [132] of the self-dual tensor-field on the M5 [54] that stabilizes [134] its anyonic soliton configurations.

in these systems may exhibit the much-desired non-Abelian braiding. It is these results that we survey in the present lecture notes.

Concretely, here we review and explain how this works, aimed at an audience assumed to be familiar with the general mechanism of *flux quantization* as surveyed in [132].

But first to briefly recall the traditional theory of fractional quantum Hall anyons:

**Quantum Hall effect** (cf. [115] [15] [144] [111]). In a very thin (atomic multi-layer) and hence effectively 2-dimensional sheet $\Sigma^2$ of (semi-)conducting material carrying magnetic flux density $B$, the energy of electron states is (cf. [149, (4-12)]) quantized by *Landau levels* $i \in \mathbb{N}$ as

$$E = \hbar\omega_B\left(i + \tfrac{1}{2}\right),$$

where each Landau level comprises of one state per magnetic flux quantum:

$$n_{\mathrm{deg}} = B/\Phi_0\,,$$

and the Lorentz force on a longitudinal electron current $J_x$ at filling fraction $\nu$ is compensated in equilibrium by an electric *Hall field*

$$E_y = \tfrac{1}{\nu}J_x\,.$$

**Integer quantum Hall effect.** Therefore, Fermi's theory of idealized *free* electrons predicts the system to be a conductor away from the energy gaps between a completely filled and the next empty Landau level, hence away from the number of electrons being integer multiples of the number of flux quanta, where longitudinal conductivity should vanish.

$$n_{\mathrm{el}} = \nu B/\Phi_0\,, \qquad \nu \in \mathbb{N}\,.$$

This is indeed observed — in fact, the vanishing conductivity is observed in sizeable neighborhoods of the critical filling fractions ("Hall plateaux", attributed to subtle disorder effects).

**Fractional quantum Hall effect** (FQHE). But in reality, the electrons are far from free. While there is little theory for strongly interacting quantum systems, experiment shows that the Fermi idealization breaks down at low enough temperature, where longitudinal conductivity also decreases in neighborhoods of certain *fractional* filling factors $\nu$.

$$\nu \in \mathbb{Q}\,, \quad \text{prominently for} \quad \nu = 1/K\,, K \in 2\mathbb{N}+1\,.$$

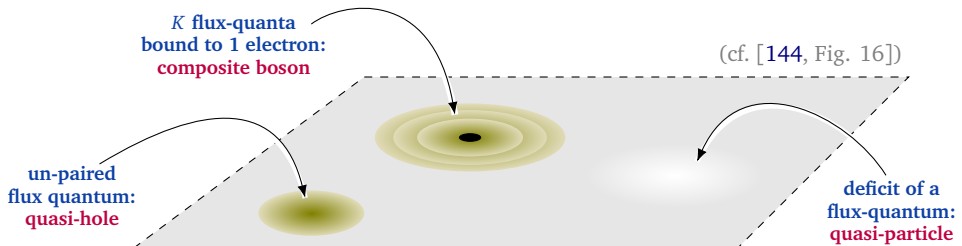

Figure 5: **Anyons in fractional quantum Hall systems** ("quasi-holes") are (vortices in the electron gas corresponding to) surplus magnetic flux quanta on top of a state of exact rational *filling fraction* where each electron is coupled/paired in some subtle way to a fixed number of flux quanta. Compare with Fig. 13.

The traditional heuristic idea is that at these filling fractions the interacting electrons each form a kind of bound state with $K$ flux quanta, making "composite bosons" (cf. [165]) that, as such, condense to produce an insulating mass gap, even inside the Landau level.

**Anyonic quasi-particles.** This heuristic model suggests that in the Hall plateau neighborhood *around* such filling fraction, there are *unpaired* flux quanta effectively "bound to" $1/K$ th of a (missing) electron: called "quasi-particles" ("quasi-holes"). These quasi-particles/holes evidently have fractional charge $\pm e/K$ and are expected to be anyonic with fractional pair exchange phase $e^{i\pi/K}$. This phase has been experimentally observed [103].

**Effective Abelian Chern-Simons theory.** The traditional ansatz for an effective field theory description of $K$-fractional quantum Hall systems postulates that the effective field is a 1-form potential $a$ for the electric current density 2-form $J$, itself minimally coupled to the *quasi-hole current* $j$, and with effective dynamics encoded by the level $= k = K/2$ Chern-Simons (CS) Lagrangian [165][155]:

$$
\begin{aligned}
\text{Electron current density 2-form} \quad & J &=& \quad \vec{J} \lrcorner \, \mathrm{dvol} \quad =: \quad \mathrm{d}\,a\,, \quad \text{Effective gauge field} \\
\text{Quasi-particle current density 2-form} \quad & j &=& \quad \vec{j} \lrcorner \, \mathrm{dvol}\,, \\
\text{Background flux density 2-form} \quad & F &=& \quad \mathrm{d}A\,, \quad \text{External gauge field} \\
\text{Effective Lagrangian density 3-form} \quad & L &:=& \quad \tfrac{K}{2}\, \underbrace{a\,\mathrm{d}a}_{\mathrm{CS}(a)} - \underbrace{A\,\mathrm{d}a + a\,j}_{AJ}\,. \quad [155, (2.11)]
\end{aligned}
$$

Its Euler-Lagrange equations of motion

$$
\frac{\delta L}{\delta a} = 0 \quad \Longleftrightarrow \quad \boxed{J = \tfrac{1}{K}\big(F - j\big),}
$$

in the case of longitudinal electron current and static quasi-particles

$$
\begin{aligned}
J &\equiv J_0\,\mathrm{d}x\,\mathrm{d}y \; - \; J_x\,\mathrm{d}t\,\mathrm{d}y\,, \\
j &\equiv j_0\,\mathrm{d}x\,\mathrm{d}y\,, \\
F &\equiv B\,\mathrm{d}x\,\mathrm{d}y \; - \; E_y\,\mathrm{d}t\,\mathrm{d}y\,,
\end{aligned}
$$

express just the hallmark properties of the FQHE that we saw above, at filling fraction $\nu = 1/K$:

$$\Longleftrightarrow \begin{cases} J_x &=& \frac{1}{K} E_y & \Longleftrightarrow & \text{Hall conductivity law at } 1/K \text{ filling,} \\ J_0 &=& \frac{1}{K} B & \Longleftrightarrow & \text{each electron binds to } k \text{ flux quanta, but} \\ && -\frac{1}{K} j_0 & & 1/K\text{th electron missing for each quasi-hole.} \end{cases}$$

**Conceptual problems.** However this can only be a *local* description on a single chart (as is common for Langrangian field theories): Neither $J$ nor $F$ may admit global coboundaries $a$ and $A$, respectively. Instead, both must be subjected to some kind of flux-quantization. For $F$ this must be classical Dirac charge quantization, which however is incompatible with integrality of $J$ when $k \neq 1$ (cf. [158, p. 35] [148, p 159]). But without this, the implications break concerning topological order from Abelian CS theory (ground state degeneracy, modular functoriality, ...).

Therefore we must ask:

**Question:** *Is there a non-Lagrangian theory for quasi-particles of properly flux-quantized FQH systems?*

**Answer:** Yes!:

**The main result** to be discussed here is that the key features of the anyonic topological order as seen in fractional quantum Hall systems are consistently, rigorously and naturally reflected by the topological light-cone quantization of the self-dual tensor field on M5-brane probes of certain orbi-singularities in 11D supergravity — once the subtle (non-Abelian) flux-quantization of this field is properly taken care of, which is the key step that has not previously received attention. This is what we explain below.

**Further aspects.** In fact, fractional quantum Hall systems exhibit further remarkable properties which have not previously been reflected in their effective (Chern-Simons) descriptions, but which are naturally reflected in the M5-brane model, among them *hidden supersymemtry*. We close this introduction by briefly indicating this phenomenon.

**$N$-Electron ground states of quantum Hall systems.** While a microscopic derivation of fractional quantum Hall ground states $\Psi$ remains missing, phenomenologically successful Ansätze exist:[3]

- At odd filling fraction $\nu = 1/q$, $q \in 2\mathbb{N} + 1$, the **Laughlin wavefunction**

$$\Psi_{\text{La}}(z^1, \cdots, z^N) := \prod_{i<j} (z^i - z^j)^q \exp\left(-\frac{1}{\ell_B^2} \sum_i |z^i|^2\right).$$

- At even filling fraction $\nu = 1/q$, $q \in 2\mathbb{N}$, the **Read-Moore wavefunction**

$$\Psi_{\text{RM}}(z^1, \cdots, z^N) := \text{Pf}\left(\frac{1}{z^{\bullet_1} - z^{\bullet_2}}\right) \Psi_{\text{La}}(z^1, \cdots z^N).$$

Here the *Pfaffian* Pf of a skew-symmetric $N \times N$ matrix $A$ is the Bererzinian integral over anti-commuting variables $(\theta^i)_{i=1}^N$:

$$\text{Pf}(A) := \int \left(\prod_i \mathrm{d}\theta^i\right) \exp\left(\tfrac{1}{2} A_{ij} \theta^i \theta^j\right).$$
pick coefficient of top $\theta$-power

**Hidden super-geometry of quantum Hall systems.** This suggests to promote the plane $\mathbb{C}^1$ to the super-space $\mathbb{C}^{1|1}$ with its *super-translation group* structure

$$(z, \theta) + (z', \theta') = (z + z' + \theta\theta', \theta + \theta').$$

---

[3]For $N$ electrons in an effectively 2D material, and assumed to be completely spin-polarized by the transverse magnetic field, their wavefunction $\Psi$ is a skew-symmetric (by Pauli exclusion) $\mathbb{C}$-valued function of $N$ complex numbers $(z^i \in \mathbb{C})_{i=1}^N$. We omit normaliztion. For the Read-Moore state $N$ must (for $\text{Pf}(-)$ to be defined) be even (which is harmless since $N$ is a macroscopic number of electrons).

Here the **super-Laughlin state** exhibits the Read-Moore state as a super-partner to the Laughlin state (up to normalization) [65] [58, (13)]:

$$\Psi_{\mathrm{sLa}}\big((z^1,\theta^1),\cdots,(z^N,\theta^N)\big) := \prod_{i<j}\big(z^i-z^j-\theta^i\theta^j\big)^q \exp\big(-\tfrac{1}{\ell_B^2}\sum_i\big|z^i\big|^2\big).$$

**Collective excitations.** The Moore-Read state is known to have two *density-wave excitations* for wave-vectors $k \in \mathbb{C}$:

**(i)** The **magneto-roton state**

$$\Psi_{\mathrm{MR},k}(z^1,\cdots,z^N) := \sum_i \exp\big(-i\overline{k}\partial_{z^i}\big)\exp\big(-\tfrac{i}{2}\overline{k}z^i\big)\Psi_{\mathrm{MR}}(z^1,\cdots,z^N).$$

**(ii)** The **neutral fermion** state

$$\Psi_{\mathrm{NF},k} \quad \text{which originally did not have a closed expression.}$$

However, lifting the magneto-roton state to super-space, for super-wavevector $(k,\kappa)\in\mathbb{C}^{1,1}$

$$\Psi_{\mathrm{MR},(k,\kappa)}(z^1,\cdots,z^N):=\int\big(\prod_i d\theta^i\big)\sum_i\exp\big(-i\overline{k}\partial_{z^i}\big)\exp\big(-\tfrac{i}{2}\overline{k}z^i\big)\exp\big(-\tfrac{i}{2}\overline{\kappa}\theta^i\big)\Psi_{\mathrm{sLa}}\big((z^1,\theta^1),\cdots,(z^N,\theta^N)\big),$$

it reproduces the magneto-roton state for even $N$, and the neutral fermion mode when an $(N+1)$st electron is added [58]:

**Hidden super-symmetry in fractional quantum Hall systems.** This super-unification predicts hidden supersymmetry in fractional quantum Hall systems — which is indeed (numerically) observed [119] [91] (also [5, §5]).

This all suggests that an accurate model for fractional quantum Hall systems should in fact itself *originate on superspace*, and this is what we start with now.

# 2 Flux-quantization on M5-probes

The first task now is to understand the flux-quantization on M5-brane probes, according to [39] [41] [136]. We will not (need to) explain in full detail the (super-)geometry of probe branes nor of their (super-)gravity backgrounds (full discussion is in [53] [54]), but do offer the following broad dictionary, for orientation:[4]

---

[4]All brane concepts we consider are well-defined and all conclusions have proofs – at no point do we rely on informal string theory folklore beyond motivation.

**M5-Brane probes** (namely *sigma-model* branes, in contrast to *black branes*) are 5-dimensional objects propagating in a gravitational target space $X$ (the "bulk"), along trajectories that are modeled by (super-)immersions of their 6D (and $\mathcal{N} = (2,0)$) worldvolume (super-)manifolds $\Sigma$

$$\text{probe M5-brane} \atop \text{(super-)worldvolume} \quad \Sigma^{1,5|2\cdot\mathbf{8}_+} \xrightarrow[\text{trajectory} \atop \text{(super-)immersion}]{\phi_s} X^{1,10|\mathbf{32}} . \quad {\text{target/background} \atop \text{(super-)spacetime}} \tag{1}$$

Here the admissible ("on-shell", meaning: satisfying the appropriate equations of motion) immersions $\phi_s$ are controlled by the (super-)geometry of $X$ – namely the brane's trajectory is subject to the gravitational- and Lorentz-forces exerted by the field content of $X$ – but $X$ itself remains unaffected by the choice of $\phi_s$ – meaning that the (gravitational) *back-reaction* of the brane on its ambient spacetime is neglected; this is what makes the brane but a *probe* of the *background* $X$.

Thereby the probe brane $(\Sigma, \phi_s)$ plays a double role:

  (i) on the one hand it is like a (higher-dimensional) fundamental particle, an "observer" of the bulk $X$ in the sense of mathematical relativity,

  (ii) on the other hand it is itself a (super-)spacetime with its own (quantum) field content:

     Remarkably, the magic of super-geometry makes such purely super-geometric immersions $\phi_s$ (1) embody not just the naïve (temporal-)spatial worldvolume trajectory, but also a 3-flux density $H_3^s$ *on* $\Sigma$ [54, §3.3]. This is (on-shell) the notorious "self-dual" flux density whose accurate quantization (traditionally neglected) is our main concern here.

This second aspect is what we are concerned with for the purpose of modeling strongly-coupled quantum systems: The (1+3)D worldvolume $M^{1,3}$ of a quantum material – or, for the intent of modeling anyons, the effectively $(1+2)D$-worldvolume $M^{1,2}$ of a sheet-like material (e.g. an atomic mono-layer akin to graphene) – is to be identified with a sub-quotient of the brane worldvolume, typically with a fixed locus (orbifold singularity) inside the base of a fibration (Kaluza-Klein reduction).

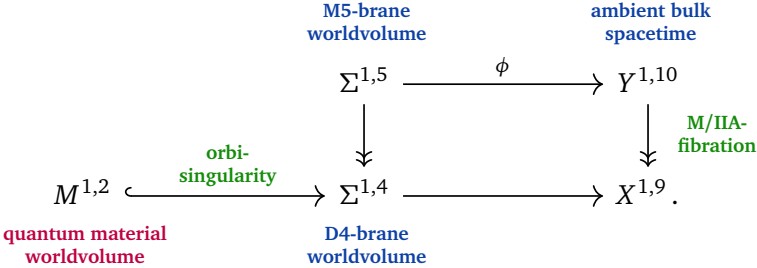

**Their flux quantization** (to recall from [132]) is then encoded in a choice of a fibration $\mathcal{A} \xrightarrow{p} \mathcal{B}$ of classifying spaces, subject to the constraint that the Bianchi identities for the (duality-symmetric) flux densities on bulk and brane are the closure/flatness condition on $\mathfrak{l}p$-valued differential forms, where $\mathfrak{l}(-)$ forms *Whitehead $L_\infty$-algebras* of these classifying fibrations (dual to their minimal relative Sullivan model).

    Given such a choice, the topological sector of the higher gauge fields on bulk and brane are given by maps from the brane-immersion into the classifying fibration:

    With these comments on perspective out of the way, **the plan of this section** are the following topics:

  (i) Bianchi identities on magnetized M5-probes.

(ii) Flux quantization in Twistorial Cohomotopy.

(iii) Aside: Projective Spaces and their Fibrations.

(iv) Orbi-worldvolumes and Equivariant charges.

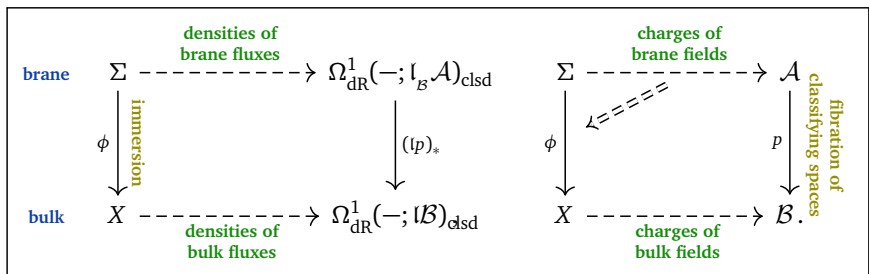

The first step of flux quantization is to identify the Bianchi identities satisfied by the flux densities:

**Bianchi identities on M5-Probes of 11D SuGra via super-geometry.** Consider the 11D super-tangent space

$$\underset{\text{super-Minkowski}}{\mathbb{R}^{1,10|\mathbf{32}}} \hookrightarrow \underset{\text{super-Poincaré}}{\mathfrak{isom}(\mathbb{R}^{1,10|\mathbf{32}})} \longrightarrow\!\!\!\!\!\! \underset{\text{Lorentz}}{\mathfrak{so}(1,10)},$$

with its super-invariant 1-forms (cf. [53, §2.1]):

$$\mathrm{CE}(\mathbb{R}^{1,10|\mathbf{32}}) \simeq \underset{\text{super-transl. invar. forms}}{\Omega^\bullet_{\mathrm{dR}}(\mathbb{R}^{1,10|\mathbf{32}})^{\mathrm{li}}} \simeq \mathbb{R}_{\mathrm{d}}\left[\begin{array}{c}(\Psi^\alpha)^{32}_{\alpha=1}\\(E^a)^{10}_{a=0}\end{array}\right] \Big/ \left(\begin{array}{l}\mathrm{d}\,\Psi^\alpha = 0\\ \mathrm{d}\,E^a = (\overline{\Psi}\,\Gamma^a\,\Psi)\end{array}\right).$$

Remarkably, the quartic Fierz identities entail that [25] [102] [53, Prop. 2.73]:

$$\left.\begin{array}{lll} G^0_4 &:=& \frac{1}{2}(\overline{\Psi}\,\Gamma_{a_1 a_2}\,\Psi)\,E^{a_1}E^{a_2}\\ G^0_7 &:=& \frac{1}{5!}(\overline{\Psi}\,\Gamma_{a_1\cdots a_5}\,\Psi)\,E^{a_1}\cdots E^{a_5}\end{array}\right\} \in \underset{\text{fully super-invariant forms}}{\mathrm{CE}(\mathbb{R}^{1,10|\mathbf{32}})^{\mathrm{Spin}(1,10)}}, \quad \text{satisfy:} \quad \begin{array}{l}\mathrm{d}\,G^0_4 = 0,\\ \mathrm{d}\,G^0_7 = \frac{1}{2}G^0_4\,G^0_4.\end{array}$$

To globalize this situation, say that an **11D super-spacetime** $X$ is a super-manifold equipped with a super-Cartan connection, locally on an open cover $\widetilde{X} \twoheadrightarrow X$ given by

$$\left.\begin{array}{l}(\Psi^\alpha)^{32}_{\alpha=1}\\(E^a)^{10}_{a=0}\\(\Omega^{ab} = -\Omega^{ba})^{10}_{a,b=0}\end{array}\right\} \in \Omega^1_{\mathrm{dR}}(\widetilde{X}), \qquad \begin{array}{l}\text{such that the}\\ \text{super-torsion}\\ \text{vanishes}\end{array} \qquad \mathrm{d}\,E^a - \Omega^a{}_b\,E^b = (\overline{\Psi}\,\Gamma^a\,\Psi),$$

and say that **C-field super-flux** on such a super-spacetime are super-forms with these co-frame components:

$$\boxed{\begin{array}{lllll} G^s_4 &:=& G_4 + G^0_4 &:=& \frac{1}{4!}(G_4)_{a_1\cdots a_4}E^{a_1}\cdots E^{a_4} + \frac{1}{2}(\overline{\Psi}\,\Gamma_{a_1 a_2}\,\Psi)E^{a_1}E^{a_2},\\ G^s_7 &:=& G_7 + G^0_7 &:=& \frac{1}{7!}(G_4)_{a_1\cdots a_7}E^{a_1}\cdots E^{a_7} + \frac{1}{5!}(\overline{\Psi}\,\Gamma_{a_1\cdots a_5}\,\Psi)E^{a_1}\cdots E^{a_5}.\end{array}}$$

**Theorem** [53, Thm. 3.1]: On an 11D super-spacetime $X$ with C-field super-flux $(G^s_4, G^s_7)$:

$$\begin{array}{l}\textit{The duality-symmetric}\\ \textit{super-Bianchi identity}\end{array} \left\{\begin{array}{ll}\mathrm{d}\,G^s_4 &= 0\\ \mathrm{d}\,G^s_7 &= \frac{1}{2}\,G^s_4\,G^s_4\end{array}\right\} \textit{is equivalent to} \quad \begin{array}{l}\textit{the full 11D SuGra}\\ \textit{equations of motion!}\end{array}$$

Next, on the super-subspace $\mathbb{R}^{1,5|2\cdot\mathbf{8}_+} \overset{\phi_0}{\hookrightarrow} \mathbb{R}^{1,10|\mathbf{32}}$ fixed by the involution $\Gamma_{012345} \in \mathrm{Pin}^+(1,10)$ we have:

$$H_3^0 \;:=\; 0 \;\in\; \mathrm{CE}\big(\mathbb{R}^{1,5|2\cdot\mathbf{8}_+}\big)^{\mathrm{Spin}(1,5)}, \quad \text{satisfies:} \quad \boxed{\mathrm{d}\,H_3^0 \;=\; \phi_0^* G_4^0.}$$

To globalize this situation, say that a super-immersion $\Sigma^{1,5|2\cdot\mathbf{8}_+} \overset{\phi_s}{\longrightarrow} X^{1,10|\mathbf{32}}$ is $^1/_2$**BPS M5** if it is "locally like" $\phi_0$, and say that **B-field super-flux** on such an M5-probe is a super-form with these co-frame components:

$$\boxed{H_3^s \;:=\; H_3 + H_3^0 \;:=\; \tfrac{1}{3!}(H_3)_{a_1 a_2 a_3} e^{a_1} e^{a_2} e^{a_3} + 0,} \qquad \big(e^{a<6} := \phi_s^* E^a\big),$$

where we are highlighting that with $H_3^0$ vanishing, by the above, the gravitino contribution to the superform vanishes.

**Theorem** [54, §3.3]: On a super-immersion $\phi_s$ with B-field super-flux $H_3^s$:

> *The super-Bianchi identity* $\big\{\mathrm{d}\,H_3^s \;=\; \phi_s^* G_4^s\big\}$, *is equivalent to*     *the M5's B-field equations of motion.*

In particular, the (non-linear self-)duality conditions on the ordinary fluxes are *implied*: $G_4 \longleftrightarrow G_7$ and $H_3 \longleftrightarrow H_3$.

Seeing from this that also trivial tangent super-cochains may have non-trivial globalization, observe next that:

$$F_2^0 \;:=\; \big(\overline{\psi}\,\psi\big) \;=\; 0 \;\in\; \mathrm{CE}\big(\mathbb{R}^{1,5|2\cdot\mathbf{8}_+}\big)^{\mathrm{Spin}(1,5)}, \quad \text{satisfies:} \quad \boxed{\mathrm{d}\,F_2^0 \;=\; 0.}$$

Globalizing this to $\Sigma^{1,5|2\cdot\mathbf{8}_+}$ via

$$\boxed{F_2^s \;:=\; F_2 + F_2^s \;:=\; \tfrac{1}{2}(F_2)_{a_1 a_2} e^{a_1} e^{a2} + 0,}$$

we have on top of the above:

**Theorem** [137, p 7]:

> *The super-Bianchi identity* $\big\{\mathrm{d}\,F_2^s \;=\; 0\big\}$ *is equivalent to*     *the Chern-Simons E.O.M.:* $F_2 = 0$.

**Flux quantization in Twistorial Cohomotopy.** In summary, a remarkable kind of higher super-Cartan geometry locally modeled on the 11D super-Minkowski spacetime $\mathbb{R}^{1,10|\mathbf{32}}$ entails that on-shell 11D supergravity probed by magnetized $^1/_2$BPS M5-branes implies and is entirely governed by these Bianchi identities on super-flux densities:

$$
\begin{aligned}
\text{A-field} \quad & \mathrm{d}\,F_2^s \;=\; 0, & \mathrm{d}\,G_4^s \;=\; 0, & \quad \text{C-field} \\[4pt]
\substack{\text{self-dual}\\ \text{B-field}} \quad & \mathrm{d}\,H_3^s \;=\; \phi_s^* G_4^s + \theta\,F_2^s F_2^s, & \mathrm{d}\,G_7^s \;=\; \tfrac{1}{2}G_4^s G_4^s, & \quad \substack{\text{dual}\\ \text{C-field}} \\[4pt]
\text{M5 probe} \quad & \Sigma^{1,5|2\cdot\mathbf{8}_+} \xrightarrow[\ ^1/_2\text{BPS immersion}\ ]{\phi_s} X^{1,10|\mathbf{32}}. & & \quad \text{SuGra bulk}
\end{aligned}
\tag{2}
$$

Here we have observed that the Green-Schwarz term $F_2^s F_2^s$ may equivalently be included for any theta-angle $\theta \in \mathbb{R}$ without affecting the equations of motion (since, recall, the CS e.o.m. $F_2^s = 0$ is already implied by $\mathrm{d}\,F_2^s = 0$).

However, non-vanishing theta-angle does affect the admissible flux-quantization laws and hence the global solitonic and torsion charges of the fields. The choice of flux quantization according to *Hypothesis H* [39] [41] is the following:

**Admissible fibrations of classifying spaces for cohomology theories** with the above character images (2). The homotopy quotient of $S^7$ is

(i) for $\theta = 0$ by the trivial action and

(ii) for $\theta \neq 0$ by the principal action of the complex Hopf fibration.

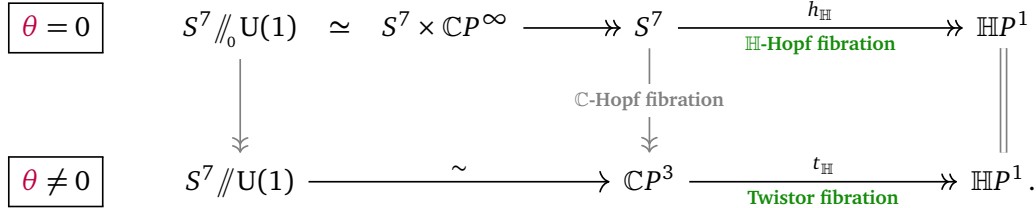

**Proof.** This may be seen as follows [41, Lem. 2.13]:

Since the real cohomology of projective space is a truncated polynomial algebra,

$$H^\bullet(\mathbb{C}P^n; \mathbb{R}) \simeq \mathbb{R}\big[\overbrace{c_1}^{\deg=2}\big]/(c_1^{n+1}), \quad H^\bullet(\overbrace{\mathbb{C}P^\infty}^{\simeq B\mathrm{U}(1)}; \mathbb{R}) \simeq \mathbb{R}[c_1],$$

$$H^\bullet(\mathbb{H}P^n; \mathbb{R}) \simeq \mathbb{R}\big[\underbrace{\tfrac{1}{2}p_1}_{\deg=4}\big]/(p_1^{n+1}), \quad H^\bullet(\underbrace{\mathbb{H}P^\infty}_{\substack{\simeq B\mathrm{Sp}(1) \simeq B\mathrm{SU}(2) \\ \simeq B\mathrm{Spin}(3)}}; \mathbb{R}) \simeq \mathbb{R}[\tfrac{1}{2}p_1],$$

the minimal dgc-algebra model for $\mathbb{C}P^n$ needs a closed generator $f_2$ to span the cohomology and a generator $h_{2n+1}$ in order to truncate it; analogously for $\mathbb{H}P^n$. Since these generators also form a graded linear basis for the rationalized homotopy groups of these spaces, they give the minimal Sullivan models (cf [132, Prop. 3.7]):

$$\mathrm{CE}\big(\mathfrak{l}\mathbb{C}P^n\big) \simeq \mathbb{R}_d\begin{bmatrix} f_2 \\ h_{2n+1} \end{bmatrix}\Big/\begin{pmatrix} \mathrm{d}\,f_2 &= 0 \\ \mathrm{d}\,h_{2n+1} &= (f_2)^{n+1} \end{pmatrix},$$

$$\mathrm{CE}\big(\mathfrak{l}\mathbb{H}P^n\big) \simeq \mathbb{R}_d\begin{bmatrix} g_4 \\ g_{4n+3} \end{bmatrix}\Big/\begin{pmatrix} \mathrm{d}\,g_4 &= 0 \\ \mathrm{d}\,g_{4n+3} &= (g_4)^{n+1} \end{pmatrix}.$$

Furthermore, since the second Chern class of a $\mathrm{U}(1) \simeq S\big(\mathrm{U}(1)^2\big) \subset \mathrm{SU}(2)$-bundle is minus the cup square of the first Chern class (by the Whitney sum rule), so that (cf. [126, (216)])

$$\begin{array}{ccccc}
\mathbb{C}P^3 & \hookrightarrow & \mathbb{C}P^\infty & \simeq & B\mathrm{U}(1) & & -(c_1)^2 \\
\downarrow{\scriptstyle t_{\mathbb{H}}} & & \downarrow & & {\scriptstyle |}\; B(c \mapsto \mathrm{diag}(c,c^*)) & & \uparrow \\
\mathbb{H}P^1 & \hookrightarrow & \mathbb{H}P^\infty & \simeq & B\mathrm{SU}(2) & & \tfrac{1}{2}p_1 = c_2\,,
\end{array}$$

the minimal model of $\mathbb{C}P^3$ *relative* to that of $\mathbb{H}P^1 \simeq S^4$ (cf. [42, Prop. 4.24]) needs to adjoin to the latter not only $f_2$ but also a generator $h_3$ imposing this relation in cohomology, whence it must be

$$\mathrm{CE}\big(\mathfrak{l}_{\mathbb{H}P^1}\mathbb{C}P^3\big) \simeq \mathbb{R}_d\begin{bmatrix} f_2 \\ h_3 \\ g_4 \\ g_7 \end{bmatrix}\Big/\begin{pmatrix} \mathrm{d}\,f_2 = 0 \\ \mathrm{d}\,h_3 = g_4 + f_2 f_2 \\ \mathrm{d}\,g_4 = 0 \\ \mathrm{d}\,g_7 = \tfrac{1}{2}g_4 g_4 \end{pmatrix},$$

which is clearly quasi-isomorphic to $\mathrm{CE}(\mathfrak{l}\mathbb{C}P^3)$. $\qquad\square$

The resulting fibration of $L_\infty$-algebras is manifestly just that classifying the desired Bianchi identities (2)

(we are showing the case $\theta \neq 0$, which by isomorphic rescaling may be taken to be $\theta = 1$):

$$
\begin{array}{ccc}
\Sigma^6 \dashrightarrow \Omega^1_{\mathrm{dR}}\big(-; \mathfrak{l}_{\mathbb{H}P^1}\mathbb{C}P^3\big)_{\mathrm{clsd}} & \Omega^\bullet_{\mathrm{dR}}(\Sigma^6) \longleftarrow \mathrm{CE}\big(\mathfrak{l}_{\mathbb{H}P^1}\mathbb{C}P^3\big) \\
\phi \Big\downarrow \qquad \Big\downarrow (\mathfrak{l}t_{\mathbb{H}})_* & \phi^* \Big\uparrow \qquad \Big\downarrow (\mathfrak{l}t_{\mathbb{H}})^* \\
\Sigma^{11} \dashrightarrow \Omega^1_{\mathrm{dR}}\big(-; \mathfrak{l}\mathbb{H}P^1\big)_{\mathrm{clsd}} & \Omega^\bullet_{\mathrm{dR}}(X^{11}) \longleftarrow \mathrm{CE}(\mathfrak{l}\mathbb{H}P^1)
\end{array}
$$

$$
\Leftrightarrow
$$

$$
\begin{array}{c|c}
F_2 & \mathrm{d}\,F_2 = 0, \\
H_3 \in \Omega^\bullet_{\mathrm{dR}}(\Sigma^6) & \mathrm{d}\,H_3 = G_4 + F_2 F_2, \\
\hline
G_4 & \mathrm{d}\,G_4 = 0, \\
G_7 \in \Omega^\bullet_{\mathrm{dR}}(X^{11}) & \mathrm{d}\,G_7 = \tfrac{1}{2}G_4 G_4.
\end{array}
$$

**Aside: Projective Spaces and their Fibrations** – Herse we used the following classical facts. Consider:

> division algebras $\mathbb{R} \hookrightarrow \mathbb{C} \hookrightarrow \mathbb{H}$, generically denoted $\mathbb{K} \in \{\mathbb{R}, \mathbb{C}, \mathbb{H}\}$
>
> groups of units $\quad \mathbb{K}^\times := \mathbb{K} \setminus \{0\}$, understood with the multiplicative group structure
>
> projective spaces $\mathbb{K}P^n := \big(\mathbb{K}^{n+1} \setminus \{0\}\big)/\mathbb{K}^\times$
>
> higher spheres $\quad S^n \simeq \big(\mathbb{R}^{n+1} \setminus \{0\}\big)/\mathbb{R}_{>0}$
>
> $\mathbb{K}$-Hopf fibrations are the quotient co-projections induced by $\iota : \mathbb{R}_{>0} \hookrightarrow \mathbb{K}$

The classical Hopf fibrations $h_\mathbb{K}$ are:

$$
\begin{array}{ccc}
S^0 \simeq \mathbb{R}^\times/\mathbb{R}_{>0} & S^1 \simeq \mathbb{C}^\times/\mathbb{R}_{>0} & S^3 \simeq \mathbb{H}^\times/\mathbb{R}_{>0} \\
\Big\downarrow\mathrm{ker} & \Big\downarrow\mathrm{ker} & \Big\downarrow\mathrm{ker} \\
S^1 \simeq \big(\mathbb{R}^2\setminus\{0\}\big)/\mathbb{R}_{>0} & S^3 \simeq \big(\mathbb{C}^2\setminus\{0\}\big)/\mathbb{R}_{>0} & S^7 \simeq \big(\mathbb{H}^2\setminus\{0\}\big)/\mathbb{R}_{>0} \\
\Big\downarrow h_\mathbb{R} \quad \Big\downarrow \iota_* & \Big\downarrow h_\mathbb{C} \quad \Big\downarrow \iota_* & \Big\downarrow h_\mathbb{H} \quad \Big\downarrow \iota_* \\
S^1 \simeq \underbrace{\big(\mathbb{R}^2\setminus\{0\}\big)/\mathbb{R}^\times}_{\mathbb{R}P^1}, & S^2 \simeq \underbrace{\big(\mathbb{C}^2\setminus\{0\}\big)/\mathbb{C}^\times}_{\mathbb{C}P^1}, & S^4 \simeq \underbrace{\big(\mathbb{H}^2\setminus\{0\}\big)/\mathbb{H}^\times}_{\mathbb{H}P^1}.
\end{array}
$$

The Hopf fibrations in higher dimensions are the attaching maps exhibiting the topological cell-complex structure of projective spaces [106], from which the (cellular) cohomology follows readily.

$$
\begin{array}{ccc}
S\big(\mathbb{K}^{n+1}\big) & \longrightarrow & * \\
h_\mathbb{K} \Big\downarrow \nearrow{}_{\mathrm{(po)}} & & \Big\downarrow \\
\mathbb{K}P^n & \hookrightarrow & \mathbb{K}P^{n+1}.
\end{array}
$$

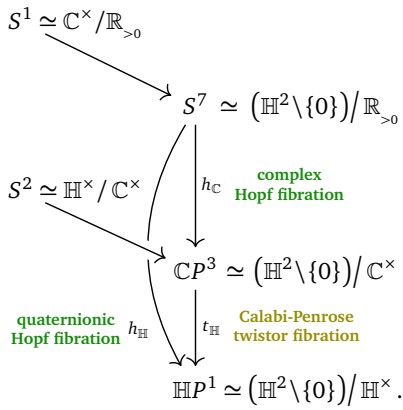

Further factor-fibrations arise by factoring the Hopf fibrations via the stage-wise quotienting along

$$\mathbb{R}_{>0} \hookrightarrow \mathbb{R} \hookrightarrow \mathbb{C} \hookrightarrow \mathbb{H}.$$

Notably, the classical quaternionic Hopf fibration $h_{\mathbb{H}}$ factors through a higher-dimensional complex Hopf fibration followed by the **Calabi-Penrose twistor fibration** $t_{\mathbb{H}}$ [41, §2].

Equivariantization: Since the quotienting is by right actions, these fibrations are equivariant under the left action of

$$\mathrm{Spin}(5) \simeq \mathrm{Sp}(2) := \left\{ g \in \mathrm{GL}_2(\mathbb{H}) \,\middle|\, g^\dagger \cdot g = \mathrm{e} \right\}.$$

For example, the involution $\sigma := \begin{bmatrix} 0 & 1 \\ 1 & 0 \end{bmatrix} \in \mathrm{Sp}(2)$ swaps the two copies of $\mathbb{H}$:

$$\begin{array}{ccc}
\mathbb{C}P^3 & \xrightarrow{\ t_{\mathbb{H}}\ } & \mathbb{H}P^1 \\
\left(\mathbb{H}\times\mathbb{H}\setminus\{0\}\right)/\mathbb{C}^\times \to \left(\mathbb{H}\times\mathbb{H}\setminus\{0\}\right)/\mathbb{H}^\times \\
\sigma \quad\quad \left(\mathbb{H}\oplus\mathbb{H}\setminus\{0\}\right)/\mathbb{C}^\times \to \left(\mathbb{H}\oplus\mathbb{H}\setminus\{0\}\right)/\mathbb{H}^\times \quad \sigma \\
\mathbb{C}P^3 & \xrightarrow{\ t_{\mathbb{H}}\ } & \mathbb{H}P^1.
\end{array}$$

The resulting $\mathbb{Z}_2$-fixed locus is the 2-sphere:

$$\begin{array}{ccccc}
\left(\mathbb{C}P^3\right)^{\mathbb{Z}_2} & \simeq & \left(\mathbb{H}\setminus\{0\}\right)/\mathbb{C}^\times & \simeq & S^2 \\
\downarrow{\scriptstyle(t_{\mathbb{H}})^{\mathbb{Z}_2}} & & \downarrow & & \downarrow \\
\left(\mathbb{H}P^1\right)^{\mathbb{Z}_2} & \simeq & \left(\mathbb{H}\setminus\{0\}\right)/\mathbb{H}^\times & \simeq & *.
\end{array}$$

This is the 2-sphere coefficient that will end up being responsible for stabilizing anyons on orbi-worldvolumes!

We next discuss how this comes about.

**Aside: Implications of Hypothesis H**, in view of traditional expectations for M-theory.

**The plain Hypothesis H** for the bulk theory says that the non-perturbative completion of the C-field in 11d supergravity is a cocycle in *differential Cohomotopy* $\widehat{\pi}^4$ [36, §4] [56, §3.1] [42, Ex. 9.3] and as such involves (exposition in [132, §3.3]) a map $\chi$ from spacetime to the homotopy type of the 4-sphere, with the C-field gauge potentials $(\widehat{C}_3, \widehat{C}_6)$ exhibiting the flux densities $(G_4, G_7)$ as $\mathbb{R}$-rational representatives of $\chi$.

As an immediate plausibility check, from the well-known homotopy groups of spheres in low degrees this implies (cf. [69, (22-3)] [126, (22)]):

- Integral quantization of charges carried by singular M5-brane branes (cf. the following (5)):
$$\pi^4\big(\mathbb{R}^{10,1} \setminus \mathbb{R}^{5,1}\big) = \pi^4\big(\mathbb{R}^{5,1} \times \mathbb{R}_+ \times S^4\big) = \pi^4(S^4) = \pi_4(S^4) = \mathbb{Z}. \tag{3}$$

- Integral quantization of charges carried by singular M2-branes... plus a torsion-contribution (a first prediction of Hypothesis H: fractional M2-branes):
$$\pi^4\big(\mathbb{R}^{10,1} \setminus \mathbb{R}^{2,1}\big) = \pi^4\big(\mathbb{R}^{2,1} \times \mathbb{R}_+ \times S^7\big) = \pi^4(S^7) = \pi_7(S^4) = \mathbb{Z} \oplus \mathbb{Z}_{12}. \tag{4}$$

**On the nature of the spheres.** In itself, the 4-sphere $S^4$ appearing in Hypothesis H is a *classifying space*, hence an abstract tool of algebraic topology, not a physical space. On the other hand, once it is as such used for flux quantization of the C-field of 11D supergravity, then every such field configuration relates the two.

In particular, when physical spacetime just so happens to itself be a product of a contractible space with a physical 4-sphere — notably for spacetimes near M5-branes as in (3) — then there arises an effective identification of the physical spatial sphere with the classifying space (cf. [69, p. 17] [126, (22)] [125]):

$$
\begin{array}{c}
\text{map classifying brane charge} \\
\mathbb{R}^{1,5} \times \mathbb{R}_+^1 \times S^4 \xrightarrow[\sim]{\mathrm{pr}_2} S^4 \dashrightarrow S^4 ,
\end{array} \tag{5}
$$

near-horizon spacetime of black M5-brane (Poincaré patch of AdS$_7 \times S^4$ ) — homotopy equivalent to — spatial 4-sphere — effective classifying map — classifying 4-sphere

in that for a single M5-brane of *unit* charge the effective classifying map (on the right) is in the homotopy class of the *identity map* from the spatial to the classifying 4-sphere. (A vaguely reminiscent kind of identification may also be recognized in [71, p. 5-6] [48, p. 5-6], there thought of as mediated by the scalar fields on the M5-brane.)

Directly analogous comments apply to the role of $S^7$ and the near horizon spacetime geometry of M2-branes, cf. (4).

**Hypothesis H with curvature corrections.** More generally, curvature corrections from the coupling to the background gravity are postulated to be reflected in *tangentially twisted* 4-Cohomotopy [39], analogous to the well-known twisting of the RR-field flux-quantization in K-theory by its background B-field:

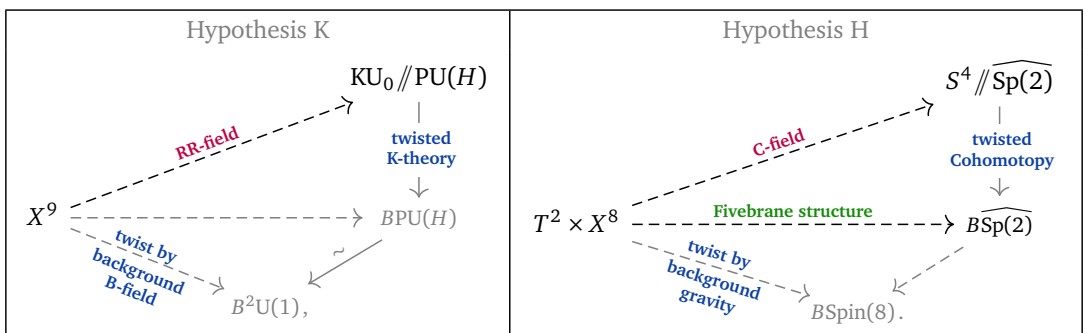

To distinguish M2/M5-charge, the tangential twisting needs to preserve the $\mathbb{H}$-Hopf fibration $\Rightarrow$ tangential Sp(2) $\hookrightarrow$ Spin(8)-structure [39, §2.3]. With this, integrality of M2's Page charge & anomaly-cancellation of the M5's Hopf-WZ term follows from trivialization of the Euler 8-class, which means lift to the *Fivebrane* 6-group $\widehat{\mathrm{Sp}(2)} \to \mathrm{Sp}(2)$ [38, §4].

This implies [39, Prop. 3.13] [38, Thm. 4.8]:

**(i)** half-integrally shifted quantization of M5-brane charge in curved backgrounds,

$$[\widetilde{G}_4] := \underbrace{[G_4]}_{\substack{\text{C-field}\\\text{4-flux}}} + \tfrac{1}{2}\big(\underbrace{\tfrac{1}{2}p_1(TX^8)}_{\substack{\text{integral Spin-}\\\text{Pontrjagin class}}}\big) \in H^4\big(X^8; \mathbb{Z}\big), \tag{6}$$

**(ii)** integral quantization of the Page charge of M2-branes:[5]

$$2[\widetilde{G}_7] := 2\big([G_7] + \tfrac{1}{2}[H_3 \wedge \widetilde{G}_4]\big) \in H^7(\widehat{X}^8; \mathbb{Z}). \tag{7}$$

Both of these quantization conditions on M-brane charge are thought to be crucial for M-theory to make any sense. But, previously, item (i) had remained enigmatic and item (ii) had remained wide open.

But there is more:

| Provable implications from Hypothesis H of subtle effects expected in M-theory: | – half-integral shift of 4-flux | [39, Prop. 3.13] |
| --- | --- | --- |
| | – DMW anomaly cancellation | [39, Prop. 3.7] |
| | – the C-field's "integral EoM" | [39, §3.6] |
| | – M2 Page charge quantization | [38, Thm. 4.8] |
| | – integrality of $\tfrac{1}{6}(G_4)^3$ | [56, Rem. 2.9] |
| | – M5-brane anomaly cancellation | [125] |
| | – non-Abelian gerbe field on M5 | [40] |

It is these and further results that suggest that Hypothesis H goes towards the correct flux-quantization law for the C-field in M-theory.

Yet more generally, Hypothesis H applies to orbifold spacetimes, where it postulates flux quantization in (twisted and) *equivariant* Cohomotopy [123] [14]. This is what we turn to next.

**Orbi-worldvolumes and Equivariant charges.** Flux-quantization generalizes to *orbifolds*[6] by generalizing the cohomology of the charges to *equivariant cohomology* [124]. In terms of classifying spaces this simply means that all spaces are now equipped with the action of a finite group $G$ and all maps are required to be $G$-equivariant.

We take $G := \mathbb{Z}_2$ and the classifying fibration to be the **twistor fibration** $p := t_{\mathbb{H}}$ equivariant under swapping the $\mathbb{H}$-summands,

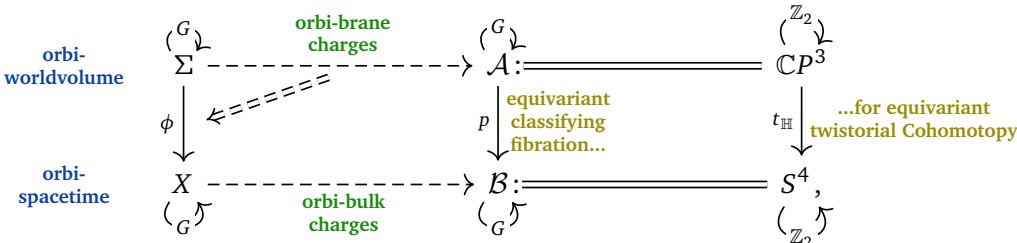

and the brane/bulk orbifold we take to be as on :

---

[5]On the right of (7), the "hat" in $\widehat{X}^8$ indicates that this holds locally, namely on suitable fibration over spacetime (cf. [39, (116)] [40, p. 6]) on which the "M-theory 3-form" $H_3$ is globally defined, which it cannot be on $X^8$ unless $\widetilde{G}_4$ is cohomologically trivial (a basic subtlety that has traditionally been glossed over in discussions of Page charge.) But in the present context of flux quantization on M5-branes the analog of this extended spacetime is in fact the worldvolume of the M5-brane, restricted to which $\widetilde{G}_4$ does trivialize in cohomology, whence we need not further dwell here on the definition of $\widehat{X}^8$.

[6]For brevity we consider here only "very good" orbifolds, namely global quotients of manifolds by the action of a finite group $G$. This is sufficient for the present purpose and anyways the case understood by default in the string theory literature.

**The orbi-brane diagram** for a flat M5-brane wrapped on a trivial Seifert-fibered orbi-singularity. Shaded is the $\mathbb{Z}_2$-fixed locus/orbi-singularity.

We are adjoining the *point at infinity* to the space $\mathbb{R}^2_{\cup\{\infty\}} \underset{\text{homeo}}{\simeq} S^2$ which is thereby designated as transverse to any worldvolume solitons to be measured in reduced cohomology.

$$
\begin{array}{c}
\overset{\mathbb{Z}_2}{\curvearrowright}\Sigma := \mathbb{R}^{1,0} \times \mathbb{R}^2_{\cup\{\infty\}} \times S^1 \times \overset{(\overset{\mathbb{Z}_2}{\curvearrowright}}{\mathbb{R}^2_{\text{sgn}}} \\
\phi \downarrow \qquad\qquad \\
\underset{(\mathbb{Z}_2}{\curvearrowright}X := \mathbb{R}^{1,0} \times \mathbb{R}^2_{\cup\{\infty\}} \times S^1 \times \overset{(\overset{\mathbb{Z}_2}{\curvearrowright}}{\mathbb{R}^2_{\text{sgn}}} \times \mathbb{R}^5 .
\end{array}
$$

time | trnsvrs space to solitons | M/IIA-circle | orbi-cone | trnsvrs space to M5-brane

But since the cone $\mathbb{Z}_2 \subsetneq \mathbb{R}^2_{\text{sgn}}$ is equivariantly contractible, the inclusion of the $\mathbb{Z}_2$-fixed loci is actually a homotopy equivalence

$$
\overset{\mathbb{Z}_2}{\curvearrowright} * \xrightarrow[\text{hmtp}]{\sim} \overset{\mathbb{Z}_2}{\curvearrowright} * \quad\Rightarrow\quad
\begin{array}{ccc}
\Sigma^{\mathbb{Z}_2} & \xrightarrow[\text{hmtp}]{\sim} & \overset{\mathbb{Z}_2}{\curvearrowright}\Sigma \\
\phi^{\mathbb{Z}_2}\downarrow & & \downarrow\phi \\
X^{\mathbb{Z}_2} & \xrightarrow[\text{hmtp}]{\sim} & \underset{(\;G}{\curvearrowright}X
\end{array} .
$$

Therefore, our equivariant classifying maps are determined up to equivariant homotopy by their restriction to the fixed-locus and hence the charges are *localized on the orbi-singularity* where they take values in 2-Cohomotopy:

$$
\left\{
\begin{array}{ccc}
\overset{\mathbb{Z}_2}{\curvearrowright}\Sigma & \dashrightarrow & \overset{(\overset{\mathbb{Z}_2}{\curvearrowright}}{\mathbb{C}P^3} \\
\phi\downarrow & \Leftarrow\!=\!= & \downarrow t_{\mathbb{H}} \\
\underset{(\mathbb{Z}_2}{\curvearrowright}X & \dashrightarrow & \underset{(\mathbb{Z}_2}{\curvearrowright}S^4 \\
& \text{charges on orbifold} &
\end{array}
\right\}
\simeq
\left\{
\begin{array}{ccc}
\Sigma^{\mathbb{Z}_2} & \dashrightarrow & (\mathbb{C}P^3)^{\mathbb{Z}_2} = S^2 \\
\phi^{\mathbb{Z}_2}\downarrow & \Leftarrow\!=\!= & \downarrow t_{\mathbb{H}}^{\mathbb{Z}_2} \qquad \downarrow \\
X^{\mathbb{Z}_2} & \dashrightarrow & (S^4)^{\mathbb{Z}_2} = * \\
& \text{charges localized on orbi-singularity} &
\end{array}
\right\}
\simeq
\left\{
\mathbb{R}^2_{\cup\{\infty\}} \times S^1 \dashrightarrow S^2
\right\} .
$$

charges in 2-Cohomotopy of B-field solitons on M5 orbi-singularity

**Moduli space of worldvolume solitons.** To be precise, the solitonic charges are to be measured in the *reduced* 2-Cohomotopy classified by *pointed maps*, enforcing the condition that solitonic fields *vanish at infinity* [132, §2.2]. In the strongly-coupled situation, where the M/IIA circle de-compactifies to $\mathbb{R}^1$, the vanishing-at-infinity must also be applied here, whence (cf. [133, §A.2]) the moduli space of topological solitons is the loop space of the reduced 2-Cohomotopy moduli of the transverse space:

Moduli space of solitons on M5 orbi-singularity

$$
\mathrm{Maps}^*\!\big(\mathbb{R}^2_{\cup\{\infty\}} \wedge S^1, S^2\big) \quad\simeq\quad \Omega\,\mathrm{Maps}^*\!\big(\mathbb{R}^2_{\cup\{\infty\}}, S^2\big)
$$

Loop space of moduli space of solitons on D4 orbi-singularity

$$
\mathrm{Maps}^*\!\big(\mathbb{R}^2_{\cup\{\infty\}}, \Omega S^2\big).
$$

(The algebraic topology of maps to $\Omega S^2$ have also found some attention in [99] [100].)

**Outlook.** Strikingly, as we explain next, this moduli space is equivalently a space of *worldsheets of strings* in $\mathbb{R}^3$ with unit *charged endpoints* forming oriented *framed links*! [134]

Such link diagrams are just the envisioned topological quantum circuit protocols, and their framing regularizes the anyonic phase observables ("Wilson loop observables").

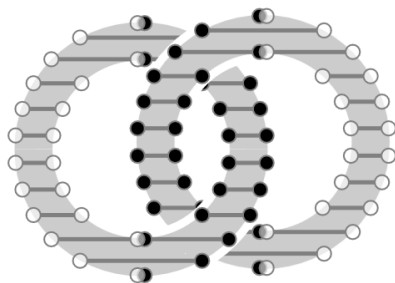

Figure 6: **Framed links as stringy worldsheets** are revealed by careful analysis as being the loops in the moduli space of solitonic cohomotopy charges of the plane.

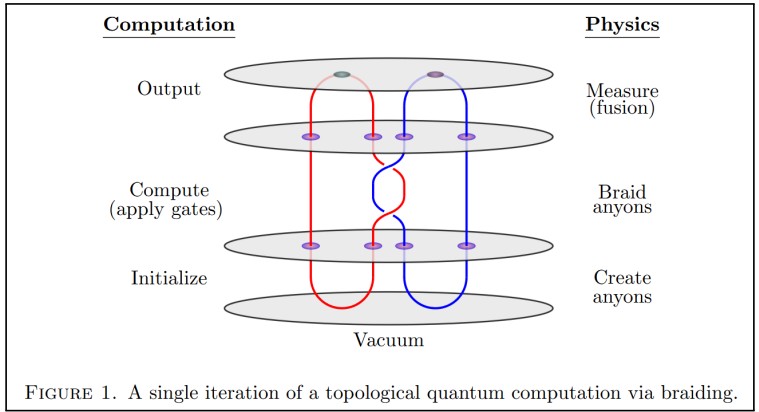

FIGURE 1. A single iteration of a topological quantum computation via braiding.

Figure 7: **Traditional protocol for topological quantum computation with anyons,** taken from Rowell ( [122], following [121, Fig. 2]).

# 3  Cohomotopy charge of solitons

Remarkably, there is an equivalence between *Cohomotopy* of spacetime/worldvolumes and *Cobordism* classes of submanifolds behaving like solitonic branes carrying the corresponding Cohomotopy charge [126, §2.2] [123, §2.1]:

The **Pontrjagin theorem** [114] [85, §IX] identifies the unstable $n$-Cohomotopy of a closed manifold with the cobordism classes of its normally framed submanifolds of co-dimension $n$.

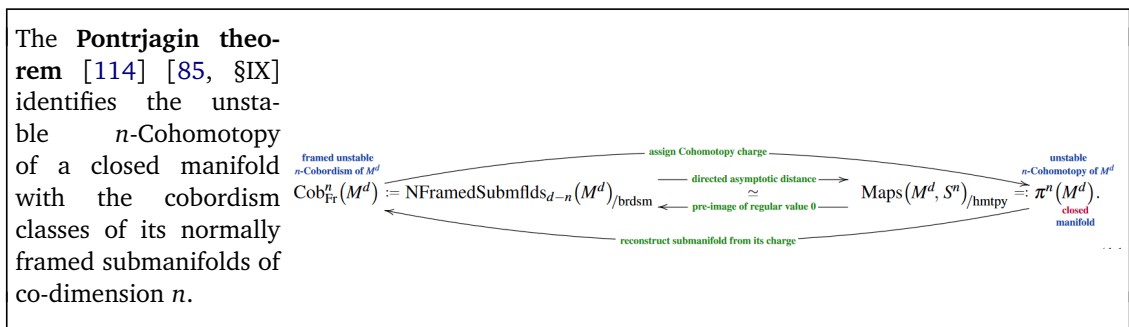

The **Cohomotopy charge** of a normally framed submanifold (aka *scanning map* or *Pontrjagin-Thom collapse*) is represented by mapping points of the ambient space to their directed distance if inside a tubular neighborhood, else to $\infty$.

Conversely, every Cohomotopy class is represented by a smooth map with 0 a regular value, whose pre-image is a normally framed submanifold with that Cohomotopy charge.

$$\Sigma^{d-n} \hookrightarrow M^d \xrightarrow{\;\;c\;\;} \mathbb{R}^n_{\mathrm{cpt}} = S^n$$

closed submanifold, normally framed — manifold — directed asymptotic distance from $\Sigma$ $\simeq$ cocycle representing Cohomotopy charge of $\Sigma$ — Cohomotopy classifying space ($n$-sphere)

constant on 0 at $\Sigma$ — directed distance near $\Sigma$ — regular value — constant on $\infty$ far away from $\Sigma$

Under this relation, homotopy of charge maps corresponds to nrml. framed **cobordism** of submnflds.

The cobordism relation exhibits a form of pair creation/annihilation of submanifolds carrying opposite Cohomotopy charges.

brane — space — anti-brane — normal framing in space — opposite normal framing — pair creation / annihilation — normal framing in spacetime — spacetime

framing charge — $f$ — $w$ — $\overline{w}$ — creation / annihilation — $f$ — branes — anti-brane

When making more ambient dimensions available, the cobordism classes eventually (quickly) exhibit **stabilization** on Abelian cobordism cohomology groups. (This might relate *Hypothesis H* to Vafa's *cobordism conjecture* cf. [126, §4]).

make more ambient spatial dimensions available to bordisms $\longrightarrow$

$$\Sigma^{d-n} \subset M^d \xhookrightarrow{(\mathrm{id},0)} M^d \times \mathbb{R}^1 \xhookrightarrow{(\mathrm{id},0)} M^d \times \mathbb{R}^2 \hookrightarrow \cdots\cdots\cdots\!\!\!> M^d \times \mathbb{R}^\infty$$

unstable framed Cobordism
$$\mathrm{Cob}^n_{\mathrm{Fr}}(M^d) \xrightarrow{\sigma} \mathrm{Cob}^{n+1}_{\mathrm{Fr}}(M^d \times \mathbb{R}^1) \xrightarrow{\sigma} \mathrm{Cob}^{n+2}_{\mathrm{Fr}}(M^d \times \mathbb{R}^2) \cdots\!\!\!> \widetilde{\mathrm{MFr}}^n(M^d_{\mathrm{cpt}})$$ stable framed Cobordism

Cohomotopy charge map — $\simeq$ — $\simeq$ — stable Pontrjagin-Thom isom.

unstable Cohomotopy
$$\widetilde{\pi}^n(M^d_{\mathrm{cpt}}) \xrightarrow{(-)\wedge S^1} \widetilde{\pi}^{n+1}\big((M^d \times \mathbb{R}^1)_{\mathrm{cpt}}\big) \xrightarrow{(-)\wedge S^1} \widetilde{\pi}^{n+1}\big((M^d \times \mathbb{R}^2)_{\mathrm{cpt}}\big) \cdots\!\!\!> \widetilde{\mathbb{S}}^n(M^d_{\mathrm{cpt}})$$ stable Cohomotopy

suspension homomorphism $\longrightarrow$

This "linearized" Cohomotopy/Cobordism is a **form of K-theory**: algebraic K-theory over the "absolute base field $\mathbb{F}_1$" (cf. [19, Thm. 5.9] [9, Cor. 2.25]).

non-Abelian Cohomotopy
$$\pi^\bullet \xrightarrow[\text{(i.e.: stabilize)}]{\text{linearize}} \mathbb{S}^\bullet \underset{\text{Pontrjagin \& Thom}}{=\!=\!=\!=} \mathrm{MFr}^\bullet$$
stable Cohomotopy — stable framed Cobordism

Barratt-Priddy & Quillen

$K\mathbb{F}_1^\bullet$

algebraic K-theory of "field with one element"

Thus flux quantization in Cohomotopy lifts to M-theory the same arguments that motivated topological K-theory in type II string theory: its character map reproduces the Bianchi identities & its equivalence relation models (anti-)brane pair-creation/annihilation.

**Moduli space of soliton configurations.** But the Pontrjagin theorem concerns only the total cohomotopical charge, identifying it with the *net* (anti-)brane content. Beyond that we have

the whole *moduli space* of charges (considered now specialized to our 2D transverse space), and **Segal's theorem** [143] says that the cohomotopy charge map (scanning map) identifies this with a moduli space of brane positions, namely with the *group-completed configuration space of points* [21] [157] [79]:

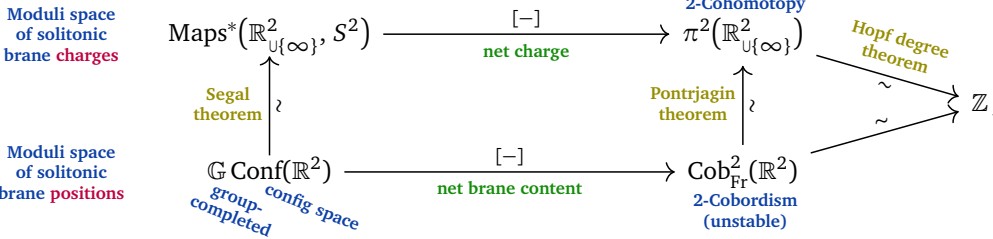

where the *configuration space of points* is the space of finite subsets of $\mathbb{R}^2$ – here understood as the space of positions of cores of solitons of unit charge $+1$,

$$\mathrm{Conf}(\mathbb{R}^2) = \left\{ \ \ \right\},$$

and its *group completion* $\mathbb{G}(-)$ is the topological completion of the topological partial monoid structure given by disjoint union of soliton configurations.

Naïvely this is given by also including **anti-solitons** in the form of configurations of $\pm$-*charged points*, topologized such as to allow for their pair annihilation/creation as shown in the left column on the right.

Remarkably, closer analysis reveals [107] that the group completion $\mathbb{G}(-)$ produces configurations of **strings** (extending parallel to one axis in $\mathbb{R}^3$) **with charged endpoints** whose pair annihilation/creation is smeared-out to string worldsheets as shown in the right column ([134, Fig. 2]):

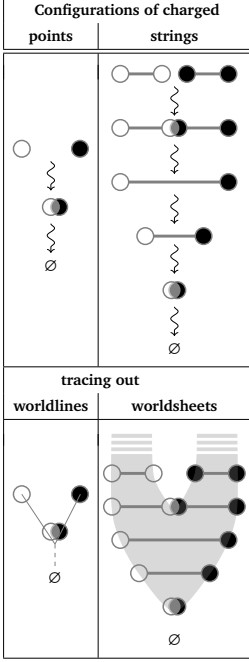

Figure 8: **The continuous relations in configuration spaces of charged points and strings** exhibit the pair annihiliation/creation of oppositely charged (end-)points.

This means (cf. [134, Prop. 3.14]) that the **vacuum-to-vacuum soliton scattering processes**, forming the loop space $\Omega\mathbb{G}\,\mathrm{Conf}(\mathbb{R}^2)$, are identified with *framed links* ( [108, p 15]); for instance:

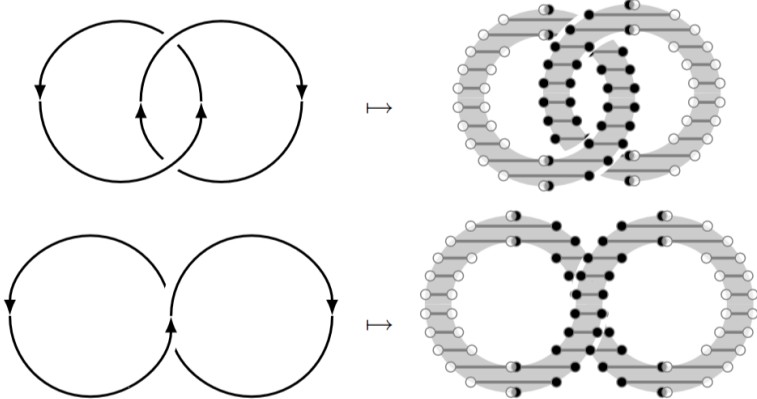

Figure 9: **Loops in the configuration space of charged strings** in the plane may be identified with (diagrams for) framed oriented links.

subject to *link cobordism* (cf. [93]):

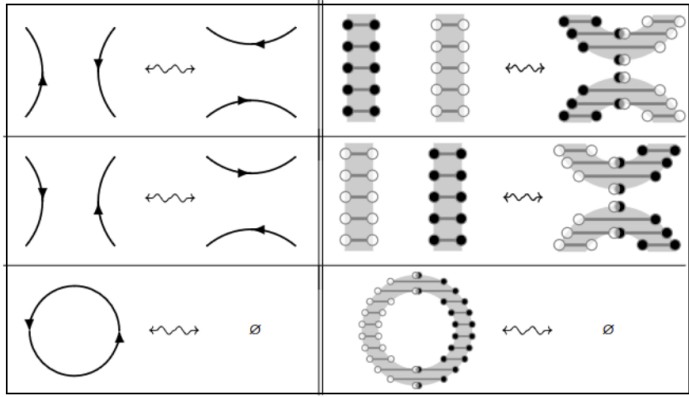

Figure 10: **Link cobordism from deformations of charged string moduli.** Shown on the right are evident continuous deformations of paths in the above configuration space of charged strings, hence in the group-completed configuration space of points. Shown on the left are the local deformations of corresponding framed links, generating the relation of *link cobordism*.

It follows [134, Thm 3.17] that the charge of a soliton scattering process $L$ is the sum over crossings of the *crossing number*

$$\#\left(\;\times\;\right) = +1, \quad \#\left(\;\times\;\right) = -1,$$

which equals the linking+framing number:

$$\Omega\mathbb{G}\mathrm{Conf}(\mathbb{R}^2) \xrightarrow{\;\sim\;} \Omega\mathrm{Maps}^*\!\left(\mathbb{R}^2_{\cup\{\infty\}}S^2\right) \xrightarrow{\;[-]\;} \pi_3(S^2) \simeq \mathbb{Z},$$

$$L \longmapsto \underset{\substack{\text{total crossing number} = \\ \text{linking + framing number}}}{\xrightarrow{\hspace{4cm}}} \#L.$$

But this is precisely the **Wilson loop observable** of $L$ in (Abelian) **Chern-Simons theory**! [134, §4]. This is what we explain next.

**The $k$-Soliton sector.** More generally, we may consider loops based in the $k$th connected component of the moduli space (cf. [134, Rem. 3.20]), corresponding to scattering process from $k$ to $k$ net number of solitons.

$$
\begin{array}{ccc}
\overset{\text{net charge } k}{\mathbb{G}\mathrm{Conf}_k(\mathbb{R}^2)} & \overset{\sim}{\longrightarrow} & \overset{\text{Hopf degree } k}{\mathrm{Maps}_k^*\big(\mathbb{R}^2_{\cup\{\infty\}}, S^2\big)} \\
\downarrow & & \downarrow \\
\mathbb{G}\mathrm{Conf}(\mathbb{R}^2) & \overset{\sim}{\longrightarrow} & \mathrm{Maps}^*\big(\mathbb{R}^2_{\cup\{\infty\}}, S^2\big).
\end{array}
$$

Since the double loop space $\mathrm{Maps}^*\big(\mathbb{R}^2_{\cup\{\infty\}}, S^2\big)$ admits the structure of a topological group, all these connected components have the same homotopy type, and hence these scattering processes $L$ are again classified by the integer total crossing number $\#L$ which is the Abelian Chern-Simons Wilson-loop observable.

$$
\begin{array}{c}
\Omega_k\,\mathbb{G}\mathrm{Conf}(\mathbb{R}^2) \\
\Downarrow \\
\pi_0\Omega_k\,\mathbb{G}\mathrm{Conf}(\mathbb{R}^2) \xrightarrow[\sim]{\qquad\qquad} \mathbb{Z}\,.
\end{array}
\quad \overset{L\,\mapsto\,\#L}{\searrow}
$$

For instance, a generic $k = 3$ process looks like this:

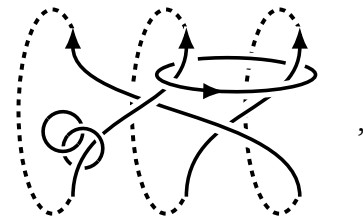

,

and via the framed cobordism moves

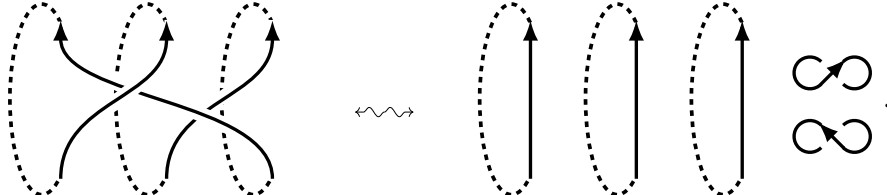

,

it computes to the trivial scattering process accompanied by $\#L$ vacuum pair braiding processes:

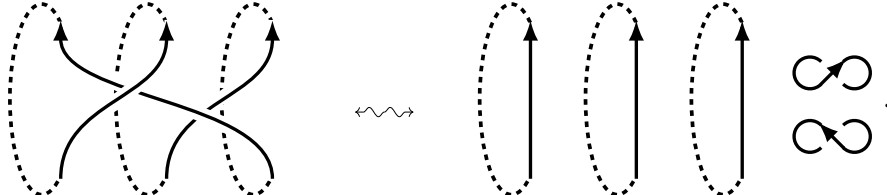

.

**Chern-Simons level.** We will see below further meanings of the number *lattice*:

| This integer $K$ is equivalently | the *number* of fractional quasi-hole vortices in a quantum Hall system, |
|---|---|
| | twice the *level* of their effective Abelian Chern-Simons theory, |
| | the *maximal denominator* for filling fractions of their quantum states. |

Generally, we will recover in a novel *non-Lagrangian* way the features of quantum Chern-Simons theory that are traditionally argued starting with the $k$th multiple of the local Lagrangian density $a \wedge \mathrm{d}a$ for a gauge potential 1-form $a$.

**The situation on the 2-Sphere.** Furthermore, consider $K$ solitons on the actual 2-sphere $S^2$. Here, the 2-Cohomotopy moduli space satisfies (cf. [62]):

$$\pi_0 \Omega_K \mathrm{Maps}\big(S^2, S^2\big) \simeq \mathbb{Z}_{2|K|},$$

and the long homotopy fiber sequence induced by point evaluation shows that the generator of this cyclic group is again identified with the basic half-braiding operation:

$$\mathrm{Maps}^*\big(\mathbb{R}^2_{\cup\{\infty\}}, S^2\big) \xrightarrow{\ \text{fiber of...}\ } \mathrm{Maps}\big(S^2, S^2\big) \xrightarrow{\ \overset{\text{point-}}{\text{evaluation}}\ } S^2\,,$$

$$\underbrace{\pi_2\big(S^2\big)}_{\mathbb{Z}} \xrightarrow{\ 2K\ } \underbrace{\pi_0\Omega_K \mathrm{Maps}^*\big(\mathbb{R}^2_{\cup\{\infty\}}, S^2\big)}_{\mathbb{Z}} \longrightarrow \underbrace{\pi_0\Omega_K \mathrm{Maps}\big(S^2, S^2\big)}_{\mathbb{Z}_{2|K|}} \longrightarrow \underbrace{\pi_1\big(S^2\big)}_{1}$$

With flux-quantized fields being equipped with a classifying space $\mathcal{A}$, there is a neat way to directly obtain the topological quantum observables – via the following observation:

**Topological flux observables in Yang-Mills theory – Theorem** [133, §1]. For $G$-Yang-Mills theory on $\mathbb{R}^{1,1} \times \Sigma^2$, with a choice of Ad-invariant lattice $\Lambda \subset \mathfrak{g}$:

**(i)** Non-perturbative quantization of the algebra of flux observables through the closed surface $\Sigma^2$ is given by the group $C^*$-algebra $\mathbb{C}[-]$ of the Fréchet-Lie group of smooth maps $\Sigma^2 \to G \ltimes (\mathfrak{g}/\Lambda)$.

**(ii)** The corresponding group algebra of topological observables (observing only the connected components of flux) coincides with the Pontrjagin homology algebra of pointed maps $(\mathbb{R}^1 \times \Sigma^2)_{\cup\{\infty\}} \longrightarrow B\big(G \ltimes (\mathfrak{g}/\Lambda)\big)$:

$$\mathbb{C}\Big[C^\infty\big(\Sigma^2, G\big) \ltimes C^\infty\big(\Sigma^2, (\mathfrak{g}/\Lambda)\big)\Big] \underset{\pi_0}{\rightarrowtail} \mathbb{C}\Big[H^0\big(\Sigma^2; G\big) \ltimes H^1\big(\Sigma^2; \Lambda\big)\Big] \simeq H_0\Big(\mathrm{Maps}^*\big((\mathbb{R}^1 \times \Sigma^2)_{\cup\{\infty\}}, B(G \ltimes (\mathfrak{g}/\Lambda))\big); \mathbb{C}\Big).$$

Non-perturbative quantum algebra of observables on flux through $\Sigma^2$     corresponding algebra of topological observables     Pontrjagin homology algebra of moduli space of soliton charges

For example, in electromagnetism, with $G = \mathrm{U}(1)$ and $\Lambda := \mathbb{Z} \hookrightarrow \mathbb{R}$ this gives [133, §2]:

$$\mathbb{C}\Big[\underbrace{H^1(\Sigma^2; \mathbb{Z})}_{\text{electric}} \times \underbrace{H^1(\Sigma^2; \mathbb{Z})}_{\text{magnetic}}\Big] \simeq H_0\Big(\mathrm{Maps}^*\big((\mathbb{R}^1 \times \Sigma^2)_{\cup\{\infty\}}, \underbrace{B\mathrm{U}(1) \times B\mathrm{U}(1)}_{\substack{\text{classifying space for}\\ \text{Dirac flux quantization}}}\big); \mathbb{C}\Big).$$

This allows us to generalize [133, §3,4]:

**Topological flux observables of any higher gauge theory.** For a higher gauge theory flux-quantized in $\mathcal{A}$-cohomology, the quantum algebra of topological flux observables on a spacetime of the form $\mathbb{R}^{1,1} \times \Sigma^{D-2}$ is the Pontrjagin homology algebra of the soliton moduli, hence in $\deg = 0$ is the group algebra of vacuum soliton processes "**on the light-cone**":

$$\begin{aligned}
\mathrm{Obs}_\bullet &:= H_\bullet\Big(\mathrm{Maps}^*\big((\mathbb{R}^1 \times \Sigma^{D-2})_{\cup\{\infty\}}, \mathcal{A}\big); \mathbb{C}\Big) \\[4pt]
&\simeq H_\bullet\Big(\Omega\,\mathrm{Maps}\big(\Sigma^{D-2}, \mathcal{A}\big); \mathbb{C}\Big), \\[4pt]
\mathrm{Obs}_0 &= \mathbb{C}\Big[\pi_0 \Omega\,\mathrm{Maps}\big(\Sigma^{D-2}, \mathcal{A}\big)\Big].
\end{aligned}$$

For this, note that the star-involution is given by the *combination* of complex conjugation (time reversal) and loop reversal (hence $x$-reversal), where $\mathbb{R}^{1,1} \simeq \mathbb{R}\langle t, x \rangle$, and the operator product is given by loop concatenation:

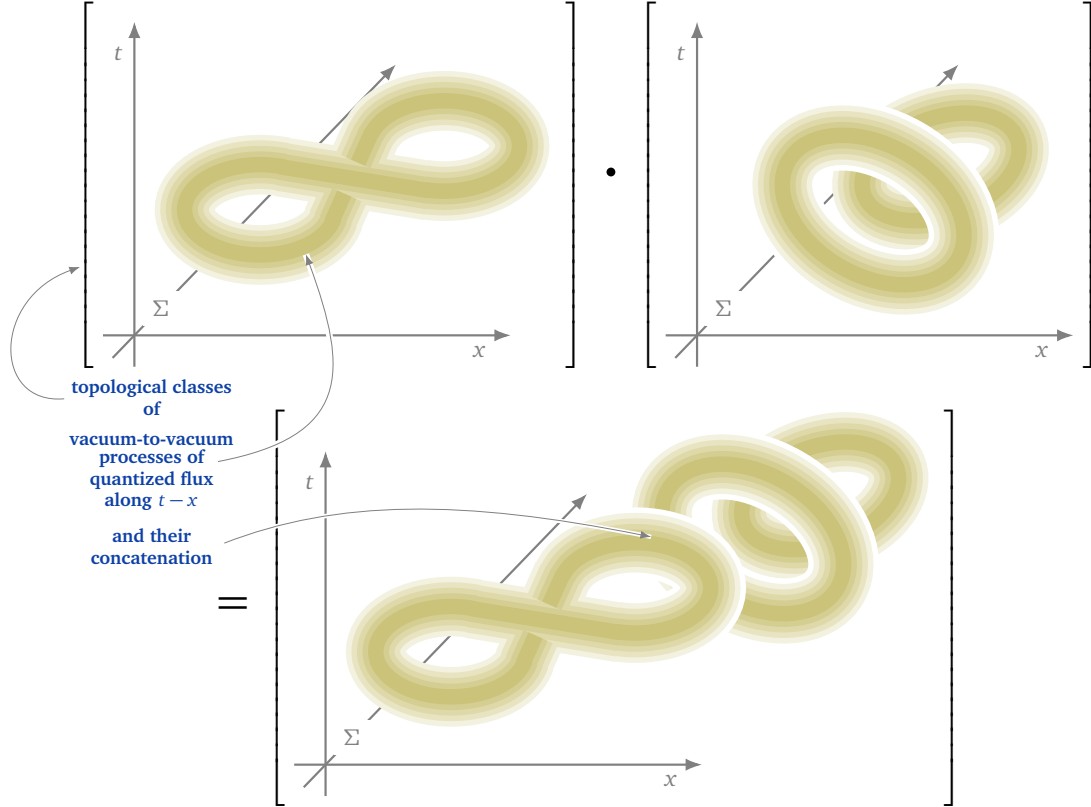

Figure 11: **Vacuum-to-vacuum processes of flux** and their consecutive evolution in light-cone time.

# 4 The topological quantum states

With the theory thus set up, we here turn to analyzing its predictions for topological quantum states and their topological order to be observed on closed surfaces. Remarkably, we find close agreement with the fine-detail of predictions of Abelian Chern-Simons theory, even though the approach here is completely different (non-Lagrangian but properly flux-quantized). Details of the following discussion are spelled out in [138, §3.1-4].

To summarize so far, we have seen that the topological sector of the flux-quantized phase space of solitons on magnetized M5-probes $\Sigma$ wrapping Seifert orbi-singularities is

$$\mathrm{Maps}\begin{pmatrix} \Sigma & \mathbb{C}P^3 \\ \downarrow, & \downarrow \\ X & S^4 \end{pmatrix}^{\mathbb{Z}_2} \simeq \mathrm{Maps}^*\big(\mathbb{R}^2_{\cup\{\infty\}} \wedge S^1, S^2\big) \simeq \Omega_0\,\mathbb{G}\mathrm{Conf}(\mathbb{R}^2) \xrightarrow{[-]} \pi_0\,\Omega_0\,\mathbb{G}\mathrm{Conf}(\mathbb{R}^2) \simeq \mathbb{Z}\,.$$
$$L \longmapsto \#L$$

**topological sector**     **2-Cohomotopy**     **loop space of**     **net**
**of flux-quantized**     **cocycle space**     **group-completed**     **charge**
**phase space**                 **configuration space**

**The topological quantum states of this system** now follow [133] [134, §4] by general algebraic quantum theory:

The gauge-invariant topological **observables** form the (higher) homology of this space

$$\mathrm{Obs}_\bullet := H_\bullet\big(\Omega_0\,\mathbb{G}\mathrm{Conf}(\mathbb{R}^2); \mathbb{C}\big),$$

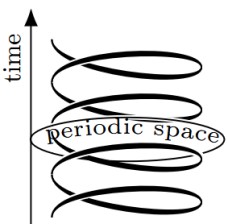

Figure 12: **Worldline of a particle travelling around a compact dimension.** In the limit of high momentum/boost the particle travels on a compactified light-cone.

making a (star-)algebra under concatenation (reversion) of loops — the *Pontrjagin algebra*.

$$\Omega_0 \, \mathbb{G}\mathrm{Conf}(\mathbb{R}^2) \xrightarrow{\substack{\text{loop reversal} \\ \mathrm{rev}}} \Omega_0 \, \mathbb{G}\mathrm{Conf}(\mathbb{R}^2),$$

$$H_\bullet\big(\Omega_0 \, \mathbb{G}\mathrm{Conf}(\mathbb{R}^2); \mathbb{C}\big) \xrightarrow{\substack{\text{Pontr. antipode} \\ \mathrm{rev}_*}} H_\bullet\big(\Omega_0 \, \mathbb{G}\mathrm{Conf}(\mathbb{R}^2); \mathbb{C}\big)$$

$$\xrightarrow[\substack{\text{Hermitian conjugation} \\ \text{of quantum observables}}]{} \quad \overline{(-)} \downarrow \substack{\text{cmplx} \\ \text{cnjgtn}}$$

$$H_\bullet\big(\Omega_0 \, \mathbb{G}\mathrm{Conf}(\mathbb{R}^2); \mathbb{C}\big).$$

This means that time-reversal goes along with the reversal of looping around the M/IIA-circle, whence we are dealing with a version of *discrete light-cone quantization* in their topological sectors.

The basic ordinary (degree=0) observables detect the deformation class of a framed link $L$.

$$
\begin{aligned}
\mathrm{Obs}_0 &\xrightarrow{\sim} \mathbb{C}\big[\pi_0\big(\Omega_0 \mathbb{G}\mathrm{Conf}(\mathbb{R}^2)\big)\big] \xrightarrow{\sim} \mathbb{C}[\mathbb{Z}], \\
\mathcal{O}_L &:= \delta_{[L]} = \delta_{\#L}, \\
\mathcal{O}_L \cdot \mathcal{O}_{L'} &= \delta_{L \sqcup L'} = \delta_{\#L + \#L'}.
\end{aligned}
\tag{8}
$$

Since these observables commute among each other, their *pure* topological quantum states are their (real & positive) algebra homomorphisms:

$$\mathrm{PureQStates}_0 \quad \simeq \quad \Big\{\rho : \mathrm{Obs}_0 \xrightarrow{\text{homo}} \mathbb{C} \;\Big|\; \rho \in \mathrm{MixedQStates}_0\Big\},$$

$$\downarrow \substack{\text{on commuting} \\ \text{observables}}$$

$$\mathrm{MixedQStates}_0 \quad := \quad \Big\{\rho : \mathrm{Obs}_0 \xrightarrow{\text{linear}} \mathbb{C} \;\Big|\; \underset{\mathcal{O} \in \mathrm{Obs}_\bullet}{\forall} \big(\underbrace{\rho(\mathcal{O}^*) = \rho(\mathcal{O})^*}_{\text{reality}}, \underbrace{\rho(\mathcal{O}^* \cdot \mathcal{O}) \geq 0 \in \mathbb{R} \hookrightarrow \mathbb{C}}_{\text{(semi-)positivity}}\big), \underbrace{\rho(1) = 1}_{\text{normalization}}\Big\}.$$

Therefore pure topological states $|m\rangle$ are determined by an **anyonic phase** $\exp(\pi\mathrm{i}/m)$ assigned to any crossing,

accumulating to the exponentiated crossing number

$$
\begin{aligned}
\mathrm{Obs}_0 &\xrightarrow{\langle m| - |m\rangle} \mathbb{C}, \\
\mathcal{O}_L &\longmapsto e^{\frac{\pi\mathrm{i}}{m}\#L}.
\end{aligned}
$$

**The resulting expectation values**

$$\langle m|\mathcal{O}_L|m\rangle \;=\; \exp\!\big(\tfrac{\pi i}{m}\,\#L\big) \;=\; \exp\!\Big(\tfrac{\pi i}{m}\Big(\underbrace{\sum_{i\neq j\in\pi_0(L)}\mathrm{lnk}(L_i,L_j)}_{\substack{\text{linking}\\\text{numbers}}} + \underbrace{\sum_{i\in\pi_0(L)}\mathrm{frm}(L_i)}_{\substack{\text{framing}\\\text{numbers}}}\Big)\Big),$$

are [134, §4] just those of *Wilson loop observables* in "spin" *Chern-Simons theory*, as **expected for Abelian anyons**! For example:

$$\Big\langle m\,\Big|\;\;\Big|\,m\Big\rangle \;=\; \Big\langle m\,\Big|\;\;\Big|\,m\Big\rangle \;=\; \exp\!\big(\pi i\,\tfrac{3}{m}\big).$$

Applying the GNS-construction to such state produces a 1-dimensional Hilbert space

$$\overbrace{\mathbb{C}[\theta,\theta^{-1}]}^{\mathbb{C}[\mathbb{Z}]}\big/\big(e^{\pi i/m}-\theta\big) \simeq \mathbb{C}\,, \tag{9}$$

which is as expected for the quantum states of Abelian Chern-Simons theory on $\mathbb{R}^2_{\cup\{\infty\}}$. (More on this on p 26.)

**Remark.** At this point, $m\in\mathbb{R}\neq 0$ may be irrational, but its rationality will be enforced by requiring compatibility with states on more general domain surfaces, see p. 26 and p. 28.

**Remark.** These *solitonic* anyons are *not* yet the controllable/parameterized defect anyons that could be used for topological braid quantum gates operating by adiabatic movement of anyonic *defects* or (quasi-)*holes*. But the latter arise as defect points among the former, we come to this on p. 30.

**Remark.** The appearance of framed links along just the above lines is known in the condensed matter theory of anyonic defect lines in the 3D "8-band model" ([45, pp 15], following [146]): From this perspective, the Cohomotopy classifying space $S^2$ plays the role of the classifying space for electron band Hamiltonians on a crystal lattice.

**Anyonic topological order on Flux-quantized M5-probes.** We now identify the promised topological order on M5-probes flux-quantized in equivariant twistorial Cohomotopy, by considering M5s wrapping closed surfaces:

**Anyonic quantum observables on closed surfaces.** Consider now a *closed orientable surface* $\Sigma^2_g$ of genus $g\in\mathbb{N}$ to replace the previous factor $\mathbb{R}^2_{\cup\{\infty\}}$ in the brane diagram:

$$\Sigma^{1,5} \;:=\; \mathbb{R}^{1,0} \;\times\; \overset{(\overset{\mathbb{Z}_2}{\curvearrowright}}{\Sigma^2_g} \;\times\; S^1 \;\times\; \overset{(\overset{\mathbb{Z}_2}{\curvearrowright}}{\mathbb{R}^2_{\mathrm{sgn}}}\,.$$

Directly analogous analysis as before gives that the topological quantum observables on the flux-quantized self-dual tensor field form the group algebra of the fundamental group of the 2-cohomotopy moduli space in the $K$th connected component

$$\mathrm{Obs}_0\big(\Sigma^2_g\big) \;:=\; H_0\big(\Omega_K\,\mathrm{Maps}\big(\Sigma^2_g,S^2\big);\,\mathbb{C}\big) \;\simeq\; \mathbb{C}\big[\pi_0\Omega_K\,\mathrm{Maps}\big(\Sigma^2_g,S^2\big)\big]\,, \tag{10}$$

where $K\in\mathbb{N}$ is the degree of the classifying maps, corresponding under the Pontrjagin theorem to a net number of $K$ (anti-)solitons on $\Sigma^2_g$.

**Theorem** (using [62, Thm 1] [86, Thm 1] [78, Cor 7.6]). This group of 2-cohomotopy charge sectors is identified as *twice* the integer Heisenberg group extension (cf. [88]) of $\mathbb{Z}^{2g}$ by $\mathbb{Z}_{2|K|}$:[7]

$$\pi_0 \Omega_K \mathrm{Maps}(\Sigma_g^2, S^2) \simeq \left\{ (\vec{a}, \vec{b}, [n]) \in \mathbb{Z}^g \times \mathbb{Z}^g \times \mathbb{Z}_{2|K|}, \begin{array}{l} (\vec{a}, \vec{b}, [n]) \cdot (\vec{a}', \vec{b}', [n']) = \\ (\vec{a} + \vec{a}', \vec{b} + \vec{b}', [n + n' + \vec{a} \cdot \vec{b}' - \vec{a}' \cdot \vec{b}\,]) \end{array} \right\} =: \widehat{\mathbb{Z}^{2g}} .$$

**Ground state degeneracy.** Hence the observable group-algebra $\mathrm{Obs}_0$ for $g = 1$, $\Sigma_1^2 = T^2$, has generators

$$\left\{ \begin{array}{l} W_a := (1, 0, [0]) \\ W_b := (0, 1, [0]) \\ \zeta := (0, 0, [1]) \end{array} \right\} ,$$

subject to the relations

$$\left\{ \begin{array}{l} W_a \cdot W_b = \zeta^2 W_b \cdot W_a \\ \zeta^{2K} = 1 \\ [\zeta, -] = 0 \end{array} \right\} .$$

This algebra is just the observable algebra expected [148, (5.28)] for anyonic topological order on the torus as described by Abelian "spin" Chern-Simons theory at *lattice norm* $K$ and *level* $k = K/2$.

|  | **symbol** | **in ordinary CS** | **in "spin" CS** | $\exp\left(\frac{\mathrm{i}}{\hbar} S_{CS}\right) =$ |
|---|---|---|---|---|
| CS level | $k$ | $\in \mathbb{N}_{>0}$ | $\in \frac{1}{2}\mathbb{N}_{>0}$ | $e^{2\pi \mathrm{i} k \int A \mathrm{d} A}$ |
| CS lattice | $K \equiv 2k$ | $\in 2\mathbb{N}_{>0}$ | $\in \mathbb{N}_{>0}$ | $e^{\pi \mathrm{i} K \int A \mathrm{d} A}$ |

(11)

The non-trivial irreps have:
– dimension $K$, this being the expected *ground state degeneracy* on the torus,
– are labeled by $\boxed{\nu := p/K}$, $p \in \{1, 2, \cdots, K\}$, as expected for fractional *filling factors*.

**Hilbert space of quantum states on the torus**

$$\mathcal{H}_{T^2} := \mathrm{Span}\left( |[n]\rangle, [n] \in \mathbb{Z}_{|K|} \right) \in \mathrm{Obs}_0(T^2)\mathrm{Modules}, \quad \dim(\mathcal{H}_{T^2}) = K,$$

$$\begin{aligned} W_a |[n]\rangle &:= e^{2\pi \mathrm{i} n \nu} |[n]\rangle, \\ W_b |[n]\rangle &:= |[n+1]\rangle, \\ \zeta |[n]\rangle &:= e^{\pi \mathrm{i} \nu} |[n]\rangle. \end{aligned}$$

**Modular equivariance.** Strikingly, in this construction modular symmetry **is manifest**, since the looped mapping space is canonically acted on by the mapping class group MCG of $\Sigma_g^2$ (cf. [33, §2.1]), simply by precomposition of maps! Inspection of the above theorem (cf. [62, bottom of p 153]) shows that this MCG-action action identifies indeed as the canonical action of $\mathrm{Sp}_{2g}(\mathbb{Z})$ on $\widehat{\mathbb{Z}^{2g}}$.

$$\begin{array}{ccc} \overbrace{\pi_0 \mathrm{Homeos}_{\mathrm{or}}(\Sigma_g^2)}^{\mathrm{MCG}(\Sigma_g^2)} & \subsetneq & \pi_0 \Omega_K \mathrm{Maps}(\Sigma_g^2, S^2) \\ \downarrow{\scriptstyle [33, §6.3]} & & \wr| \\ \mathrm{Sp}_{2g}(\mathbb{Z}) & \subsetneq & \widehat{\mathbb{Z}^{2g}} . \end{array}$$

---

[7] Here $\mathbb{Z}_n := \mathbb{Z}/(n)$ (with $\mathbb{Z}_0 = \mathbb{Z}$) are the (in-)finite cyclic groups.

Hence, we may ask for a lift of the $\widehat{\mathbb{Z}^{2g}}$ action on quantum states to an action of the semidirect product $\widehat{\mathbb{Z}^{2g}} \rtimes \mathrm{Sp}_{2g}(\mathbb{Z})$. For $g = 1$ and even $K$ one readily checks that this gives the modular transformations of states known [95, p 65] from Abelian Chern-Simons theory:

$$m(W) \cdot m\big(\big|[n]\big\rangle\big) = m\big(W\big|[n]\big\rangle\big), \qquad \forall \begin{cases} m & \in \mathrm{Sp}_{2g}(\mathbb{Z}), \\ W & \in \widehat{\mathbb{Z}^{2g}}, \\ \big|[n]\big\rangle \in & \mathcal{H}_g, \end{cases}$$

$$S\big(\big|[n]\big\rangle\big) = \frac{1}{\sqrt{|K|}} \sum_{[\widehat{n}]} e^{2\pi \mathrm{i} \frac{n\widehat{n}}{K}} \big|[\widehat{n}]\big\rangle, \qquad T\big(\big|[n]\big\rangle\big) = e^{\left(-\pi \mathrm{i}/12 + \mathrm{i}\pi \frac{n^2}{K}\right)} \big|[n]\big\rangle.$$

Generally, writing $(\vec{e}_i \in \mathbb{Z}^g)_{i=1}^g$ for the canonical basis vectors, the observable group-algebra $\mathrm{Obs}_0$ for general $g$ has generators

$$\begin{cases} W_a^i := (\vec{e}_i, 0, [0]) \\ W_b^i := (0, \vec{e}_j, [0]), 1 \le i \le g \\ \zeta := (0, 0, [1]) \end{cases},$$

subject to the relations

$$\begin{cases} W_a^i \cdot W_b^j = \delta^{ij} \zeta^2 W_b^j \cdot W_a^i \\ \zeta^{2K} = 1 \\ \text{all other commutators vanish} \end{cases}.$$

Requiring the reps $\mathcal{H}_g$ of this algebra to analogously support modular equivariance requires them to have dimension $|K|^g$ — which is the result expected [95, p 40] for Abelian topological order on $\Sigma_g^2$:

**Hilbert space of quantum states on genus=$g$ surface**
$$\mathcal{H}_{\Sigma_g^2} \in \mathrm{Obs}_0(\Sigma_g^2)\mathrm{Modules}, \quad \dim\big(\mathcal{H}_{\Sigma_g^2}\big) = |K|^g.$$

Here, the generators $W_{a,b}^i$ correspond to the classical generators of the surface's fundamental group. **Oriented closed surfaces** are all obtained (cf. [47, p 100]) by identifying in the regular $4g$-gon, for *genus* $g \in \mathbb{N}$:

(i) all boundary vertices with a single point; and, going clockwise for $r \in \{0, \cdots, g-1\}$,

(iia) the $(4r+1)$st boundary edge with the reverse of the $(4r+3)$rd,

(iib) the $(4r+2)$nd boundary edge with the reverse of the $(4r+4)$th.

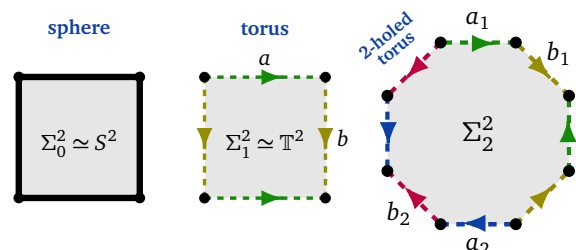

In other words, the homotopy type of the surface sits in a (pointed) homotopy co-fiber sequence of this form:

$$S^1 \xrightarrow{\prod_i [a_i, b_i]} \bigvee_g \big(S_a^1 \vee S_b^1\big) \longrightarrow \Sigma_g^2 \xrightarrow{\delta} S^2,$$

whence its fundamental group is the quotient of the free group on $2g$ generators $(a_i, b_i)_{i=1}^g$ by the normal subgroup generated by that polygon's boundary:

$$\pi_1\big(\Sigma_g^2\big) \simeq \big\langle a_1, b_1, \cdots, a_g, b_g \big\rangle \Big/ \prod_i [a_i, b_i].$$

**2-Cohomotopy moduli of oriented closed surfaces.** Mapping this co-fiber sequence into $S^2$ and applying $\pi_0\Omega_K$, it collapses [62, Prop. 2] to twice [86, Thm 1] the integer Heisenberg central extension of $\mathbb{Z}^{2g}$ by $\mathbb{Z}_{2|g|}$:

$$1 \to \underbrace{\pi_0\Omega_K \operatorname{Maps}(S^2, S^2)}_{\mathbb{Z}_{2|K|}} \xrightarrow{\delta^*} \underbrace{\pi_0\Omega_K \operatorname{Maps}(\Sigma_g^2, S^2)}_{\text{integer Heisenberg group}} \longrightarrow \underbrace{\pi_0\Omega_* \operatorname{Maps}^*(\bigvee_g(S_a^1 \vee S_b^1), S^2)}_{\mathbb{Z}^{2g}} \to 1.$$

**The phase generators.** Hence these integer Heisenberg groups inject into each other as the surfaces are surjected onto each other by collapsing pairs of 1-cycles:

$$\Sigma_g^2 \twoheadleftarrow\xleftarrow{\quad p \quad} \Sigma_{g+1}^2 ,$$

$$\pi_0\Omega_K \operatorname{Maps}(\Sigma_g^2, S^2) \xrightarrow{\pi_1(p^*,K)} \pi_0\Omega_K \operatorname{Maps}(\Sigma_{g+1}^2, S^2)$$

$$\wr\! \vert \qquad\qquad\qquad\qquad\qquad \wr\! \vert$$

$$\widehat{\mathbb{Z}^{2g}} \lhook\joinrel\xrightarrow{\hspace{4cm}} \widehat{\mathbb{Z}^{2(g+1)}} .$$

Thereby, their central generator $\zeta$ represents the previously identified half-braiding operation of solitons on these surfaces. This is the "reason" for the central extension being by $\mathbb{Z}_{2|K|}$ instead of just $\mathbb{Z}_{|K|}$:

The phase generator $\zeta$ does not correspond to full rotations (such as around the square on the right) but to "*particle exchange*" by half-braiding — as expected for anyons.

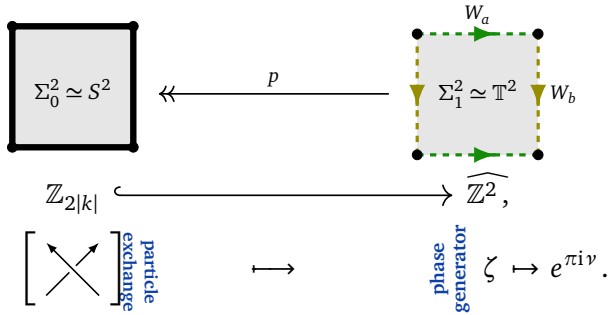

**Non-orientable closed surfaces** are all obtained by identifying in the regular $2h$-gon, for *crosscap number* $h \in \mathbb{N}_{\geq 1}$:

(i) all boundary vertices with a single point and, going clockwise for $r \in \{0, \cdots, h-1\}$,

(ii) the $(2r+1)$st boundary edge with the reverse of the $(2r+2)$nd.

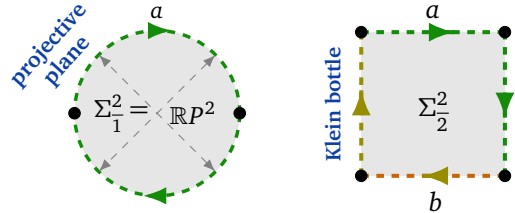

In other words, the homotopy type of the surface sits in a (pointed) homotopy co-fiber sequence of this form:

$$S^1 \xrightarrow{\prod_i a_i^2} \bigvee_h S^1 \longrightarrow \Sigma_{\bar{h}}^2 \xrightarrow{\delta} S^2 .$$

**2-Cohomotopy moduli of non-orientable closed surfaces.** Mapping this co-fiber sequence into $S^2$ and applying $\pi_0\Omega_k$, it induces [62, Prop. 3] an extension of $\mathbb{Z}^{h-1}$ by $\mathbb{Z}_2$ which as such

is trivial [86, Thm. 2]:

$$1 \to \underbrace{\mathrm{coker}\big((\Sigma \textstyle\prod_i a_i^2)^*\big)}_{\mathbb{Z}_2} \xrightarrow{\delta^*} \underbrace{\pi_0 \Omega_k \, \mathrm{Maps}^*\big(\Sigma_{\overline{h}}^2, S^2\big)}_{\mathbb{Z}_2 \times \mathbb{Z}^{h-1}} \longrightarrow \underbrace{\mathrm{ker}\big((\textstyle\prod_i a_i^2)^*\big)}_{\mathbb{Z}^{h-1}} \to 1 \,.$$

Again, the exponent appearing, $h-1$, is just that expected for Abelian Chern-Simons ground state degeneracy, where (cf. [16, (73)]):

$$\dim\big(\mathcal{H}_{\Sigma_{\overline{h}}^2}\big) = |k|^{h-1} \,.$$

# 5 The topological quantum gates

Where the results of the previous §4 establish that on closed (non-punctured) surfaces the predictions of our theory on solitonic anyons and topological order agree with the fine detail of those of Abelian Chern-Simons theory, we now analyze the corresponding predictions for punctured surfaces, and find that the punctures behave like possibly non-Abelian *defect anyons*. Details for the following material are spelled out in [138, §3.5-7].

**Defects via punctured worldvolumes.** It is now immediate to bring *adiabatically movable defect anyons* into the picture, missing in traditional discussion but crucially needed for topological quantum gates (cf. [101, §3]). Namely, we may simply further generalize the surfaces $\Sigma_g^2$ to their *n-punctured* versions, obtained by deleting the positions of a subset of points – thus literally creating defects!

$$\Sigma_{g,n}^2 := \Sigma_{g,n}^2 \setminus \{s_1, \cdots s_n\},$$
$$\text{for } \{s_1, \cdots s_n\} \subset \Sigma_g^2 \,.$$

That these defects are void of the dynamical solitons is elegantly enforced by identifying all their positions with the point-at-infinity (where, recall, the soliton's very nature is to not be present):

domain for solitons in the presence of $n$ defects =
one-point compactification of $n$-punctured surface
$$\big(\Sigma_{g,n}^2\big)_{\cup\{\infty\}} \,, \qquad \text{e.g.:} \quad \big(\Sigma_{0,1}^2\big)_{\cup\{\infty\}} \simeq \mathbb{R}^2_{\cup\{\infty\}} \,.$$

In this generality, our previous brane diagram now is the following, with algebra of soliton quantum observables as shown, by the same kind of argument as before:

$$\Sigma^{1,6} := \mathbb{R}^{1,0} \times \big(\Sigma_{g,n}^2\big)_{\cup\{\infty\}}^{(\mathbb{Z}_2\searrow)} \times S^1 \times \mathbb{R}^2_{\mathrm{sgn}}^{(\mathbb{Z}_2\searrow)} \rightsquigarrow \mathrm{Obs}_0\big(\Sigma_{g,n}^2\big) := H_0\Big(\Omega_K \mathrm{Maps}^*\big((\Sigma_{g,n}^2)_{\cup\{\infty\}}, S^2\big); \mathbb{C}\Big).$$

**Braid group action.** This algebra of observables is faithfully acted on by the mapping class group of the punctured surface – again simply by precomposition of maps. But, with punctures, that group is now an extension (cf. [96, Thm. 3.13]) of the plain mapping class group by the *surface braid group* that acts by ("adiabatically") moving the defects around each other!

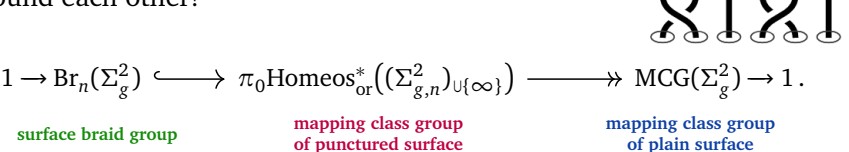

$$1 \to \mathrm{Br}_n(\Sigma_g^2) \hookrightarrow \pi_0 \mathrm{Homeos}^*_{\mathrm{or}}\big((\Sigma_{g,n}^2)_{\cup\{\infty\}}\big) \longrightarrow \mathrm{MCG}(\Sigma_g^2) \to 1 \,.$$

surface braid group     mapping class group of punctured surface     mapping class group of plain surface

In deducing this, we used that $\mathrm{Homeos}^*\big((\Sigma^2_{g,n})_{\cup\{\infty\}}\big) \simeq \mathrm{Homeos}\big(\Sigma^2_{g,n}\big)$, since $(-)_{\cup\{\infty\}}$ is functorial on homeos.

Concretely, observe that the homotopy type of the one-point compactification of a punctured closed surface is the wedge sum of the original surface with $n-1$ circles (cf. [66, p 11], whose graphics we are adapting):

$$\big(\Sigma^2_{g,n}\big)_{\cup\{\infty\}} \simeq \Sigma^2_g \vee \bigvee_{n-1} S^1, \tag{12}$$

This means that the punctures are effectively topology changing defects (as such reminiscent of the *genon*-ic anyon defects considered in [8]) and it implies that their bare quantum observables are (the group algebra) of:

$$\pi_1\mathrm{Maps}^*\big((\Sigma^2_{g,n})_{\cup\{\infty\}}, S^2\big) \simeq \pi_1\mathrm{Maps}^*\big(\Sigma^2_g, S^2\big) \times \mathbb{Z}^{n-1} \underset{g=0}{\simeq} \mathbb{Z}^n, \tag{13}$$

where on the right we recognize that associated with each of the $n$ punctures is one copy of the braid phase observable algebra (8) for the nearby solitonic anyons. This observable algebra of defects gets further enhanced by the corresponding mapping class group:

**Side-remark: Defect-braiding on M5s as a quantum-gravitational effect.** Noting that the mapping class group is equivalently the group of large *diffeomorphisms* of the punctured surface (cf. [33, p 45]),

$$\pi_0\mathrm{Homeos}^*_{\mathrm{or}}\big((\Sigma^2_{g,n})_{\cup\{\infty\}}\big) \simeq \pi_0\mathrm{Diffeos}_{\mathrm{or}}(\Sigma^2_{g,n}), \tag{14}$$

we see that braiding of anyonic defects is reflected in equipping the moduli spaces of cohomotopical charges on the brane worldvolume with the action by diffeomorphisms, hence by passing to the action *groupoid* of moduli quotiented by diffeos:

$$\mathrm{GnrlCovariantModuli}(\Sigma) \simeq \mathrm{Moduli}(\Sigma) /\!\!/ \mathrm{Diffeos}(\Sigma).$$

But this is the hallmark of *generally covariant* systems (cf. [29]), such as are our probe M5-branes.

Ultimately we are to consider surfaces $\Sigma^2_{g,n,b}$ that may feature $b \in \mathbb{N}$ boundary components, and then determine the normal subgroup *pure gauge* diffeomorphisms inside (14) which are trivial on the boundary. The resulting quotient group will be the (ADM/BSM-like) group of diffeos that serve in practice as experimentally observable boundary charges.

**Observables on soliton + defect anyons.** So the covariantized quantum observables on the disk in the presence of $n$ defects is the group algebra of the subgroup of vanishing total framing of the spherical *framed braid group* [84], namely of the *wreath product* $\mathbb{Z} \wr \mathrm{Br}_n$ (cf. [11, §8]) of the soliton monodromy group $\mathbb{Z}$ with the actual braid group $\mathrm{Br}_n(\Sigma^2_0)$ of the defect anyons,

$$\mathbb{C}\big[\overset{\text{solitonic anyons}}{\mathbb{Z}^{n-1}} \rtimes \overset{\text{defect anyons}}{\mathrm{Br}_n}\big] \subset \mathbb{C}\big[\mathbb{Z}^n \rtimes \mathrm{Br}_n\big] = \mathbb{C}\big[\mathbb{Z} \wr \mathrm{Br}_n\big] = \mathbb{C}\big[\mathrm{FBr}_n\big], \tag{15}$$

where the braid group acts on $\mathbb{Z}^n$ through permutation of the factors $\mathrm{Sym}_n$ of the defect anyons:

$$\mathbb{Z}^n \rtimes \mathrm{Br}_n(\Sigma^2_g) \longrightarrow\!\!\!\!\rightarrow \mathbb{Z}^n \rtimes \mathrm{Sym}_n \simeq \Big\{\big((n_i)^n_{i=1}, \sigma\big) \,\big|\, \big((n_\bullet), \sigma\big) \cdot \big((n'_\bullet), \sigma'\big) = \big((n_\bullet + n'_{\sigma(\bullet)}), \sigma\sigma'\big)\Big\}. \tag{16}$$

| Anyons as seen in Cohomotopy | Nature | Number | Braiding |
|---|---|---|---|
| Solitonic anyons | concentrations of flux density | net charge, CS-level: $k$ | by (LC-)time evolution |
| Defect anyons | punctures in worldvolume | $n$ in $\Sigma^2_{g,n}$ | by worldvolume diffeomorphisms |

field solitons/
quasi-particles/
-holes/vortices:
frmd submanifolds

flux-expelling defects:
punctures in the surface

Figure 13: **Solitonic and defect anyons from flux quantized in cohomotopy.** Compare the similarity to the situation of FQH anyons in Figure 5.

Just such *para-statistical* (cf. [153]) wreath-group statistics of defect anyons is seen in condensed matter [45].

**Topologically protected rotation gates.** The above action of $\mathrm{Sym}_n$ on $\mathbb{Z}^{n-1} \subset \mathbb{C}^{n-1}$ is the "*standard representation*" of the symmetric group (the complement of the trivial 1d representation inside the defining permuation representation)

$$\mathcal{H}_{\Sigma^2_{0,1,n}} \;\simeq\; \boxed{\phantom{xxxxx}} \;.$$

Via the cyclic subgroup $\mathbb{Z}_n \subset \mathrm{Sym}_n$ of cyclic permutations, this standard representation contains what in quantum computing are known as q*d*it-based *rotation gates* [162]. For example, for $n = 3$ inspection readily shows that the unitarized standard representation is generated from a Pauli *Z-gate* and a qbit-rotation around the (conventional) $y$-axis, like this:

$$\boxed{\phantom{xx}} \;\simeq\; \left\{ (213) \mapsto \underbrace{\begin{bmatrix} 1 & 0 \\ 0 & -1 \end{bmatrix}}_{Z}, \quad (231) \mapsto \underbrace{\begin{bmatrix} \cos(\alpha) & -\sin(\alpha) \\ \sin(\alpha) & \cos(\alpha) \end{bmatrix}}_{R_y(2\alpha)}, \quad \text{where } \alpha = 4\pi/3 \right\},$$

which implies at once that in the standard rep $\boxed{\phantom{xxxxx}}$ of $\mathrm{Sym}_6$ we find also the corresponding qbit-*controlled* rotation, and so on.

Together with the global phase rotations of solitonic anyons given by the first wreath factor in (15), (controlled), such rotation gates are the workhorse in the quantum Fourier transform [104, §5] [152, §3.2.1] (hence notably in Shor's algorithm for prime faxctorization) and their precision and error protection is a major bottleneck in the implementation of useful quantum algorithms (cf. [44, §III]).

Here we see that our geometric engineering predicts the relevant gates to have topologically error-protected implementation by braiding of defects in FQH systems.

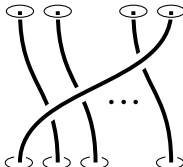

Figure 14: **Topological rotation gates**, obtained by cyclic braiding of defect anyons, combined with the global phase rotations given by braiding of solitonic anyons, would provide intrinsically exact and topologically protected gates of the kind that make up the quantum Fourier transform (in qdit-bases), and with it many other quantum algorithms.

**Side-remark: "Parastatistics" as the most stable anyon braid gates.** Such braid representations on irreps of the symmetric group have traditionally received little to no attention in topological quantum computing (popular are instead solutions to the Knizhnik-Zamolodchikov equation, cf. [131], and of the Yang-Baxter equation). Elsewhere they are discussed as speculative *parastatistics* [63] [113] of fundamental particles instead of as adiabatic Berry-transformations of defect anyons. Therefore Jordan 2010 [75], who is the first to propose symmetric irreps as a model for quantum computation – aka *permutational quantum computing* [76] –, admitted "not [to] worry too much about the physical justification for the model" [75, p 109].

This seems to be a blind spot in the literature: Irreps of $\mathrm{Sym}_n$ are in particular surface braid representations via the surjection $\mathrm{Br}_n(\Sigma_g^2) \twoheadrightarrow \mathrm{Sym}_n$ — regarded as anyon braid gates they are in fact the *most stabilized* such, in that the gate operation is independent not just of isotopy but even of homotopy of the adiabatic transformation.

# 6 Conclusion: Better anyon theory

**New theory of anyonic topological order, engineered on flux-quantized M5s.** In summary, we have seen that global completion by flux-quantization of 11D supergravity with M5-probes (here: in equivariant twistorial cohomotopy – "Hypothesis H"), makes the quantized topological sector of the self-dual tensor field on M5-probes (wrapping Seifert orbi-singularities) reproduce key phenomena of Abelian Chern-Simons theory thought of as an effective field theory for Abelian anyons in fractional quantum Hall (FQH) systems:

**(i) Flux tubes bound to anyons.** The central assumption in the traditional heuristic understanding of the FQHE is that the anyonic solitons have flux quanta "attached" to them [144, p 883]. It is crucially this assumption that motivates and justifies Abelian Chern-Simons theory as an effective field theory for FQH anyons, since variation of the sum of the Abelian Chern-Simons term with the standard source term predicts that the gauge field flux is localized at the source particles (cf. [148, (5.25)] [158, (3.6)]).

In contrast, in the present approach **this effect is a consequence of cohomotopical flux-quantization**, via the Pontrjagin theorem: The classifying map of the 2-Cohomotopy charge identifies an open neighborhood of each anyon with the 2-sphere minus its point at infinity, and the flux density $F_2$ is the pullback of the sphere's volume form along this map (cf. p 51), hence supported on just these open neighborhoods.

**(ii) Anyons subject to each other's Aharonov-Bohm phases.** Traditional discussion furthermore assumes from these attached flux tubes that the anyons must pick up Aharonov-Bohm quantum phases when circling around each other. While this is plausible, rigorous quantum field-theoretic derivation of this statement may not have found much attention.

In contrast, in the approach discussed here, this effect is again a direct consequence of

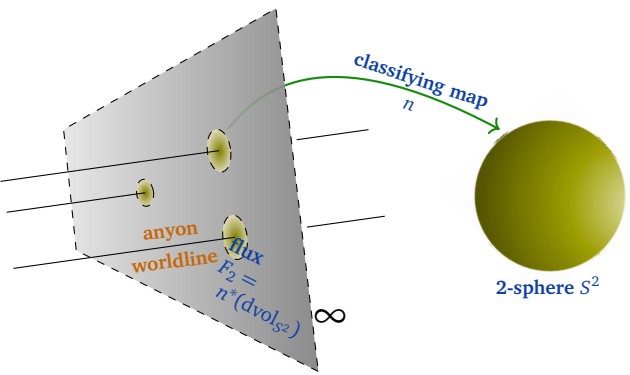

Figure 15: Anyon flux quantized in 2-cohomotopy via Pontrjagin's theorem.

cohomotopical flux-quantization, now via algebro-topological theorems of Segal and others, which serve to identify the cohomotopy charge moduli space with configuration spaces of soliton cores, whose fundamental group reflects the anyon braid phases (and thereby also the ground state degeneracy / topological order).

$$\pi_0 \mathrm{Maps}^*\big(\mathbb{R}^2_{\cup\{\infty\}}, \, S^2 \big) \xrightarrow{\quad \sim \quad} \pi_0 \mathrm{Maps}^*\big(\mathbb{R}^2_{\cup\{\infty\}}, \, B^2\mathbb{Z}\big)$$
$$\text{\footnotesize{$\|\wr$}} \qquad\qquad\qquad\qquad\qquad\qquad \text{\footnotesize{$\|\wr$}}$$
$$\mathbb{Z} \qquad\qquad \textcolor{green}{\text{same net charges...}} \qquad\qquad \mathbb{Z},$$

$$\pi_1 \mathrm{Maps}^*\big(\mathbb{R}^2_{\cup\{\infty\}}, \, S^2 \big) \xrightarrow{\qquad\qquad\qquad} \pi_1 \mathrm{Maps}^*\big(\mathbb{R}^2_{\cup\{\infty\}}, \, B^2\mathbb{Z}\big).$$
$$\text{\footnotesize{$\|\wr$}} \qquad\qquad\qquad\qquad\qquad\qquad \text{\footnotesize{$\|\wr$}}$$
$$\pi_1 \mathbb{G}\mathrm{Conf}(\mathbb{R}^2) \qquad \textcolor{green}{\text{...but different moduli}} \qquad 1$$
$$\text{\footnotesize{config space}} \qquad\qquad\qquad\qquad\qquad\qquad \text{\footnotesize{no structure}}$$

Note how both these effects come about by changing the traditional flux-quantization of the Chern-Simons field from the classifying space for complex line bundles to just its first "cell". This preserves the quantization of charges but makes their moduli exhibit anyonic effects.

$$S^2 \simeq \mathbb{C}P^1 \quad\xhookrightarrow[\textcolor{green}{\text{1st cell inclusion}}]{\qquad\qquad}\quad \mathbb{C}P^\infty \simeq B^2\mathbb{Z}.$$
$$\textcolor{blue}{\substack{\text{classifying space} \\ \text{for 2-Cohomotopy}}} \qquad\qquad\qquad\qquad \textcolor{blue}{\substack{\text{classifying space for} \\ \text{ordinary 2-cohomology}}}$$

**(iii) Topological order.** The traditional way of establishing topological order is by applying geometric quantization to Wilson line observables, with respect to some effective action, which is a somewhat convoluted process involving ad-hoc choices and regularizations. In contrast, in the approach discussed here, the quantum observables obtain immediately, without further choices, from the topological light-cone quantization of the flux-quantized moduli space (as its Pontrjagin homology algebra).

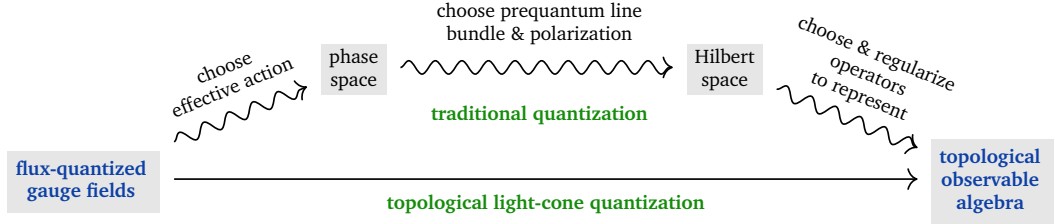

Here the looping $\Omega_k$ that drives this quantum dynamics reflects dependence of moduli on the M/IIA circle.(!)

**(iv) Defect anyons** — as opposed to the solitonic anyons tracing out "Wilson lines" — seem to have previously found little to no attention in quantum Hall theory in general and its effective Abelian Chern-Simons theories in particular. And yet, it is only such classically parameterized and hence, in principle, externally controllable defect anyons which may support braid quantum gates as envisioned in topological quantum computation.

In our approach, defect braiding emerges just as readily as the solitonic anyons, as a mild kind of quantum gravitational effect on M5-worldvolumes having a punctured surface factor space. This may be seen as a theoretical prediction of defect anyons in quantum Hall systems which might inform future search for experimental realization.

---

**Summary of results:**

On super-space, the equations of motion
of **11D supergravity** with magnetic ¹/₂BPS **M5-brane** probes
are equivalent to these Bianchi identities on the super-flux densities:

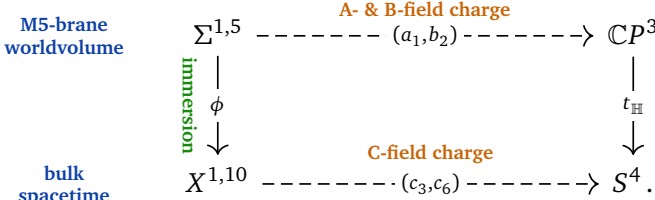

One admissible choice of **flux-quantization** law (the simplest in number of CW cells)
is **twistorial Cohomotopy**, where the charges are classified by dashed maps like this:

$$
\begin{array}{ccc}
\overset{\textbf{M5-brane}}{\textbf{worldvolume}} & \Sigma^{1,5} \; \text{-----}\; \overset{\textbf{A- \& B-field charge}}{(a_1,b_2)} \text{------>} \; \mathbb{C}P^3 \\
& \text{immersion} \downarrow \phi \qquad\qquad\qquad\qquad \downarrow t_{\mathbb{H}} \\
\overset{\textbf{bulk}}{\textbf{spacetime}} & X^{1,10} \; \text{-------}\; \underset{\textbf{C-field charge}}{(c_3,c_6)} \text{------>} \; S^4 \,.
\end{array}
$$

For (very good) $G \subset \mathrm{Sp}(2)$-orbifold domains, these maps are to be $G$-equivariant.

---

This flux-quantization implies a list of topological effects expected in M-theory.
⇒ **Hypothesis H**: This is the right choice of flux-quantization for M-theory.

---

Choosing ("engineering") the M5-probe to be:

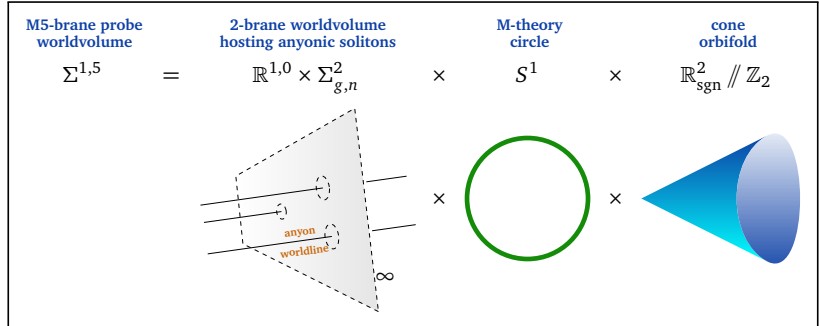

the **moduli space of solitons** becomes: $\mathrm{Moduli} \simeq \mathrm{Maps}^*\!\left(\left(\mathbb{R}^1 \times \Sigma^2_{g,n}\right)_{\cup\{\infty\}}, S^2\right).$

---

The algebra of **topological quantum observables** on theses solitons is:

$$\underset{\substack{\text{topological}\\\text{quantum observables}}}{\mathrm{Obs}_0} \;\; := \;\; \underset{\substack{\text{Pontrjagin homology algebra}}}{H_0\Big(\mathrm{Maps}^*\big(\big(\mathbb{R}^1 \times \Sigma^2_{g,n}\big)_{\cup\{\infty\}}, S^2\big); \mathbb{C}\Big)} \;\; \simeq \;\; \underset{\substack{\text{group algebra}}}{\mathbb{C}\big[\pi_1\mathrm{Maps}^*\big(\Sigma^2_{g,n}, S^2\big)\big]},$$

acted on by large diffeomorphisms (**general covariance** on the brane):

$$1 \longrightarrow \underset{\substack{\text{braid group}}}{\mathrm{Br}_n(\Sigma^2_g)} \lhook\joinrel\longrightarrow \underset{\substack{\text{large diffeomorphism group}}}{\pi_0\mathrm{Homeos}^*_{\mathrm{or}}\big((\Sigma^2_{g,n})_{\cup\{\infty\}}\big)} \longrightarrow\kern-1.5ex\rightarrow \underset{\substack{\text{mapping class group}}}{\mathrm{MCG}(\Sigma^2_g)} \longrightarrow 1\,.$$

---

The corresponding **topological quantum states**:

| | | |
|---|---|---|
| on $\Sigma^2_{0,0} = S^2$ | reflect Abelian braiding of **solitonic anyons** | |
| on $\Sigma^2_{g,0} = \Sigma^2_{1,0}\#\cdots\#\Sigma^2_{1,0}$ | have $k^g$-fold degeneracy: **topological order** | *as for Abelian Chern-Simons QFT* |
| on $\Sigma^2_{1,0} = \mathbb{T}^2$ | exhibit irred $\mathrm{SL}_2(\mathbb{Z})$-**modular equivariance** | |
| on $\Sigma^2_{0,n} = S^2 \setminus \{z^1,\cdots,z^n\}$ | reflect Abelian braiding of *defect anyons* | |

new & needed for
topological quantum gates!

## Engineering of Anyons on M5-Branes via Flux-Quantization

A broad lesson following immediately from our successful geometric engineering of topological qbits is the plausible existence of more exotic anyonic states than traditionally envisioned: Namely the "duality symmetry" [112] [28, §6] of M-theory predicts that any geometrically engineered quantum system has "dual" incarnations with isomorphic quantum observables but entirely different geometric realization, where ordinary space is replaced by more abstract parameter spaces. Notably "T-duality" [151] [37] [55] applied to topological quantum materials has been argued [97] [98] [61] to exchange the roles of ordinary space with that of reciprocal "momentum space".

**(2) Novel experimental pathways towards anyons.** Indeed, while anyonic solitons are traditionally envisioned as being localized in "position space" (meaning that the anyon cores are points in the plane of the crystal lattice) the physical principle behind topological quantum gates — namely [3] [4] [46, p 6] [110, p 50] the *quantum adiabatic theorem* [120] — is unspecific to position space and only requires the material's Hamiltonian to *depend on any continuous parameters* (such as external voltage or strain) varying in any abstract parameter space.

**The general physical conditions for topological quantum gates** given by the *quantum adiabatic theorem,* listed (a) - (e) on the right, are much more general than traditionally considered for anyon braid gates — the latter are only the special case where the parameters are configurations of points in the plane of the 2D crystal lattice.

(a) **Ground state degeneracy** (when frozen at absolute zero, the system still has more than one state to be in, even up to phase).

(b) **Spectral gap** (quanta of energy smaller than a given gap $\epsilon > 0$ cannot excite these ground states).

(c) **Control parameters** (the above properties hold for a range of continuously tunable external parameters).

(d) **Parameter topology** (there exist closed parameter paths that cannot be continuously contracted).

(e) **Local invariance** (continuously deformed parameter paths induce the same transformation on ground states).

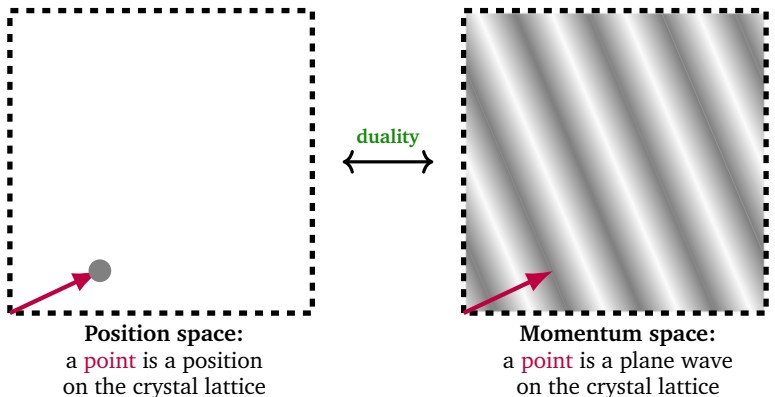

**Position space:**
a point is a position
on the crystal lattice

**Momentum space:**
a point is a plane wave
on the crystal lattice

Figure 16: Crystal Fourier transform establishes a duality between the actual space inhabited by a crystalline sample, the "position space" and its "reciprocal space" of crystal momenta.

This means that, in principle, the possibilities in which anyonic quantum states could arise in the laboratory are far more general than what has been explored to date.

Concretely, a key example of alternative parameters for ground states of a quantum material are points in their reciprocal *momentum space*: This is the space of (quasi-)momenta, hence of wave-vectors for plane quasi-particle waves going through the crystalline material.

We have observed before that candidate anyon-like solitons localized (not in position space but) in momentum space are plausible both theoretically [131] as well as experimentally [161] [147] [74] and may have been hiding in plain sight: as band nodes of (interacting) topological semimetals.

Indeed, momentum space naturally features key properties that are typically assumed for anyon braid gates but remain elusive in position space:

**(i) toroidal base topology** is routinely assumed [154] [156] [90] in order to achieve the required ground-state degeneracy, but is quite unrealistic in position space, even more so when meant to be punctured by defect anyons — while the momentum space of a crystal is automatically a torus (the *Brillouin torus*).

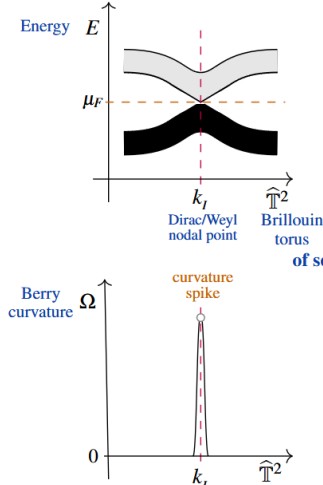

Figure 17: Band nodes in semimetals are submanifolds in the reciprocal space of crystal momenta where the gap between the valence band and the conduction band closes. These band nodes behave like source for Berry curvature.

**(ii) stable defect points** need special engineering in position space but arise automatically in momentum space in the guise of *band nodes* of topological semi-metals [131, Fig. 6]

**(iii) defect point movement** in a *controlled* way is necessary for braid gates but remains elusive in position space, while band nodes in momentum space have already shown to be movable in a variety of systems, by tuning of external parameters (e.g., strain).

The geometric engineering of anyons discussed here goes towards providing also fundamental theoretical underpinning of the possibility of more "exotic" anyon realizations than have traditionally been envisioned.

# 7 Digest for algebraic topologists

We are concerned with algebro-topological phenomena arising when magnetic flux penetrates a semi-conducting surface $\Sigma^2$. The "gauge group" of the electromagnetic field is $G \equiv U(1)$ and *ordinarily* such flux is classified by maps to $BU(1) \simeq \mathbb{C}P^\infty$. The following theorem turns the analysis of this situation first into a problem of differential topology, and then into a problem of algebraic topology.

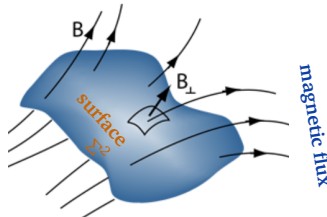

| | | |
|---|---|---|
| $G$ | Lie group ("gauge group") | soliton on $X$ |
| $\mathfrak{g}$ | its Lie algebra | $=$ topological field configuration |
| $C^\infty(\text{-},\text{-})$ | manifold of smooth functions | that vanishes at the ends of $X$ |
| $(\text{-}) \ltimes (\text{-})$ | semidirect product via adjoint | $\Rightarrow$ classified by *pointed* map |
| $\mathbb{C}[-]$ | group convolution $C^*$-algebra | $X_{\cup\{\infty\}} \to BG$ |
| $\pi_0(-)$ | path-connected components | from one-point compactification |

Figure 18: Magnetic flux through a surface is mathematically reflected in (differential) cohomology classes.

**Theorem [133] (Yang-Mills flux quantum observables):** *For ordinary gauge fields on a space-time $\simeq \mathbb{R}^{1,1} \times \Sigma^2$, and for $\Lambda \subset \mathfrak{g}$ an Ad-invariant lattice. the **quantum observables of field flux** through $\Sigma^2$ form the group-convolution $C^*$-algebra*

$$\mathbb{C}\Big[C^\infty\big(\Sigma^2, G \ltimes (\mathfrak{g}/\Lambda)\big)\Big].$$
quantum flux observables

**Commercial-value quantum computing** will require **robust** quantum observables, insensitive to local fluctuations, only depending on **topological sectors** of field configurations.

$$C^\infty\big(\Sigma^2, G \ltimes (\mathfrak{g}/\Lambda)\big) \xrightarrow{\;[-]\;} \pi_0 \, C^\infty\big(\Sigma^2, G \ltimes (\mathfrak{g}/\Lambda)\big).$$
all quantum flux observables            robust topological observables

**Proposition [133] (topological sector observables):** The topological flux quantum observables form the homology Pontrjagin algebra of maps from space to classifying space. (shown

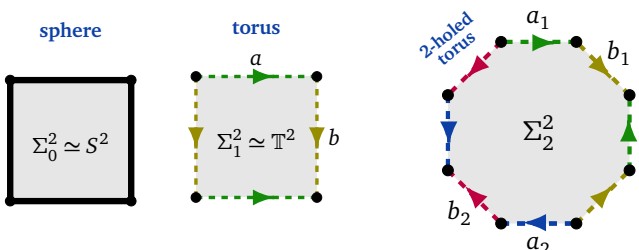

Figure 19: **Fundamental polygons for closed surfaces.**

now assuming $\Lambda = 0$, for simplicity):

$$\underbrace{\mathbb{C}\Big[\pi_0\, C^\infty\big(\Sigma^2, G\big)\Big]}_{\text{Topological flux quantum observables}} \simeq \mathbb{C}\Big[\pi_0\,\mathrm{Maps}\big(\Sigma^2, G\big)\Big] \simeq \underbrace{\mathbb{C}\Big[\pi_1\,\mathrm{Maps}\big(\Sigma^2, BG\big)\Big]}_{\substack{\text{group algebra of fundamental group}\\\text{of maps to classifying space}}} \simeq \underbrace{H_0\Big(\mathrm{Maps}^*\big(\big(\mathbb{R}^1 \times \Sigma^2\big)_{\cup\{\infty\}}, BG\big); \mathbb{C}\Big)}_{\substack{\text{homology Pontrjagin algebra of}\\\text{soliton moduli space}}}.$$

**Example:** $\mathbb{C}\big[\pi_0\,\mathrm{Maps}\big(\Sigma_g^2, \mathrm{U}(1)\big)\big] \simeq \mathbb{C}\big[H^1(\Sigma_g^2; \mathbb{Z})\big] \simeq \boxed{\mathbb{C}\big[\mathbb{Z}^{2g}\big]}$, for $\Sigma_g^2$ an orientable surface of genus=$g$.

**Effective flux of "fractional quantum Hall systems"**(FQH). However, at very low temperature, experiment suggests instead of $\mathbb{Z}^{2g}$ its 2nd $\boxed{\text{integer Heisenberg extension } \widehat{\mathbb{Z}^{2g}}}$

$$\widehat{\mathbb{Z}^{2g}} := \Big\{(\vec{a}, \vec{b}, n) \in \mathbb{Z}^g \times \mathbb{Z}^g \times \mathbb{Z}, (\vec{a}, \vec{b}, n) \cdot (\vec{a}', \vec{b}', n') = \big(\vec{a} + \vec{a}', \vec{b} + \vec{b}', n + n' + \overset{\substack{\text{twice the unit}\\\text{central extension}}}{\vec{a} \cdot \vec{b}' - \vec{a}' \cdot \vec{b}}\big)\Big\},$$

being the observables of an **"effective Chern-Simons field"**, where the center $\mathbb{Z} \hookrightarrow \widehat{\mathbb{Z}^{2g}}$ observes an **anyon braiding phase**.

**Question:** Is there classifying space $\mathcal{A}$ for this effective CS field?

**Answer:** Yes! The 2-sphere $\boxed{S^2 \simeq \mathbb{C}P^1 \hookrightarrow \mathbb{C}P^\infty \simeq B\mathrm{U}(1)}$

**Theorem [62] [86]:** The cofiber presentation of the surface

$$S^1 \xrightarrow{\prod_i[a_i, b_i]} \bigvee_g (S_a^1 \vee S_b^1) \longrightarrow \Sigma_g^2 \longrightarrow S^2,$$

induces short exact sequence exhibiting the Heisenberg extension:

$$1 \longrightarrow \underbrace{\pi_1\mathrm{Maps}\big(S^2, S^2\big)}_{\mathbb{Z}} \longrightarrow \underbrace{\pi_1\mathrm{Maps}\big(\Sigma_g^2, S^2\big)}_{\widehat{\mathbb{Z}^{2g}}} \longrightarrow \underbrace{\pi_1\mathrm{Maps}^*\big(\bigvee_{2g} S^1, S^2\big)}_{\mathbb{Z}^{2g}} \longrightarrow 1.$$

**Question:** Can we identify the center $\mathbb{Z}$ as arising from braiding?

**Answer:** Yes!

**Theorem [134]:** $\mathrm{Maps}^*(S^2, S^2)$ is configurations of charged strings such that $\Omega\mathrm{Maps}^*(S^2, S^2)$ is framed links subject to cobordism, $\pi_1\mathrm{Maps}^*(S^2, S^2)$ generated from framed unknot with 1 braiding

$$\begin{array}{ccc}
\Omega\mathrm{Maps}^*\big(S^2, S^2\big) & \xrightarrow{\;[-]\;} & \pi_3(S^2) \simeq \mathbb{Z}, \\
L & \longmapsto & \#L, \\
\underset{\text{framed link}}{} & & \underset{\substack{\text{linking + framing}\\\text{number}}}{}
\end{array}$$

is CS observable ("Wilson loop").

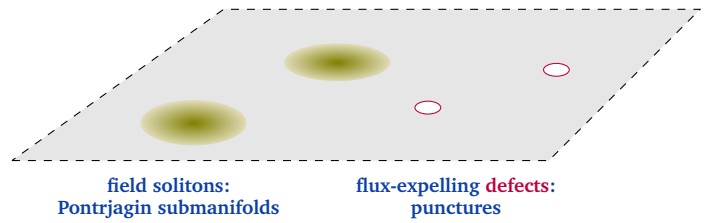

$$\#\left(\begin{smallmatrix}\nearrow & \nwarrow \\ & \times & \end{smallmatrix}\right) = +1, \quad \#\left(\begin{smallmatrix}\nwarrow & \nearrow \\ & \times & \end{smallmatrix}\right) = -1.$$

**Ergo:** Remarkably, topological quantum observables of effective flux in quantum Hall systems is algebro-topologically described by replacing the classifying space $BU(1) \simeq \mathbb{C}P^\infty$ with its 2-skeleton $S^2 \simeq \mathbb{C}P^1$.

**Question 1:** Is there a deeper rationale for such replacement?

**Answer:** Yes [132] [137]: *Hypothesis H*.

**Question 2:** Does this new model make novel predictions?

**Answer:** Yes – *defect anyons* in FQH-systems:

With the classifying space identified for known situations, we find its implications for previously inaccessible cases:

Namely generalize now to *n-punctured* surfaces $\Sigma^2_{g,n}$, reflecting *n defect points* in the semiconductor where the magnetic field is *expelled* (type-I superconducting spots).

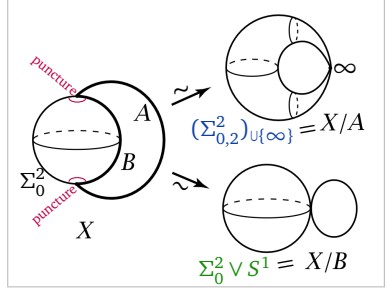

**field solitons:**
**Pontrjagin submanifolds**

**flux-expelling defects:**
**punctures**

Figure 20: Due to the condition that charges vanish at infinity, punctures in the surface are the mathematical reflection of defects in the material where the magnetic flux is expelled.

**Proposition.** The observables are, in this generality:

$$
\begin{aligned}
\mathrm{Obs}_0 &\simeq \mathbb{C}\Big[\pi_1 \mathrm{Maps}^*\big((\Sigma^2_{g,b,n})_{\cup\{\infty\}}, S^2\big)\Big] \\
&\simeq \mathbb{C}\Big[\pi_1 \mathrm{Maps}^*\big(\Sigma^2_{g,b} \vee \bigvee_{n-1} S^1, S^2\big)\Big] \\
&\simeq \mathbb{C}\Big[\pi_1 \mathrm{Maps}^*\big(\Sigma^2_{g,b}, S^2\big) \times \mathbb{Z}^{n-1}\Big] \\
&\underset{\substack{g=0 \\ b=1}}{\simeq} \mathbb{C}\Big[\mathbb{Z}^{n-1}\Big],
\end{aligned}
$$

Figure 21: Topology change due to defects! (cf. [66, p 11]) Different to but not unlike the *genon*-proposal [8].

subject to the **diffeomorphism action** by:

$$1 \longrightarrow \mathrm{Br}_n(\Sigma^2_g) \hookrightarrow \pi_0\mathrm{Homeos}^*_{\mathrm{or}}\big((\Sigma^2_{g,n})_{\cup\{\infty\}}\big) \longrightarrow\!\!\!\!\!\rightarrow \mathrm{MCG}(\Sigma^2_g) \longrightarrow 1.$$

**surface braid group**   **mapping class group of punctured surface**   **mapping class group of plain surface**

Therefore the equivariant quantum states (jargon: "generally covariant") on $\Sigma_{0,n}^2$ are representations of the *wreath product of solitonic and defect phases*:

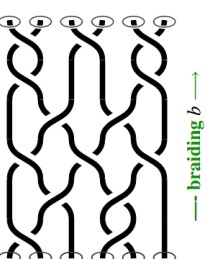

$$\mathbb{Z} \wr \mathrm{Br}_n(\Sigma_0^2) \;=\; \mathbb{Z}^{n-1} \rtimes \mathrm{Br}_n(\Sigma_0^2) \;\twoheadrightarrow\; \mathbb{Z}^{n-1} \rtimes \mathrm{Sym}_n\,.$$

Such *braid representations for defects* have not previously been derived for FQH systems – but are just what is needed for the grand goal of *topological quantum gates*: programmable unitary transformations of quantum systems, insensitive to continuous deformations (hence to noise!)

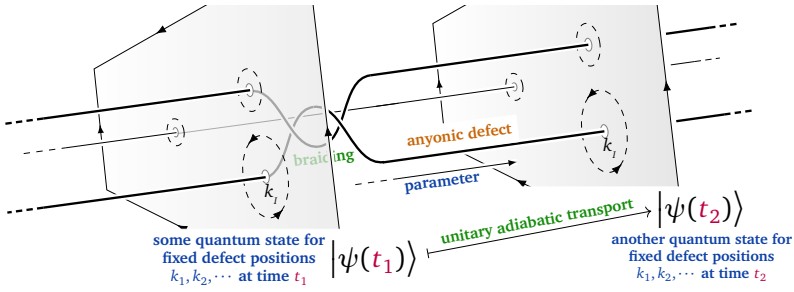

Figure 22: The quantum adiabatic theorem asserts that the sufficient slow (adiabatic) braiding of worldlines of anyonic defects in a 2-dimensional quantum material acts on the gapped ground states by unitary braid representation operators.

Concretely, the action on $\mathbb{Z}^{-1} \subset \mathbb{C}^{n-1}$ is that of the *standard representation*, the complement of the trivial 1d rep inside the defining permutation representation of $\mathrm{Sym}_n$. This yields what are known as *controlled qdit-rotation gates*, the workhorse of quantum algorithms & the bottleneck for noise-protection, here topologically protected as cylic defect braidings:

$$\text{e.g.: } \boxed{\square\square} \;\simeq\; \left\{ (213) \mapsto \overbrace{\begin{bmatrix} 1 & 0 \\ 0 & -1 \end{bmatrix}}^{Z}, \; (231) \mapsto \overbrace{\begin{bmatrix} \cos(\alpha) & -\sin(\alpha) \\ \sin(\alpha) & \cos(\alpha) \end{bmatrix}}^{R_y(2\alpha)} \text{where} \atop \alpha = 4\pi/3 \right\}.$$

**Conclusion & Outlook.** With non-linear flux-quantization laws taken into account in physics, substantial algebraic topology reveals previously unrecognized phenomena potentially visible in experiment and relevant for quantum technology (potentially a more fruitful commercial AlgTop-application than topological data analysis). This opens the opportunity to make AlgTop research inform quantum technology.

**Symmetry-protection and Equivariant homotopy theory.** A noteworthy class of open problems in this regard is the generalization of all of the phenomena discussed here from spaces to *G-spaces* equipped with continuous actions of a finite group, with *G*-equivariant maps between them.

- On the physics side this corresponds to the generic situation of *G-symmetry protected* topological materials (see pointers in [131, §2.3]) particularly important for crystalline symmetry in "anomalous" quantum Hall systems [139].

- On the math side this corresponds to enhancing the flux quantization laws to exotic *equivariant cohomology*, specifically to equivariant Cohomotopy (cf. [127, §4.5] [124, §6]).

Concretely for $G \subsetneq \Sigma_{g,n}^2$ a surface equipped with (crystalline) $G$-symmetry action for $G \subset \mathrm{Pin}(2)$ a finite subgroup, and understanding the canonical $\mathrm{Pin}(2) \hookrightarrow \mathrm{Spin}(3) \twoheadrightarrow \mathrm{SO}(3)$-action on $S^2 \simeq S(\mathbb{R}^3)$, the $G$-symmetry protected enhancement of the above algebra of quantum observables will be formed with the subspace $\mathrm{Map}(-,-)^G \subset \mathrm{Map}(-,-)$ of $G$-equivariant maps

$$\underset{\substack{\text{topological quantum observables} \\ \text{in } G\text{-symmetry protected material}}}{} \quad G\mathrm{Obs}_0 \;:=\; \mathbb{C}\Big[\pi_1 \mathrm{Map}\big(\Sigma_{g,n}^2, S^2\big)^G\Big]. \quad \underset{\substack{\text{fundamental group algebra of} \\ G\text{-equivariant mapping space}}}{}$$

While the analog of the above theorem for non-trivial such $G$ actions remains open, this formula reduces a great deal of subtle physics of topologically ordered quantum materials to a precise question in pure algebraic topology.

**Vista: Homotopy Quantum Logic.** Let us shift gears. We have seen that:

> Topological quantum states $\mathcal{H}_\Sigma$ of solitonic field fluxes with classifying space $\mathcal{A}$ on spacetime domain $\mathbb{R}^{1,1} \times \Sigma$
>
> form representations of $\pi_1$ of the soliton moduli space
>
> $\mathrm{Fields}_\Sigma := \mathrm{Maps}^*\big(\Sigma, \mathcal{A}\big) /\!\!/ \mathrm{Aut}^*(\Sigma)$

This is remarkable because such representations are equivalent to *vector bundles* $\mathcal{H}_\Sigma$ on $\mathrm{Fields}_\Sigma$ *with flat connections* $\nabla$, that is *local systems* on moduli with the homotopy type of $\mathrm{Fields}_\Sigma$ understood as an $\infty$-groupoid, (physics newspeak: generalized symmetry) flat vector bundles are equivalently functors $\vdash \mathcal{H}_\Sigma$ to the groupoid $\mathrm{Mod}_{\mathbb{C}}$:

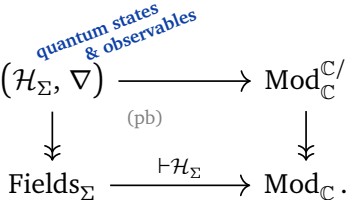

This is the special case of $\infty$-*local systems* [130]: chain complex-bundles with flat $\infty$-connection. These are equivalently $\mathrm{Fields}_\Sigma$- *parameterized module spectra* for the $E_\infty$-ring $H\mathbb{C}$ hence $H\mathbb{C}[\Omega\mathrm{Fields}_\Sigma]$-modules detecting higher structure in the moduli space:

Here

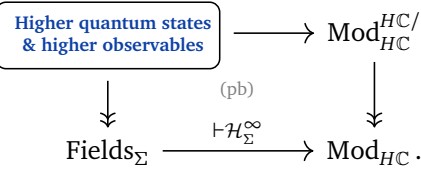

- $H\mathbb{C}$ denotes the *homotopy complex numbers*: the EM-ring spectrum of $\mathbb{C}$;

- $H\mathbb{C}[\Omega\mathrm{Fields}_\Sigma]$ is the *homotopy Pontrjagin algebra* whose $\pi_\bullet$ is $\mathrm{Obs}_\bullet$.

These objects form the **tangent $\infty$-topos** $T\mathrm{Grpd}_\infty$ (over $H\mathbb{C}$), which is [129] [130]:

    **(i)** the arena of parameterized stable homotopy theory,

    **(ii)** categorial semantics of a novel quantum programming language.

Remarkably, this provides an AlgTop angle on an ill-understood but central physics aspect:

> *What exactly is **quantum measurement** of anyonic topological order?*

**Fact.** [129] Given quantum states $\mathcal{H} \in \mathrm{Mod}_{\mathbb{C}}^{\mathrm{Fields}}$,

- a *quantum measurement basis* is

    – a choice of space $W$ (of "*possible worlds*"),

    – a map $W \xrightarrow{i} \mathrm{Fields}$ whose base change is *ambidextrous*: $\quad \mathrm{Mod}_{\mathbb{C}}^W \underset{i^*}{\overset{i_! \simeq i_*}{\underset{\perp \top}{\rightleftarrows}}} \mathrm{Mod}_{\mathbb{C}}^{\mathrm{Fields}},$

    – a $V \in \mathrm{Mod}_{\mathbb{C}}^W$ which (co)induces $\mathcal{H} \simeq i_* V$;

- the *measurement & collapse operation* is is the counit $\quad i^*\mathcal{H} \simeq i^* i_* V \xrightarrow{\mathrm{ret}_V^i} V$.

**Example.** Focusing on Fields := $* /\!/ \pi_0 \mathrm{Homeo}(\Sigma^2_{g,n,b})$, such measurement bases are given by finite index subgroups of $\pi_0 \mathrm{Homeo}(\Sigma^2_{g,n,b})$. There is a rich theory of these, potentially of direct relevance for realizing topological quantum computing...

More on these quantum-information theoretic aspects can be found in [142] [138].

## A   Background on homotopy theory

We collect some notions used in the main text to establish notation and give basic pointers to the literature.

**Homotopy theory** (cf. [145]). For $f_0, f_1 : X \to Y$ a pair of continuous maps between (topological) spaces a *homotopy* $\eta : f_0 \Rightarrow f_1$ is a continuous deformation between them: a continuous map $\eta : [0,1] \times X \to Y$ such that

$$\begin{aligned} \eta(0,x) &= f_0(x), \\ \eta(1,x) &= f_1(x), \end{aligned} \quad \text{denoted} \quad X \overset{f_0}{\underset{f_1}{\Rrightarrow}} Y.$$

For example, a square "homotopy- commutative diagram"

$$\begin{array}{ccc} \Sigma & \overset{b}{\dashrightarrow} & \mathcal{A} \\ \phi \downarrow & {\scriptstyle \eta} & \downarrow p \\ X & \underset{c}{\dashrightarrow} & \mathcal{B}, \end{array} \quad \text{means that} \quad \begin{aligned} \eta &: [0,1] \times \Sigma \to \mathcal{B}, \\ \eta(0,s) &= p(b(s)), \\ \eta(1,s) &= c(\phi(s)). \end{aligned}$$

If one declares – and we do – to work in a "convenient" full sub-category of all topological spaces (such as that of *compactly generated* or of *Delta-generated* topological spaces, cf. [127, p 21, 131]) then the topological space $\mathrm{Maps}(X,Y)$ of all continuous maps $X \to Y$ satisfies the adjointness relation

$$\{P \to \mathrm{Maps}(X,Y)\} \simeq \{P \times X \to Y\}.$$

For $P \equiv [0,1]$, this says that homotopies are equivalently paths in mapping spaces, and that homotopy-classes of maps are the mapping spaces' path-connected components:

$$\pi_0 \mathrm{Maps}(X,Y) \simeq \mathrm{Maps}(X,Y)_{/\mathrm{hmtp}}.$$

Since homotopies are maps themselves, there are homotopies-between-homotopies and ever higher-homotopies.

Thereby, topological spaces constitute a model for **higher categorical symmetry** namely for higher groupoids. As such, they represent both cohomology as well as higher gauge fields in the topological sector.[8]

| **Cohomology** | cocycle | coboundary | higher coboundary | ... |
|---|---|---|---|---|
| homotopy | $X \xrightarrow{f} \mathcal{B}$ | $X \overset{f}{\underset{f'}{\Rrightarrow}} \mathcal{B}$ | $X \overset{f}{\underset{f'}{\underset{\eta'}{\overset{\eta}{\Rightarrow}}}} \mathcal{B}$ | ... |
| **Physics** | field | gauge transf. | higher gauge transf. | ... |

---

[8]Beyond the topological sector, full higher gauge fields are still represented by maps $X \to \mathcal{B}$ etc., only that now $\mathcal{B}$ is no longer just a topological space but a "smooth $\infty$-stack", cf. [35] [42, pp 41].

In this vein, spaces are homotopy-*equivalent* $\mathcal{B} \simeq \mathcal{B}'$ if they are gauge equivalent namely if we have maps

$$\mathcal{B} \underset{g}{\overset{f}{\rightleftarrows}} \mathcal{B}', \quad \text{with} \quad \begin{array}{l} g \circ f \Rightarrow \mathrm{id}_{\mathcal{B}}, \\ f \circ g \Rightarrow \mathrm{id}_{\mathcal{B}'}. \end{array}$$

For example $\mathbb{R}^n \simeq *$ in homotopy theory, reflecting the fact that there is no non-trivial topological sector for fields on $\mathbb{R}^n$.

For actually computing homotopy classes of maps — hence cohomology, hence gauge-equivalence classes of fields in the topological sector — tools from *model category theory* are indispensable, which largely say how to "absorb homotopies into spaces" (cf. [42, §1]).

For example, if $p : \mathcal{A} \to \mathcal{B}$ is a *Serre fibration*, such as a fiber bundle, and $\Sigma$ is a *cell complex*, such as a manifold, then sections-up-to-homotopy of $p$ pulled back to $\Sigma$ are homotopy equivalent to plain sections:

$$\left\{ \begin{array}{ccc} \Sigma & \dashrightarrow^{b} & \mathcal{A} \\ \phi\downarrow & {}^{\nearrow}_{\eta} & \downarrow p \\ X & \xrightarrow{c} & \mathcal{B} \end{array} \right\}_{/\mathrm{hmtp}} \quad \overset{\substack{\Sigma \in \mathrm{Cof} \\ p \in \mathrm{Fib}}}{\simeq} \quad \left\{ \begin{array}{ccc} \Sigma & \dashrightarrow^{b} & \mathcal{A} \\ \phi\downarrow & {/\!\!/} & \downarrow p \\ X & \xrightarrow{c} & \mathcal{B} \end{array} \right\}_{/\mathrm{hmtp}} .$$

**Pointed homotopy theory** (cf. [73, §3]). To reflect the condition that *solitonic fields are localized* in that they *vanish at infinity* we

– equip domain spaces $X$ with a *point at infinity*, $\infty_X \in X$,

– equip classifying spaces $\mathcal{B}$ with a *point representing zero*, $0_{\mathcal{B}} \in \mathcal{B}$,

– require maps $f : (X, \infty_X) \to (\mathcal{B}, 0_{\mathcal{B}})$ to respect these base points

so that maps literally vanish at infinity

$$\begin{array}{ccc} X & \xrightarrow{c} & \mathcal{B} \\ \Uparrow & & \Uparrow \\ \{\infty_X\} & \longrightarrow & \{0_{\mathcal{B}}\}. \end{array}$$

For instance, to make fields on $\mathbb{R}^n$ vanish at infinity, we adjoin its would-be "point at infinity" to it (jargon: "one-point compactification") to obtain $\mathbb{R}^n_{\cup\{\infty\}} \simeq S^n$. On the other hand, if we want fields on some $X$ without a vanishing condition, we may adjoin a *disjoint* point-at-infinity, then pointed maps $X_{\sqcup\{\infty\}} \to \mathcal{B}$ are ordinary $X \to \mathcal{B}$. For example,

| based loop space | free loop space | maps out of contractible |
|---|---|---|
| $\mathrm{Maps}^*(\mathbb{R}^1_{\cup\{\infty\}}, \mathcal{B}) = \Omega\mathcal{B},$ | $\mathrm{Maps}^*(S^1_{\sqcup\{\infty\}}, X) =: \mathcal{L}\mathcal{B},$ | $\mathrm{Maps}^*(\mathbb{R}^1_{\sqcup\{\infty\}}, \mathcal{B}) = \mathcal{B}.$ |

Given a pair of pointed spaces $(X, \infty_X)$, $(Y, \infty_Y)$, in their product space $X \times Y$ any point should be regarded as being at infinity which is so with respect to either factor space; this yields the *smash product*:

$$X \wedge Y := \frac{X \times Y}{\{\infty_X\} \times Y \cup X \times \{\infty_Y\}},$$

to which the sub-space $\mathrm{Maps}^*(-, -)$ of pointed maps is again adjoint:

$$\left\{ P \xrightarrow{\mathrm{pntd}} \mathrm{Maps}^{*/}(X, Y) \right\} \simeq \left\{ P \wedge X \xrightarrow{\mathrm{pntd}} Y \right\}.$$

For example,

$$S^n \wedge S^m \simeq \mathbb{R}^n_{\cup\{\infty\}} \wedge \mathbb{R}^m_{\cup\{\infty\}} \simeq (\mathbb{R}^n \times \mathbb{R}^m)_{\cup\{\infty\}} \simeq S^{n+m},$$

so that, for instance:

$$\mathrm{Maps}^*\big(X \wedge S^1, \mathcal{B}\big) \simeq \mathrm{Maps}^*\big(S^1, \mathrm{Maps}^{*/}(X, \mathcal{B})\big) =: \Omega \, \mathrm{Maps}^*(X, \mathcal{B}).$$

**The differential character map $\mathbf{ch}_{\mathcal{A}}$**, at the heart of flux-quantization in the generality of flux densities with non-linear Bianchi identities:

- takes maps into a classifying space $\mathcal{A}$ (classifying **charges**),

- to maps into the moduli $\infty$-stack of closed $\mathfrak{l}\mathcal{A}$-valued differential forms (classifying corresponding **flux densities**),

- thereby allowing **gauge potentials** to relate local flux densities to global charges.

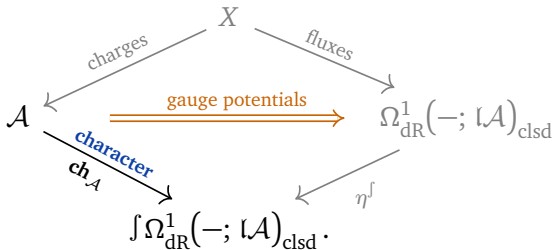

At a high level, this $\mathbf{ch}_{\mathcal{A}}$ is readily described: It is the smooth differential-form model for the $\mathbb{R}$-**rationalization** of $\mathcal{A}$, followed by derived extension of scalars $\mathbb{Q} \to \mathbb{R}$ — as indicated in the following paragraphs.

However, under the hood, this construction makes use of a fair bit of model category-theoretic rational-homotopy theory which we do not have space nor inclination to review here (all details in [42]), whence the following should be ignored by readers without serious background in (rational) homotopy theory — or else taken as motivation to learn it! (Start at [42, §1].)   Here is how it goes:

**Fundamental theorem of homotopy theory.** Regarding (classifying) spaces up to (weak) homotopy equivalence means equivalently to regard them as their $\infty$-*groupoids* (Kan simplicial sets) $\mathrm{Sing}(-)$ of points, paths, 2-paths, etc., in that there is a Quillen equivalence [42, Ex. 1.13]

$$\mathrm{TopSp}_{\mathrm{Qu}} \underset{\mathrm{Sing}}{\overset{\simeq_{\mathrm{Qu}}}{\longleftarrow\!\!\!-\!\!\!-\!\!\!\longrightarrow}} \Delta\mathrm{Set}_{\mathrm{Qu}}.$$

**Fundamental theorem of dg-algebraic rational homotopy theory.** Sending simplicial sets to their dgc-algebras of simplex-wise $\mathbb{Q}$-polynomial differential forms ("piecewise linear", PL) is the left adjoint in a Quillen adjunction [42, Prop. 5.5]

$$\big(\mathrm{dgcAlgs}^{\geq 0}\big)_{\mathrm{proj}}^{\mathrm{op}} \underset{\mathrm{Hom}\big((-),\Omega_{\mathrm{P}\mathbb{Q}\mathrm{LdR}}(\Delta^\bullet)\big)}{\overset{\Omega_{\mathrm{P}\mathbb{Q}\mathrm{LdR}}^\bullet}{\underset{\perp_{\mathrm{Qu}}}{\longleftarrow\!\!\!-\!\!\!-\!\!\!\longrightarrow}}} \Delta\mathrm{Sets}_{\mathrm{Qu}},$$

whose derived adjunction-unit models rationalization of (connected, nilpotent, $\mathbb{Q}$-finite) homotopy types $\mathcal{A}$ [42, Prop. 5.6].

$$\mathcal{A} \xrightarrow{\ \eta_{\mathcal{A}}^{\mathbb{Q}}\ } L^{\mathbb{Q}}\mathcal{A}.$$

**For $\mathbb{R}$-rational homotopy.** The analogous Quillen adjunction with $\mathbb{R}$-polynomial forms

$$\left(\mathrm{dgcAlgs}^{\geq 0}\right)^{\mathrm{op}}_{\mathrm{proj}} \xrightleftharpoons[\mathrm{Hom}\left((-),\Omega_{\mathrm{P}\mathbb{R}\mathrm{LdR}}(\Delta^\bullet)\right)]{\overset{\Omega^\bullet_{\mathrm{P}\mathbb{R}\mathrm{L}}}{\underset{\perp \, \mathrm{Qu}}{\longleftarrow}}} \Delta\mathrm{Sets}_{\mathrm{Qu}},$$

models rationalization followed by derived extension of scalars from $\mathbb{Q}$ to $\mathbb{R}$ (no longer a localization but still denoted like one) [42, Prop. 5.8].

$$\mathcal{A} \xrightarrow{\;\eta^{\mathbb{Q}}_{\mathcal{A}}\;} L^{\mathbb{Q}}\mathcal{A} \longrightarrow L^{\mathbb{R}}\mathcal{A}.$$

Now with $\mathbb{R}$-coefficients, we may equivalently use simplex-wise *smooth* differential forms (*piecewise smooth*, PS)

$$\left(\mathrm{dgcAlgs}^{\geq 0}\right)^{\mathrm{op}}_{\mathrm{proj}} \xrightleftharpoons[\mathrm{Hom}\left((-),\Omega_{\mathrm{PSdR}}(\Delta^\bullet)\right)]{\overset{\Omega^\bullet_{\mathrm{PSdR}}}{\underset{\perp \, \mathrm{Qu}}{\longleftarrow}}} \Delta\mathrm{Sets}_{\mathrm{Qu}}.$$

In fact, we may equivalently use smooth differential forms on simplices times any $\mathbb{R}^n$ [42, Prop. 5.10].

$$\left(\mathrm{dgcAlgs}^{\geq 0}\right)^{\mathrm{op}}_{\mathrm{proj}} \xrightleftharpoons[\mathrm{Hom}\left((-),\Omega_{\mathrm{PSdR}}(\mathbb{R}^n \times \Delta^\bullet)\right)]{\overset{\Omega^\bullet_{\mathrm{PSdR}}}{\underset{\perp \, \mathrm{Qu}}{\longleftarrow}}} \Delta\mathrm{Sets}_{\mathrm{Qu}}.$$

**Taking values in deformations of flux densities.** Via the minimal Sullivan model $\mathrm{CE}(\mathfrak{l}\mathcal{A})$ of $\mathcal{A}$, this derived adjunction takes values in closed smooth $\mathfrak{l}\mathcal{A}$-valued differential forms [42, (9.9)]

$$\Omega^1_{\mathrm{dR}}\left(\mathbb{R}^n \times \Delta^\bullet, \mathfrak{l}\mathcal{A}\right)_{\mathrm{clsd}} := \mathrm{Hom}\left(\mathrm{CE}(\mathfrak{l}\mathcal{A}), \Omega_{\mathrm{dR}}(\mathbb{R}^n \times \Delta^\bullet)\right),$$

which is the value on $\mathbb{R}^n$ of the homotopy-constant $\infty$-stack that is the *shape* $\smallint(-)$ of the sheaf of closed forms [127, Prop. 3.3.48]

$$\smallint \Omega^1_{\mathrm{dR}}\left(-; \mathfrak{l}\mathcal{A}\right)_{\mathrm{clsd}} \in \mathrm{Sh}_\infty(\mathrm{CartSp}).$$

In total, regarding also $\mathcal{A} \in \mathrm{Sh}_\infty(*) \xrightarrow{\mathrm{Disc}} \mathrm{Sh}_\infty(\mathrm{CartSp})$, this establishes the *differential character* map as promised [42, Def. 9.2]

$$\mathcal{A} \xrightarrow{\;\mathbf{ch}_{\mathcal{A}}\;} \smallint \Omega^1_{\mathrm{dR}}\left(-; \mathfrak{l}\mathcal{A}\right)_{\mathrm{clsd}}.$$

# B   Background on TED cohomotopy

**Gauge potentials in twistorial Cohomotopy — and the Green-Schwarz mechanism.** Consider the Whitehead $L_\infty$-algebra of the twistor fibration $\mathbb{C}P^3 \xrightarrow{t_{\mathbb{H}}} \mathbb{H}P^1 \simeq S^4$,

$$\mathrm{CE}\left(\mathfrak{l}_{S^4}\mathbb{C}P^3\right) = \mathbb{R}_{\mathrm{d}}\left[\begin{array}{c} f_2 \\ h_3 \\ g_4 \\ g_7 \end{array}\right] \Bigg/ \left(\begin{array}{rcl} \mathrm{d}\,f_2 &=& 0 \\ \mathrm{d}\,h_3 &=& g_4 + f_2\,f_2 \\ \mathrm{d}\,g_4 &=& 0 \\ \mathrm{d}\,g_7 &=& \tfrac{1}{2}g_4\,g_4 \end{array}\right),$$

and bigons parameterized like this:
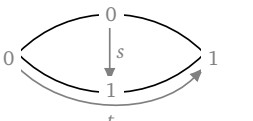
.

**Theorem** ( [53, p 23] [54, §4.1]). Given a manifold $U_i$ (generically: a coordinate chart):

**(i)** Closed $\iota_{s^4}\mathbb{C}P^3$-valued differential forms are in natural bijection with **flux densities** of this form:

$$
\left\{
\begin{array}{c}
U_i \\
| \\
(F_2, H_3, G_4, G_7) \\
\downarrow \\
\Omega^1_{\mathrm{dR}}\big(-; \iota_{s^4}\mathbb{C}P^3\big)_{\mathrm{clsd}}
\end{array}
\right\}
\quad
\begin{array}{c}
i_0 \circ p_0 = \mathrm{id} \\
\xrightarrow{\;\;p_0\;\;} \\
\underset{p_0 \circ i_0 = \mathrm{id}}{\overset{i_0}{\longleftarrow}}
\end{array}
\quad
\left\{
\begin{array}{l|l}
F_2 \in \Omega^2_{\mathrm{dR}}(U_i) & \mathrm{d}\,F_2 = 0 \\
H_3 \in \Omega^3_{\mathrm{dR}}(U_i) & \mathrm{d}\,H_3 = G_4 + \color{blue}{F_2\,F_2} \\
G_4 \in \Omega^4_{\mathrm{dR}}(U_i) & \mathrm{d}\,G_4 = 0 \\
G_7 \in \Omega^7_{\mathrm{dR}}(U_i) & \mathrm{d}\,G_7 = \tfrac{1}{2}\,G_4\,G_4
\end{array}
\right\}.
$$

**(ii)** Given one of these, its set of coboundaries (null-concordances) naturally retracts onto the set of **gauge potentials** of this form:

$$
\left\{
\begin{array}{c}
U_i \xrightarrow{\hspace{1.5cm}} * \\
| \quad {\color{orange}\swarrow}^{(\widehat{F}_2, \widehat{H}_3, \widehat{G}_4, \widehat{G}_7)} \; \downarrow 0 \\
(F_2, H_3, G_4, G_7) \\
\downarrow \\
\Omega^1_{\mathrm{dR}}\big(-; \iota_{s^4}\mathbb{C}P^3\big)_{\mathrm{clsd}} \xrightarrow{\eta^{\smallint}} \smallint\Omega^1_{\mathrm{dR}}\big(-; \iota_{s^4}\mathbb{C}P^3\big)_{\mathrm{clsd}}
\end{array}
\right\}
\;\;
\begin{array}{c}
\xrightarrow{\;\;p_1\;\;} \\
\underset{p_1 \circ i_1 = \mathrm{id}}{\overset{i_1}{\longleftarrow}}
\end{array}
\;\;
\left\{
\begin{array}{l|l}
A_1 \in \Omega^1_{\mathrm{dR}}(U_i) & \mathrm{d}A_1 = F_2 \\
B_2 \in \Omega^2_{\mathrm{dR}}(U_i) & \mathrm{d}B_2 = H_3 - C_3 - A_1 F_2 \\
C_3 \in \Omega^3_{\mathrm{dR}}(U_i) & \mathrm{d}C_3 = G_4 \\
C_6 \in \Omega^6_{\mathrm{dR}}(U_i) & \mathrm{d}G_6 = G_7 - \tfrac{1}{2} C_3 G_4
\end{array}
\right\},
$$

$$
\left(
\begin{array}{rcl}
\widehat{F}_2 &:=& t\,F_2 + \mathrm{d}t\,A_1 \\
\widehat{H}_3 &:=& t\,H_3 + \mathrm{d}t\,B_2 + \color{blue}{(t^2 - t)A_1 F_2} \\
\widehat{G}_4 &:=& t\,G_4 + \mathrm{d}t\,C_3 \\
\widehat{G}_7 &:=& t^2\,G_7 + 2t\,\mathrm{d}t\,C_6
\end{array}
\right)
\quad \xrightarrow{\;\;\;\;} \quad \xleftarrow{\;\;\;\;} \quad
\left(
\begin{array}{rcl}
A_1 &:=& \int_{[0,1]} \widehat{F}_2 \\
B_2 &:=& \int_{[0,1]} \big(\widehat{H}_3 - \big(\int_{[0,-]}\widehat{F}_2\big)\widehat{F}_2\big) \\
C_3 &:=& \int_{[0,1]} \widehat{G}_4 \\
C_6 &:=& \int_{[0,1]} \big(\widehat{G}_7 - \tfrac{1}{2}\big(\int_{[0,-]}\widehat{G}_4\big)\widehat{G}_4\big)
\end{array}
\right).
$$

**(iii)** Given a pair of these, the set of higher coboundaries (2nd-order concordances) between them naturally retracts onto the set of **gauge transformations** of this form:

$$
\left\{
\begin{array}{c}
(\widehat{F}_2, \widehat{H}_3, \widehat{G}_4, \widehat{G}_7) \\
{\color{orange}\overset{\Rrightarrow}{\Big(\!\!\begin{smallmatrix}\widehat{\widehat{F}}_2, \widehat{\widehat{H}}_3 \\ \widehat{\widehat{G}}_4, \widehat{\widehat{G}}_7\end{smallmatrix}\!\!\Big)}} \\
0 \xrightarrow{\hspace{2cm}} (F_2, H_3, G_4, G_7) \\
(\widehat{F}'_2, \widehat{H}'_3, \widehat{G}'_4, \widehat{G}'_7)
\end{array}
\right\}
\;\;
\begin{array}{c}
\xrightarrow{\;\;p_2\;\;} \\
\underset{p_2 \circ i_2 = \mathrm{id}}{\overset{i_2}{\longleftarrow}}
\end{array}
\;\;
\left\{
\begin{array}{l|l}
\alpha_0 \in \Omega^0_{\mathrm{dR}}(U_i) & \mathrm{d}\,\alpha_0 = A'_1 - A_1 \\
\beta_1 \in \Omega^1_{\mathrm{dR}}(U_i) & \mathrm{d}\,\beta_1 = B'_2 - B_2 + \gamma_2 + \alpha_0 F_2 \\
\gamma_2 \in \Omega^2_{\mathrm{dR}}(U_i) & \mathrm{d}\,\gamma_2 = C'_3 - C_3 \\
\gamma_5 \in \Omega^5_{\mathrm{dR}}(U_i) & \mathrm{d}\,\gamma_5 = C'_6 - C_6 - \tfrac{1}{2} C'_3 C_3
\end{array}
\right\},
$$

$$
\left(
\begin{array}{l}
\widehat{\widehat{F}}_2 := t\,F_2 + \mathrm{d}t\,A_1 + s\,\mathrm{d}t\big(A'_1 - A_1\big) - \mathrm{d}s\,\mathrm{d}t\,\alpha_0 \\
\widehat{\widehat{H}}_3 := t\,H_3 + \mathrm{d}t\,B_2 + s\,\mathrm{d}t\,(B'_2 - B_2) - \mathrm{d}s\,\mathrm{d}t\,\beta_1 \\
\quad + \color{blue}{(t^2 - t)A_1 F_2 + (t^2 - t)s(A'_1 - A_1)F_2} \\
\quad + \color{blue}{(t^2 - t)\mathrm{d}s\,\alpha_0 F_2} \\
\widehat{\widehat{G}}_4 := t\,G_4 + \mathrm{d}t\,C_3 + s\,\mathrm{d}t\big(C'_3 - C_3\big) - \mathrm{d}s\,\mathrm{d}t\,\gamma_2 \\
\widehat{\widehat{G}}_7 := t^2\,G_7 + 2t\,\mathrm{d}t\,C_6 + 2st\,\mathrm{d}t(C'_6 - C_6) \\
\quad - 2\mathrm{d}s\,t\,\mathrm{d}t\big(\gamma_5 + \tfrac{1}{2}\gamma_2 C_3\big)
\end{array}
\right)
\; \xrightarrow{\;\;} \; \xleftarrow{\;\;} \;
\left(
\begin{array}{l}
\alpha_0 := \int_{s\in[0,1]}\int_{t\in[0,1]} \widehat{\widehat{F}}_2 \\
\beta_1 := \int_{s\in[0,1]}\int_{t\in[0,1]} \big(\widehat{\widehat{H}}_3 - \big(\int_{t'\in[0,-]}\widehat{\widehat{F}}_2\big)\widehat{\widehat{F}}_2\big) \\
\gamma_2 := \int_{s\in[0,1]}\int_{t\in[0,1]} \widehat{\widehat{G}}_4 \\
\gamma_5 := \int_{s\in[0,1]}\int_{t\in[0,1]} \big(\widehat{\widehat{G}}_7 - \tfrac{1}{2}\big(\int_{t'\in[0,-]}\widehat{\widehat{G}}_4\big)\widehat{\widehat{G}}_4\big) \\
\quad - \tfrac{1}{2}\gamma_2 C_3
\end{array}
\right).
$$

Notice the expression for flux density subject to an (Abelian) Green-Schwarz mechanism:

$$
H_3 = \mathrm{d}\,B_2 + \color{blue}{A_1 F_2} + C_3\,.
$$

*Proof.* With the blue terms discarded, this is the statement of [53, p 23] [54, §4.1]. We compile the full argument.

SciPost Phys. Lect. Notes 107 (2025)

To see that $p_1$ is well-defined:

    – for $C_3, C_6$ this is [53, (70)],

    – for $A_1$ it works just as for $C_3$,

    – for $B_2$ we compute, in generalization of [54, below (138)], like this:

$$
\begin{aligned}
\mathrm{d}B_2 &\equiv \mathrm{d}\int_{[0,1]}\Big(\widehat{H}_3 - \big(\textstyle\int_{[0,-]}\widehat{F}_2\big)\widehat{F}_2\Big) \\
&= \underbrace{\iota_1^*\Big(\widehat{H}_3 - \big(\textstyle\int_{[0,-]}\widehat{F}_2\big)\widehat{F}_2\Big)}_{H_3 - A_1 F_2} - \underbrace{\iota_0^*\Big(\widehat{H}_3 - \big(\textstyle\int_{[0,-]}\widehat{F}_2\big)\widehat{F}_2\Big)}_{=0} - \int_{[0,1]}\underbrace{\mathrm{d}\Big(\widehat{H}_3 - \big(\textstyle\int_{[0,-]}\widehat{F}_2\big)\widehat{F}_2\Big)}_{\widehat{G}_4} \\
&= H_3 - A_1 F_2 - C_3\,.
\end{aligned}
$$

To see that $i_1$ is well-defined:

    – for $\widehat{G}_4, \widehat{G}_7$ this is [53, (72)],

    – for $\widehat{F}_2$ it works just as for $\widehat{G}_4$,

    – for $\widehat{H}_3$ we compute, in generalization of [54, further below (138)], as follows:

$$
\left.
\begin{aligned}
&\mathrm{d}\big(t H_3 + \mathrm{d}t\, B_2 + (t^2 - t)A_1 F_2\big) \\
&= \mathrm{d}t\, H_3 + t G_4 + t F_2 F_2 \\
&\quad - \mathrm{d}t\, H_3 + \mathrm{d}t\, C_3 + {\color{blue}\mathrm{d}t\, A_1 F_2} \\
&\quad + \mathrm{d}\big((t^2 - t)A_1 F_2\big)
\end{aligned}
\right\}
\quad\text{hence indeed:}\quad
\mathrm{d}\widehat{H}_3 = \underbrace{t\, G_4 + \mathrm{d}t\, C_3}_{\widehat{G}_4} + \overbrace{\underbrace{(t F_2 + \mathrm{d}t\, A_1)}_{\widehat{F}_2}\underbrace{(t F_2 + \mathrm{d}t\, A_1)}_{\widehat{F}_2}}^{=\mathrm{d}(t^2 A_1 F_2)}\,.
$$

Moreover, it is immediate from inspection that $\iota_1^*\widehat{H}_3 = H_3$ and $\iota_0^*\widehat{H}_3 = 0$.

To see that $p_1 \circ i_1 = \mathrm{id}$:

    – for $C_3, C_6$ this is [53, below (72)],

    – for $A_1$ this works just as for $C_3$,

    – for $B_2$ we immediately compute:

$$
\int_{[0,1]}\Big(\widehat{H}_3 - \big(\textstyle\int_{[0,-]}\widehat{F}_2\big)F_2\Big) = \underbrace{\int_{[0,1]}\mathrm{d}t\, B_2}_{B_2} - \int_{[0,1]}\underbrace{t A_1\, \mathrm{d}t\, A_1}_{=0} = B_2\,.
$$

To see that $p_2$ is well-defined:

– for $\widehat{\widehat{G}}_4, \widehat{\widehat{G}}_7$ this is [53, (74-5)],

– for $\widehat{\widehat{F}}_2$ this works just as for $\widehat{F}_2$,

– for $\widehat{\widehat{H}}_3$ we compute, in generalization of [54, below (140)], as follows:

$$
\begin{aligned}
\mathrm{d}\beta_1 &\equiv \mathrm{d}\int_{s\in[0,1]}\int_{t\in[0,1]}\Big(\widehat{\widehat{H}}_3 - \big(\textstyle\int_{t'\in[0,-]}\widehat{\widehat{F}}_2\big)\widehat{\widehat{F}}_2\Big) \\
&= \iota_{s=1}^*\int_{t\in[0,1]}\big(\widehat{\widehat{H}}_3 - \cdots\big) - \iota_{s=0}^*\int_{t\in[0,1]}\big(\widehat{\widehat{H}}_3 - \cdots\big) - \int_{s\in[0,1]}\mathrm{d}\int_{t\in[0,1]}\big(\widehat{\widehat{H}}_3 - \cdots\big) \\
&= \int_{t\in[0,1]}\iota_{s=1}^*\big(\widehat{\widehat{H}}_3 - \cdots\big) - \int_{t\in[0,1]}\iota_{s=0}^*\big(\widehat{\widehat{H}}_3 - \cdots\big) - \int_{s\in[0,1]}\iota_{t=1}^*\big(\widehat{\widehat{H}}_3 - \cdots\big) \\
&\quad + \int_{s\in[0,1]}\int_{t\in[0,1]}\mathrm{d}\big(\widehat{\widehat{H}}_3 - \cdots\big) \\
&= \int_{t\in[0,1]}\big(\widehat{H}_3' - \cdots\big) - \int_{t\in[0,1]}\big(\widehat{H}_3 - \cdots\big) + \big(\textstyle\int_{s\in[0,1]}\int_{t\in[0,1]}\widehat{\widehat{F}}_2\big)F_2 + \int_{s\in[0,1]}\int_{t\in[0,1]}\widehat{\widehat{G}}_4 \\
&= B_2' - B_2 + \alpha_0 F_2 + \gamma_2\,.
\end{aligned}
$$

To see that $i_2$ is well-defined:

- for $\gamma_2, \gamma_5$ this is [53, (76)],

- for $\alpha_0$ this works just as for $\gamma_2$,

- for $\beta_1$ we compute as follows:

$$\mathrm{d}\left(t\,H_3 + \mathrm{d}t\,B_2 + s\,\mathrm{d}t\,(B_2' - B_2) - \mathrm{d}s\,\mathrm{d}t\,\beta_1\right) = \overbrace{t\,G_4 + \mathrm{d}t\,C_3 + s\,\mathrm{d}t\,(C_3' - C_3) - \mathrm{d}s\,\mathrm{d}t\,\gamma_2}^{\widehat{\widehat{G}}_4}$$
$$+ t\,F_2 F_2 + \mathrm{d}t\,A_1 F_2 + s\,\mathrm{d}t\,(A_1' - A_1)F_2 - \mathrm{d}s\,\mathrm{d}t\,\alpha_0 F_2$$

$$\mathrm{d}\left(\begin{array}{c}(t^2-t)A_1 F_2 + (t^2-t)s(A_1' - A_1)F_2 \\ +(t^2-t)\,\mathrm{d}s\,\alpha_0 F_2\end{array}\right) = \overbrace{t^2 F_2 F_2 + 2t\mathrm{d}t\,A_1 F_2 + 2t\mathrm{d}t\,s(A_1' - A_1) + 2t\mathrm{d}t\,\mathrm{d}s\,\alpha_0 F_2}^{\widehat{\widehat{F}}_2\widehat{\widehat{F}}_2}$$
$$- t\,F_2 F_2 - \mathrm{d}t\,A_1 F_2 - \mathrm{d}t\,s(A_1' - A_1)F_2 - \mathrm{d}t\,\mathrm{d}s\,\alpha_0 F_2$$

$$\mathrm{d}\widehat{\widehat{H}}_3 \qquad = \qquad \widehat{\widehat{G}}_4 + \widehat{\widehat{F}}_2\widehat{\widehat{F}}_2.$$

Moreover, it is immediate from inspection that $\iota_{s=0}^* \widehat{\widehat{H}}_3 = \widehat{H}_3$, $\iota_{s=1}^* \widehat{\widehat{H}}_3 = \widehat{H}_3'$ and $\iota_{t=0}^* = 0$, $\iota_{t=1}^* = H_3$.

To see that $p_2 \circ i_2 = \mathrm{id}$, we directly compute, first

$$\int_{s\in[0,1]}\int_{t\in[0,1]} \widehat{\widehat{G}}_4 = \int_{s\in[0,1]}\int_{t\in[0,1]} (-\mathrm{d}s\,\mathrm{d}t\,\gamma_2) = \gamma_2,$$
$$\int_{s\in[0,1]}\int_{t\in[0,1]} \widehat{\widehat{F}}_2 = \int_{s\in[0,1]}\int_{t\in[0,1]} (-\mathrm{d}s\,\mathrm{d}t\,\alpha_0) = \alpha_0,$$

then

$$\int_{s\in[0,1]}\int_{t\in[0,1]}\left(\widehat{\widehat{G}}_7 - \tfrac{1}{2}\left(\int_{t'\in[0,t]}\widehat{\widehat{G}}_4\right)\widehat{\widehat{G}}_4\right) - \tfrac{1}{2}\gamma_2 C_3 = \int_{s\in[0,1]}\int_{t\in[0,1]}\widehat{\widehat{G}}_7$$
$$- \tfrac{1}{2}\int_{s\in[0,1]}\int_{t\in[0,1]}\left(t\,C_3 + st(C_3' - C_2) + t\mathrm{d}s\,\gamma_2\right)$$
$$\times \left(t\,G_4 + \mathrm{d}t\,C_3 + s\,\mathrm{d}t\,(C_3' - C_3) - \mathrm{d}s\,\mathrm{d}t\,\gamma_2\right) - \tfrac{1}{2}\gamma_2 C_3$$
$$= \left(\gamma_5 + \tfrac{1}{2}\gamma_2 C_3\right)$$
$$\underbrace{-\tfrac{1}{2}C_3\gamma_2 - \tfrac{1}{4}(C_3' - C_3)\gamma_2 + \tfrac{1}{2}\gamma_2 C_3 + \tfrac{1}{4}\gamma_2(C_3' - C_3)}_{0}$$
$$- \tfrac{1}{2}\gamma_2 C_3$$
$$= \gamma_5,$$

and analogously

$$\int_{s\in[0,1]}\int_{t\in[0,1]}\left(\int_{t'\in[0,-]}\widehat{\widehat{F}}_2\right)\widehat{\widehat{F}}_2 = \int_{s\in[0,1]}\int_{t\in[0,1]}\left(tA_1 + st(A_1' - A_1) + t\mathrm{d}s\,\alpha_0\right)$$
$$\times \left(t\,F_2 + \mathrm{d}t\,A_1 + s\,\mathrm{d}t\,(A_1' - A_1) - \mathrm{d}s\,\mathrm{d}t\,\alpha_0\right)$$
$$= \tfrac{1}{2}A_1\alpha_0 + \tfrac{1}{4}(A_1' - A_1)\alpha_0 - \tfrac{1}{2}\alpha_0 A_1 - \tfrac{1}{4}\alpha_0(A_1' - A_1)$$
$$= 0,$$

so that also

$$\int_{s\in[0,1]}\int_{t\in[0,1]}\left(\widehat{\widehat{H}}_3 - \left(\int_{t'\in[0,-]}\widehat{\widehat{F}}_2\right)\widehat{\widehat{F}}_2\right) = \int_{s\in[0,1]}\int_{t\in[0,1]}(-\mathrm{d}s\,\mathrm{d}t\,\beta_1) = \beta_1.$$

$\square$

**Cocycles in differential 2-Cohomotopy and the Abelian Chern-Simons invariant on the 3-Sphere.** Notice that the Bianchi identities encoded by 2-Cohomotopy are the characteristic property of the Abelian Chern-Simons term:

$$\mathrm{CE}(\mathbb{S}^2) \simeq \mathbb{R}_{\mathrm{d}}\begin{bmatrix} f_2 \\ h_3 \end{bmatrix} \Big/ \begin{pmatrix} \mathrm{d}\,f_2 = 0 \\ \mathrm{d}\,h_3 = f_2 f_2 \end{pmatrix} \quad \Rightarrow \quad \Omega^1_{\mathrm{dR}}\big(X; \mathbb{S}^2\big)_{\mathrm{clsd}} \simeq \left\{ \begin{array}{c} F_2 \in \Omega^2_{\mathrm{dR}}(X) \\ H_3 \in \Omega^3_{\mathrm{dR}}(X) \end{array} \;\middle|\; \begin{array}{c} \mathrm{d}\,F_2 = 0 \\ \mathrm{d}\,H_3 = F_2 F_2 \end{array} \right\} .$$

We may bring this out more concretely:

**Gauge-field configurations on $\mathbb{R}^3$ flux-quantized in 2-Cohomotopy** and vanishing in a neighborhood of infinity are cocycles in differential 2-Cohomotopy on $\mathbb{R}^3_{\cup\{\infty\}}$, hence dashed homotopies as shown on the right [132, §3.3].

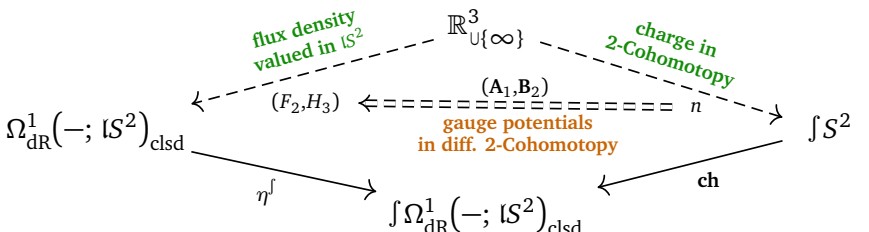

**Theorem.** For each $[n] \in \pi^2\big(\mathbb{R}^3_{\cup\{\infty\}}\big) \simeq \mathbb{Z}$ this exists with $H_3 = 0$ and $[n] = \int_{\mathbb{R}^3} A_1 F_2$ the Chern-Simons invariant.

To see this, first consider:

**Lemma.** *On a smooth manifold $\Sigma$, every cocycle $\alpha$ in rational 3-Cohomotopy is represented by a globally defined differential form $H_3$,*

$$X \xrightarrow[H_3]{\phantom{xxxxxxxxxxx}} \int \Omega^1_{\mathrm{dR}}\big(-; \mathbb{S}^3\big)_{\mathrm{clsd}} .$$

$$\Omega^1_{\mathrm{dR}}\big(-; \mathbb{S}^3\big)_{\mathrm{clsd}} \xrightarrow{\eta^{\int}}$$

*Proof of the Lemma.* Since $\mathbb{S}^3 \simeq \mathbb{B}^3\mathbb{Q}$ this is just the degree=3 case of the statement that cocycles in de Rham hyper-cohomology have global representatives on smooth manifolds (using partitions of unity). $\qquad\square$

*Proof of the Theorem.* Stereographic projection provides a homeomorphism $\mathbb{R}^3_{\cup\{\infty\}} \xrightarrow{\sim} S^3$ which is smooth away from the point at infinity, which we may slightly deform to a smooth degree=1 map that is constant on a neighborhood of infinity. Since $\pi^2(S^3) \simeq \pi_2(S^3) \simeq \mathbb{Z}$ we may find a smooth map $n : S^3 \to S^2$, with compact support away from the base point, so that $\mathbb{R}^3_{\cup\{\infty\}} \to S^3 \xrightarrow{n} S^2$ represents the charge $[n]$.

Now the 2-cohomotopical character map for charges on $S^3$, shown in black, factors as shown in blue (by naturality of rationalization), which furthermore factors as shown in orange (by the above Lemma).

$$\begin{array}{ccccc} \mathbb{R}^3_{\cup\{\infty\}} \to S^3 & \xrightarrow{\;\eta^{\int}\;} & \int S^3 & \xrightarrow{\;n\;} & \int S^2 \\ {\scriptstyle n\cdot\mathrm{dvol}_{S^3}}\downarrow & & {\scriptstyle \mathbf{ch}_{S^3}}\downarrow & & \downarrow{\scriptstyle \mathbf{ch}_{S^2}} \\ \Omega^1_{\mathrm{dR}}(-; \mathbb{S}^3)_{\mathrm{clsd}} & \xrightarrow{\;\eta^{\int}\;} & \int \Omega^1_{\mathrm{dR}}(-; \mathbb{S}^3)_{\mathrm{clsd}} & \xrightarrow{\;(\mathbb{l}n)_*\;} & \int \Omega^1_{\mathrm{dR}}(-; \mathbb{S}^2)_{\mathrm{clsd}} . \end{array} \qquad (17)$$

Hence, to get a differential cocycle as desired, it is sufficient to exhibit gauge potentials $(A_1, B_2)$ encoding a concordance filling the diagram on the right

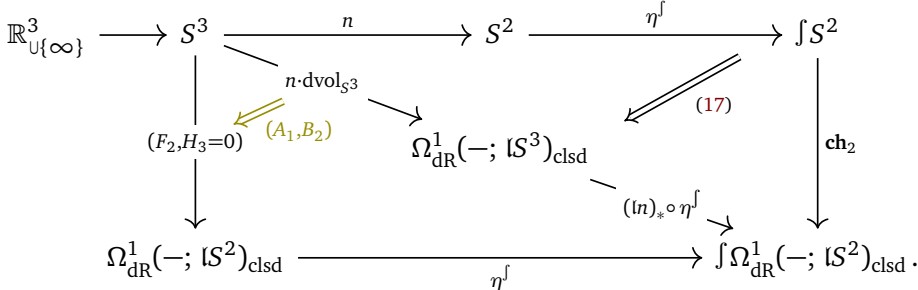

However, since $H^2_{\mathrm{dR}}(S^3) = 0$, and by the *Whitehead integral formula* (cf. [57, p 134] [12, p 228] [38, p 19]) there exists:

$$\begin{cases} A_1 \in \Omega^1_{\mathrm{dR}}(S^3), \\ B_2 \in \Omega^2_{\mathrm{dR}}(S^3), \end{cases} \text{s.t.} \quad \boxed{\begin{aligned} \mathrm{d}A_1 &= F_2 := n^*\mathrm{dvol}_{S^2}, \\ \mathrm{d}B_2 &= n \cdot \mathrm{dvol}_{S^3} - A_1 F_2. \end{aligned}} \tag{18}$$

From this we get the the desired concordance:

$$(0, n \cdot \mathrm{dvol}_{S^3}) \Rightarrow (F_2, 0) : \begin{cases} \widehat{F}_2 := t\,F_2 + \mathrm{d}t\,A_1, \\ \widehat{H}_3 := (t-1)n\,\mathrm{dvol}_{S^3} + \mathrm{d}t\,B_2 + (t^2 - t)A_1 F_2, \end{cases} \begin{aligned} (\widehat{F}_2, \widehat{H}_3)|_{t=0} &= (0, n \cdot \mathrm{dvol}_{S^3}), \\ (\widehat{F}_2, \widehat{H}_3)|_{t=1} &= (F_2, 0), \\ \mathrm{d}\widehat{F}_2 &= 0, \ \mathrm{d}\widehat{H}_3 = \widehat{F}_2 \widehat{F}_2. \end{aligned}$$

$\square$

**Cartesian M5-Probes charged in Cohomotopy.** The equations of motion for a(n orbifolded) cartesian M5-probe demand that the flux $H_3 = \text{const}$ [54, Ex. 3.14], and thus its solitonic vanishing-at-infinity implies $H_3 = 0$. The above theorem says that such solutions still support non-vanishing cohomotopical charge, in fact that the vanishing of $H_3$ forces the charge to be carried by the Chern-Simons invariant of the auxiliary gauge field $\mathbf{A}_1$ that is brought in by the cohomotopical flux quantization.

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
