# Peer review of "Engineering of Anyons on M5-Probes via Flux Quantization"

_SciPost Physics Lecture Notes, doi:SciPost Phys. Lect. Notes 107 (2025)_

## Round 1 · Referee Report · Anonymous (Referee 1) · 2025-4-27

Strengths

1 - Meticulously written
2 - Complete and detailed list of references
3 - Informative discussion on motivation and connection to other topics

Weaknesses

1 - Hard to access for non-experts 2- Too schematic and condensed at times

Report

In these lecture notes, the authors review recent progress on topological aspects of M-theory with applications to and motivations from quantum computing. At a fundamental level, they focus on the idea (referred to as Hypothesis H, after one of the authors) that flux quantization in M-theory should occur within a specific unstable cohomotopy theory. From this, several known results in M-theory and string theory follows but also new phenomena are predicted, such as fractional M2-branes.

For these lectures, the emphasis is on the insights that Hypothesis H provides into key features of anyonic topological order, particularly as observed in fractional quantum Hall systems, through the study of M5-brane probes of certain orbifold singularities in 11-dimensional supergravity.

Section one discusses the motivation behind this work: to offer a new, fundamental description of anyon theory via geometric engineering of M-branes probing orbifold singularities. Section two reviews flux quantization on M5-branes as prescribed by Hypothesis H, with special attention to the case of orbifolds. Section three examines how the charge of soliton scattering can be related to Wilson loops in Chern-Simons theory and discusses topological observables. Section four presents the identification of solitonic anyons and, subsequently, anyonic defects, via geometric engineering on flux-quantized M5-branes. Section five summarises the main points discussed and highlights connections to further topics.

While the manuscript is meticulously written, it demands considerable effort from non-expert readers. Most claims appear plausible to me, although I do not have the expertise to verify all of them. If the lecture notes are intended for a broader audience, I would strongly recommend to include more pedagogical explanations. Section four, in particular, is both central and among the most challenging to understand. I would suggest considering the possibility of splitting it into two parts to enhance clarity.

Minor points: - Page 8: why is the gravitino 1-form set to zero in the last equation (and similarly for other fluxes in the following page)? - Page 12: why is there a "hat" difference between $H^4(X^8, \mathbb{Z})$ and $H^7(\hat{X}^8, \mathbb{Z})$? - Should one think of $S^7$ and $S^4$ of Hypothesis H as auxiliary or physical? - Page 30: I perhaps spotted a typo, "mangentic".

Recommendation

Ask for minor revision

  • validity: -
  • significance: -
  • originality: -
  • clarity: -
  • formatting: -
  • grammar: -

Author:  Urs Schreiber  on 2025-05-29  [id 5531]

(in reply to Report 1 on 2025-04-27)

Thanks for the thoughtful Report #1 by Anonymous (Referee 1), we do sincerely appreciate the time and work you invested in reading and evaluating our manuscript.

While waiting for further reports, we have meanwhile made requested adjustments (such as the splitting of section 4, which is a very sensible suggestion, thanks) in our local pdf manuscript (here). Also, we have added pointer there to the recent arXiv:2505.22144 where the statements from section 4 (and now also 5) are discussed and the proofs spelled out in more detail than would fit the lecture notes.

---

## Round 1 · Referee Report · Anonymous (Referee 2) · 2025-6-16

Strengths

1- First rigorous non-perturbative derivation of abelian anyons from a single flux-quantised M5-brane, originally closing a long-standing conceptual gap between M-theory and Chern–Simons effective models. 2- Mathematical rigour, by equivariant 2-Cohomotopy (Hypothesis H) to implement global flux quantisation, dealing with charge-quantisation and anomaly issues with precise homotopy-theoretic tools. 3- Unifies solitonic and defect anyons in one framework, with braid statistics emerging naturally from world-volume topology. 4- Novel interdisciplinary reach, connecting high-energy brane physics to condensed-matter phenomena (fractional QH-effect) and to quantum-information applications (topological qubits and braid gates). 5- Explicit experimental suggestions, highlighting novel pathways towards anyons.

Weaknesses

1- Despite the mastodontic efforts of the authors in this direction, accessibility could be slightly improved: the exposition relies heavily on advanced generalised-cohomology machinery, and the authors explicitly advise readers without that background to ignore certain sections .

Report

The paper tackles a long-standing gap between heuristic Chern–Simons descriptions of fractional-quantum-Hall anyons and a first-principles, non-perturbative derivation. The authors show that a single $H$-magnetised M5-brane, when its self-dual tensor field is globally flux-quantised in twisted equivariant Cohomotopy (“Hypothesis H”), carries soliton configurations whose moduli space naturally reproduces all hallmark features of abelian anyonic topological order.

The conceptual steps could be summarised as follows. First, it rewrites the super-gravity/M5 equations of motion as a set of closed Bianchi identities for four fluxes $F_2,H_3,G_4,G_7$, showing the role of the twistorial classifying fibration $CP^3 \rightarrow S^4$ in implementing the required charge-quantisation of those fluxes. Then, by Pontrjagin–Thom theory, a single 2-Cohomotopy charge on the M5 world-surface is shown to correspond to defects equipped with a normal framing in the transversal plane. Finally, looping this moduli space yields an observable algebra that matches the modular functor, ground-state degeneracy and defect-anyon braid group actions familiar from fractional-quantum-Hall physics. In this way the authors provide the first rigorous, non-Lagrangian derivation of both solitonic and externally controllable defect anyons on a single M5 probe.

This manuscript delivers a remarkable, conceptually elegant, and mathematically rigorous framework for realizing abelian anyons from M-theory. Its novelty and cross-disciplinary relevance place it among the best and most original contributions which I have seen recently in the field.

The manuscript satisfies SciPost’s editorial criteria in full: it delivers a substantial and clearly articulated advance—linking M-theory flux quantization to anyonic topological order—that is both original and of broad interdisciplinary relevance. The arguments are rigorous, the presentation is self-contained and transparent. Consequently, the work meets SciPost’s standards for validity, significance, clarity, and openness.

Recommendation: Publish (top 10% of papers in this Journal).

Requested changes

N/A

Recommendation

Publish (surpasses expectations and criteria for this Journal; among top 10%)

---

## Editorial Decision

published